# A Game-Theoretic Analysis of the Empirical Revenue Maximization Algorithm with Endogenous Sampling

**Xiaotie Deng**
Center on Frontiers of Computing Studies,
Computer Science Dept.,
Peking University
xiaotie@pku.edu.cn

**Ron Lavi**
Technion - Israel Institute of Technology
ronlavi@ie.technion.ac.il

**Tao Lin**
Center on Frontiers of Computing Studies,
Computer Science Dept.,
Peking University
lin_tao@pku.edu.cn

**Qi Qi**
Hong Kong University of Science and Technology
kaylaqi@ust.hk

**Wenwei Wang**
Alibaba Group
wwangaw@connect.ust.hk

**Xiang Yan**
Shanghai Jiao Tong University
xyansjtu@163.com

## Abstract

The Empirical Revenue Maximization (ERM) is one of the most important price learning algorithms in auction design: as the literature shows it can learn approximately optimal reserve prices for revenue-maximizing auctioneers in both repeated auctions and uniform-price auctions. However, in these applications the agents who provide inputs to ERM have incentives to manipulate the inputs to lower the outputted price. We generalize the definition of an incentive-awareness measure proposed by Lavi et al (2019), to quantify the reduction of ERM's outputted price due to a change of $m \geq 1$ out of $N$ input samples, and provide specific convergence rates of this measure to zero as $N$ goes to infinity for different types of input distributions. By adopting this measure, we construct an efficient, approximately incentive-compatible, and revenue-optimal learning algorithm using ERM in repeated auctions against non-myopic bidders, and show approximate group incentive-compatibility in uniform-price auctions.

## 1 Introduction

In auction theory, it is well-known [32] that, when all buyers have values that are independently and identically drawn from a regular distribution $F$, the revenue-maximizing auction is simply the second price auction with anonymous reserve price $p^* = \arg\max\{p(1 - F(p)\}$: if the highest bid is at least $p^*$, then the highest bidder wins the item and pays the maximum between the second highest bid and $p^*$. The computation of $p^*$ requires the exact knowledge of the underlying value distribution, which is unrealistic because the value distribution is often unavailable in practice. Many works (e.g., [13, 18, 25]) on sample complexity in auctions have studied how to obtain a near-optimal reserve price based on samples from the distribution $F$ instead of knowing the exact $F$. One of the most important (and most fundamental) price learning algorithms in those works is the *Empirical Revenue Maximization* (ERM) algorithm, which simply outputs the reserve price that is optimal on the uniform distribution over samples (plus some regularization to prevent overfitting).

**Definition 1.1** (*c*-Guarded Empirical Revenue Maximization, ERM$^c$). *Draw $N$ samples from a distribution $F$ and sort them non-increasingly, denoted by $v_1 \geq v_2 \geq \cdots \geq v_N$. Given some regularization parameter $0 \leq c < 1$, choose:*

$$i^* = \arg\max_{cN < i \leq N}\{i \cdot v_i\}, \qquad \text{define } \mathrm{ERM}^c(v_1, \ldots, v_N) = v_{i^*}.$$

*Assume that the smaller sample (with the larger index) is chosen in case of ties.*

ERM$^c$ was first proposed by Dhangwatnotai et al. [18] and then extensively studied by Huang et al. [25]. They show that the reserve price outputted by ERM$^c$ is asymptotically optimal on the underlying distribution $F$ as the number of samples $N$ increases if $F$ is bounded or has monotone hazard rate, with an appropriate choice of $c$. Other papers [5, 27] have continued to study ERM$^c$.

However, when ERM is put into practice, it is unclear how the samples can be obtained since many times there is no impartial sampling source. A natural solution is *endogenous sampling*. For example, in repeated second price auctions, the auctioneer can use the bids in previous auctions as samples and run ERM to set a reserve price at each round. But this solution has a challenge of strategic issue: since bidders can affect the determination of future reserve prices, they might have an incentive to underbid in order to increase utility in future auctions.

Another example of endogenous sampling is the uniform-price auction. In a uniform-price auction the auctioneer sells $N$ copies of a good at some price $p$ to $N$ bidders with i.i.d. values $\boldsymbol{v}$ from $F$ who submit bids $\boldsymbol{b}$. Bidders who bid at least $p$ obtain one copy and pay $p$. The auctioneer can set the price to be $p = \mathrm{ERM}^0(\boldsymbol{v})$ to maximize revenue if bids are equal to values. But Goldberg et al. [20] show that this auction is not incentive-compatible as bidders can lower the price by strategic bidding. Therefore, the main question we consider in this paper is: *To what extent the strategic behavior of agents will affect ERM with endogenous sampling?*

To formally answer the question, we adopt a notion called "incentive-awareness measure" originally proposed by Lavi et al. [28] under bitcoin's fee market context, which measures the reduction of a price learning function $P$ due to a change of at most $m$ samples out of the $N$ input samples.

**Definition 1.2** (Incentive-awareness measures). *Let $P : \mathbb{R}_+^N \to \mathbb{R}_+$ be a function (e.g., ERM$^c$) that maps $N$ samples to a reserve price. Draw $N$ i.i.d. values $v_1, \ldots, v_N$ from a distribution $F$. Let $I \subseteq \{1, \ldots, N\}$ be an index set of size $|I| = m$, and $v_I = \{v_i \mid i \in I\}$, $v_{-I} = \{v_j \mid j \notin I\}$. A bidder can change $v_I$ to any $m$ non-negative bids $b_I$, hence change the price from $P(v_I, v_{-I})$ to $P(b_I, v_{-I})$. Define the* incentive-awareness measure*:*

$$\delta_I(v_I, v_{-I}) = 1 - \frac{\inf_{b_I \in \mathbb{R}_+^m} P(b_I, v_{-I})}{P(v_I, v_{-I})},$$

*and* worst-case incentive-awareness measures*:*

$$\delta_m^{\mathrm{worst}}(v_{-I}) = \sup_{v_I \in \mathbb{R}_+^m} [\delta_I(v_I, v_{-I})], \qquad \Delta_{N,m}^{\mathrm{worst}} = \mathbb{E}_{v_{-I} \sim F}[\delta_m^{\mathrm{worst}}(v_{-I})].$$

If the reserve price $P(b_I, v_{-I})$ can be reduced by a lot due to some choices of $b_I$, then the incentive-awareness measure is large. Since the reduction of reserve price usually increases bidders' utility, a smaller incentive-awareness measure implies that a bidder benefits less from strategic bidding, hence the name "incentive-awareness measure".[1]

In this paper, we focus on the measure $\Delta_{N,m}^{\mathrm{worst}}$. Lavi et al. [28] defined incentive-awareness measures only for $m = 1$ and showed that for any distribution $F$ with a finite support size, $\Delta_{N,1}^{\mathrm{worst}} \to 0$ as $N \to \infty$. Later, Yao [33] showed that $\Delta_{N,1}^{\mathrm{worst}} \to 0$ for any continuous distribution with support included in $[1, D]$. They did not provide specific convergent rates of $\Delta_{N,1}^{\mathrm{worst}}$. We generalize their definition to allow $m \geq 1$, which is crucial in our two applications to be discussed. Our main theoretical contribution is to provide upper bounds on $\Delta_{N,m}^{\mathrm{worst}}$ for two types of value distributions $F$: the class of Monotone Hazard Rate (MHR) distributions where $\frac{f(v)}{1 - F(v)}$ is non-decreasing over the support of the distribution (we use $F$ to denote the CDF and $f$ for PDF) and the class of bounded

distributions which consists of all (continuous and discontinuous) distributions with support included in $[1, D]$. MHR distribution can be unbounded so we are the first to consider incentive-awareness measures for unbounded distributions.

**Theorem 1.3** (Main). *Let $P = \mathrm{ERM}^c$. The worst-case incentive-awareness measure is bounded by*

- *for MHR $F$, $\Delta_{N,m}^{\mathrm{worst}} = O\left(m \frac{\log^3 N}{\sqrt{N}}\right)$, if $m = o(\sqrt{N})$ and $\frac{m}{N} \le c \le \frac{1}{4e}$.*[2]

- *for bounded $F$, $\Delta_{N,m}^{\mathrm{worst}} = O\left(D^{8/3} m^{2/3} \frac{\log^2 N}{N^{1/3}}\right)$, if $m = o(\sqrt{N})$ and $\frac{m}{N} \le c \le \frac{1}{2D}$.*

*The constants in the two big $O$'s are independent of $F$ and $c$.*

This theorem implies that as long as the fraction of samples controlled by a bidder is relatively small, the strategic behavior of each bidder has little impact on ERM provided that other bidders are truthful. Meanwhile, if more than one bidder bid non-truthfully, no bidder can benefit a lot from lying as long as the total number of bids from all non-truthful bidders does not exceed $m$. We will discuss intuitions and difficulties of the proof later in this section and give an overview of the proof in Section 4.

**Repeated auctions against non-myopic bidders.** Besides theoretical analysis, we apply the incentive-awareness measure to real-world scenarios to demonstrate the effect of strategic bidding on ERM. The main application we consider is repeated auctions where bidders participate in multiple auctions and have incentives to bid strategically to affect the auctions the seller will use in the future (Section 2). We consider a two-phase learning algorithm: the seller first runs second price auctions with no reserve for some time to collect samples, and then use these samples to set reserve prices by ERM in the second phase. The upper bound on the incentive-awareness measure of ERM implies that this algorithm is approximately incentive-compatible.

Kanoria and Nazerzadeh [27], Liu et al. [29], and Abernethy et al. [1] consider repeated auctions scenarios similar to ours. Kanoria and Nazerzadeh [27] set *personalized* reserve prices by ERM in repeated second price auctions, so at least two bidders are needed in each auction and they will face different reserve prices. We use *anonymous* reserve price so we allow only one bidder to participate in the auctions and when there are more than one bidder they face the same price, thus preventing discrimination. Liu et al. [29] and Abernethy et al. [1] design approximately incentive-compatible algorithms using differential privacy techniques rather than pure ERM. Comparing with them, our two-phase ERM algorithm is more practical as it is much simpler, and their algorithms rely on the boundedness of value distributions while we allow unbounded distributions. Moreover, their results require a large number of auctions while ours need a large number of samples in the first phase which can be obtained by either few bids in many auctions, many bids in few auctions, or combined.

**Uniform-price auctions and incentive-compatibility in the large.** Another scenario to which we apply the incentive-awareness measure of ERM is uniform-price auctions (Section 3). Azevedo and Budish [4] show that, uniform-price auctions are *incentive-compatible in the large* in the sense that truthful bidding is an approximate equilibrium when there are many bidders in the auction. In fact, incentive-compatibility in the large is the intuition of Theorem 1.3: when $N$ is large, no bidders can influence the learned price by much. The proof in [4] directly makes use of this intuition, showing that the bid of one bidder can affect the empirical distribution consisting of the $N$ bids only by a little. However, their argument, which crucially relies on the assumption that bidders' value distribution has a *finite* support and bids must be chosen *from this finite support* as well, fails when the value distribution is *continuous* or *bids can be any real numbers*, as what we allow. We instead, appeal to some specific properties of ERM to show that it is incentive-compatible in the large.

**Additional related works.** Previous works on ERM mainly focus on its sample complexity, started by Cole and Roughgarden [13]. While ERM is suitable for the case of i.i.d. values [25], the literature on sample complexity has expanded to more general cases of non-i.i.d. values and multi-dimensional values, e.g. [31, 17, 22, 23], or considering non-truthful auctions, e.g. [24]. Babaioff et al. [5] study the performance of ERM with just two samples. While this literature assumes that samples are exogenous, our main contribution is to consider endogenous samples that are collected from bidders who are affected by the outcome of the learning algorithm and hence have incentive to manipulate the samples.

Some works study repeated auctions but with myopic (non-strategic) bidders [7, 30, 12, 10]. Existing works about non-myopic (strategic) bidders make different assumptions on bidders' behavior: bidders can play best responses [2, 3, 15, 21], use no-regret learning algorithms [8], or are assumed to bid truthfully if the mechanism is $\epsilon$-approximately incentive-compatible [1]. Our work makes the last assumption. This assumption is reasonable because in our models, there is a large pool of bidders, making it difficult for bidders to find best responses. To find a best response, bidders have to collect a lot of information about other bidders and do a large amount of computation; when the cost of searching for a better response exceeds $\epsilon$, the bidder is better-off bidding truthfully.

Existing works on repeated auctions against strategic bidders have proposed various types of algorithms to maximize revenue (e.g., [2, 3, 15, 21, 29, 1]). We complement that line of works by showing that ERM, the most fundamental algorithm we believe, has good performance as well.

Other works about incentive-aware learning (e.g., [16, 26, 6, 19]) consider settings different from ours. For example, [16] and [26] study repeated auctions where buyers' values are drawn from some distribution at first and then fixed throughout. The seller knows the distribution and tries to learn the exact values, which is different from our assumption that the distribution is unknown to the seller.

## 2 Main Application: A Two-Phase Model

Here we consider a *two-phase model* as a real-world scenario where strategic bidding affects ERM: the seller first runs second price auctions with no reserve for some time to collect samples, and then use these samples to set reserve prices by ERM in the second phase. This model can be regarded as an "exploration and exploitation" learning algorithm in repeated auctions, and we will show that this algorithm can be approximately incentive-compatible and revenue-optimal.

### 2.1 The Model

A two-phase model is denoted by $\mathrm{TP}(\mathcal{M}, P; F, \boldsymbol{T}, \boldsymbol{m}, \boldsymbol{K}, S)$, where $\mathcal{M}$ is a truthful, prior-independent mechanism, $P$ is a price learning function, and $\boldsymbol{T} = (T_1, T_2)$ are the numbers of auctions in the two phases. We do not assume that every bidder participates in all auctions. Instead, we assume that each bidder participates in no more than $m_1, m_2$ auctions in the two phases, respectively; $\boldsymbol{m} = (m_1, m_2)$. We use $\boldsymbol{K} = (K_1, K_2)$ to denote the numbers of bidders in the auctions of the two phases.[3] $S = S_1 \times \cdots \times S_n$ is the strategy space, where $s_i \in S_i : \mathbb{R}_+^{m_{i,1}+m_{i,2}} \to \mathbb{R}_+^{m_{i,1}}$ is a strategy of bidder $i = 1, \ldots, n$. The procedure is:

- At the beginning, each bidder realizes $\boldsymbol{v}_i = (v_{i,1}, \ldots, v_{i,m_{i,1}+m_{i,2}})$ i.i.d. drawn from $F$. Let $\boldsymbol{v}_{-i}$ denote the values of bidders other than $i$. Bidder $i$ knows $\boldsymbol{v}_i$ but does not know $\boldsymbol{v}_{-i}$.

- In the exploration phase, $T_1$ auctions are run using $\mathcal{M}$ and bidders bid according to some strategy $s \in S$. Each auction has $K_1$ bidders and each bidder $i$ participants in $m_{i,1} \leq m_1$ auctions. The auctioneer observes a random vector of bids $\boldsymbol{b} = (b_1, \ldots, b_{T_1 K_1})$ with the following distribution: let $I$ be an index set corresponding to bidder $i$, with size $|I| = m_{i,1}$; then $\boldsymbol{b} = (b_I, b_{-I})$, where $b_I \sim s_i(\boldsymbol{v}_i)$, and $b_{-I} \sim s_{-i}(\boldsymbol{v}_{-i})$.

- In the exploitation phase, $T_2$ second price auctions ($K_2 \geq 2$) or posted price auctions ($K_2 = 1$) are run, with reserve price $p = P(\boldsymbol{b})$. Each auction has $K_2$ bidders and each bidder $i$ participants in $m_{i,2} \leq m_2$ auctions. The auctions in this phase are truthful because $p$ has been fixed.

**Utilities.** Denote the utility of bidder $i$ as:

$$U_i^{\mathrm{TP}}(\boldsymbol{v}_i, b_I, b_{-I}) = U_i^{\mathcal{M}}(\boldsymbol{v}_i, b_I, b_{-I}) + \sum_{t=m_{i,1}+1}^{m_{i,1}+m_{i,2}} u^{K_2}(v_{i,t}, P(b_I, b_{-I})), \tag{1}$$

where $U_i^{\mathcal{M}}(v_I, b_I, b_{-I})$ is the utility of bidder $i$ in the first phase, and $u^{K_2}(v, p)$ is the interim utility of a bidder with value $v$ in a second price auction with reserve price $p$ among $K_2 \geq 1$ bidders:

$$u^{K_2}(v, p) = \mathbb{E}_{X_2, \ldots, X_{K_2} \sim F}\left[\left(v - \max\{p, X_2, \ldots, X_{K_2}\}\right) \cdot \mathbb{I}\left[v > \max\{p, X_2, \ldots, X_{K_2}\}\right]\right]. \quad (2)$$

The *interim utility* of bidder $i$ in the two-phase model is $\mathbb{E}_{\boldsymbol{v}_{-i} \sim F}\left[U_i^{\mathrm{TP}}(\boldsymbol{v}_i, b_I, b_{-I})\right]$.

**Approximate Bayesian incentive-compatibility.** We use the additive version of the solution concept of an $\epsilon$-Bayesian-Nash equilibrium ($\epsilon$-BNE), i.e., in such a solution concept, no player can improve her utility by more than $\epsilon$ by deviating from the equilibrium strategy. We say a mechanism is $\epsilon$-approximately Bayesian incentive-compatible ($\epsilon$-BIC) if truthful bidding is an $\epsilon$-BNE, i.e., if for any $\boldsymbol{v}_i \in \mathbb{R}_+^{m_{i,1}+m_{i,2}}$, any $b_I \in \mathbb{R}_+^{m_{i,1}}$,

$$\mathbb{E}_{\boldsymbol{v}_{-i} \sim F}\left[U_i^{\mathrm{TP}}(\boldsymbol{v}_i, b_I, v_{-I}) - U_i^{\mathrm{TP}}(\boldsymbol{v}_i, v_I, v_{-I})\right] \leq \epsilon,$$

If a mechanism is $\epsilon$-BIC and $\lim_{n \to \infty} \epsilon = 0$, then each bidder knows that if all other bidders are bidding truthfully then the gain from any deviation from truthful bidding is negligible for her. To realize that strategic bidding cannot benefit them much, bidders do not need to know the underlying distribution, but only the fact that the mechanism is $\epsilon$-BIC. We are therefore going to assume in this paper that, in such a case, all bidders will bid truthfully.[4]

**Approximate revenue optimality.** We say that a mechanism is $(1 - \epsilon)$ revenue optimal, for some $0 < \epsilon < 1$, if its expected revenue is at least $(1 - \epsilon)$ times the expected revenue of Myerson auction.[5] Huang et al. [25] show that a one-bidder auction with posted price set by $\mathrm{ERM}^c$ (for an appropriate $c$) and with $N$ samples from the value distribution is $(1 - \epsilon)$ revenue optimal with $\epsilon = O((N^{-1} \log N)^{2/3})$ for MHR distributions and $\epsilon = O(\sqrt{DN^{-1} \log N})$ for bounded distributions.

**The i.i.d. assumption.** Our assumption of i.i.d. values is reasonable because in our scenario there is a large population of bidders, and we can regard this population as a distribution and each bidder as a sample from it. So from each bidder's perspective, the values of other bidders are i.i.d. samples from this distribution. Then the $\epsilon$-BIC notion implies that when others bid truthfully, it is approximately optimal for bidder $i$ to bid truthfully no matter what her value is.

## 2.2 Incentive-Compatibility and Revenue Optimality

Now we show that, as the incentive-awareness measure of $P$ becomes lower, the price learning function becomes more incentive-aware in the sense that bidders gain less from non-truthful bidding:

**Theorem 2.1.** *In* $\mathrm{TP}(\mathcal{M}, P; F, \boldsymbol{T}, \boldsymbol{m}, \boldsymbol{K}, S)$, *truthful bidding is an* $\epsilon$-BNE, *where,*

- *for any $P$ and any bounded $F$, $\epsilon = m_2 D \Delta_{T_1 K_1, m_1}^{\mathrm{worst}}$, and*

- *for any MHR $F$, if we fix $P = \mathrm{ERM}^c$ with $\frac{m_1}{T_1 K_1} \leq c \leq \frac{1}{4e}$ and $m_1 = o(\sqrt{T_1 K_1})$, then*
$\epsilon = O\left(m_2 v^* \Delta_{T_1 K_1, m_1}^{\mathrm{worst}}\right) + O\left(\frac{m_2 v^*}{\sqrt{T_1 K_1}}\right)$, *where* $v^* = \arg\max_v\{v[1 - F(v)]\}$.

*The constants in big O's are independent of $F$ and $c$.*

Combined with Theorem 1.3 which upper bounds the incentive-awareness measure, we can obtain explicit bounds on truthfulness of the two-phase model by plugging in $N = T_1 K_1$ and $m = m_1$. Precisely, for any bounded $F$, $\epsilon = O\left(D^{11/3} m_2 m_1^{2/3} \frac{\log^2(T_1 K_1)}{(T_1 K_1)^{1/3}}\right)$ if $m_1 = o(\sqrt{T_1 K_1})$ and $\frac{m_1}{T_1 K_1} \leq c \leq \frac{1}{2D}$. For any MHR $F$, $\epsilon = O\left(v^* m_2 m_1 \frac{\log^3(T_1 K_1)}{\sqrt{T_1 K_1}}\right)$ if $m_1 = o(\sqrt{T_1 K_1})$ and $\frac{m_1}{T_1 K_1} \leq c \leq \frac{1}{4e}$.

Thus, for both cases, keeping all the parameters except $T_1$ constant (in particular $m_1$ and $m_2$ are constants) implies that $\epsilon \to 0$ at a rate which is not slower than $O((T_1)^{-1/3} \log^3 T_1)$ as $T_1 \to +\infty$.

To simultaneously obtain both approximate BIC and approximate revenue optimality, a certain balance between the number of auctions in the two phases must be maintained. Few auctions in the first phase and many auctions in the second phase hurt truthfulness as the loss from non-truthful bidding (i.e., losing in the first phase) is small compared to the gain from manipulating the reserve price in the second phase. Many auctions in the first phase are problematic as we do not have any good revenue guarantees in the first phase (since we allow any truthful $\mathcal{M}$). Thus, a certain balance must be maintained, as expressed formally in the following theorem:

**Theorem 2.2.** *Assume that $K_2 \geq K_1 \geq 1$. Let $\overline{m} = m_1 + m_2$. In $\mathrm{TP}(\mathcal{M}, \mathrm{ERM}^c; F, \boldsymbol{T}, \boldsymbol{m}, \boldsymbol{K}, S)$, to simultaneously obtain $\epsilon_1$-BIC and $(1 - \epsilon_2)$ revenue optimality (assuming truthful bidding), it suffices to set the parameters as follows:*

- *If $F$ is an MHR distribution, $\frac{\overline{m}}{T_1 K_1} \leq c \leq \frac{1}{4e}$, $\overline{m} = o(\sqrt{T_1 K_1})$, then*

$$\epsilon_1 = O\left(v^* \overline{m}^2 \frac{\log^3(T_1 K_1)}{\sqrt{T_1 K_1}}\right), \text{ and } \epsilon_2 = O\left(\frac{T_1}{T} + \left[\frac{\log(T_1 K_1)}{T_1 K_1}\right]^{\frac{2}{3}}\right).$$

- *If $F$ is bounded and regular, $\frac{\overline{m}}{T_1 K_1} \leq c \leq \frac{1}{2D}$, $\overline{m} = o(\sqrt{T_1 K})$, then*

$$\epsilon_1 = O\left(D^{11/3} \overline{m}^{5/3} \frac{\log^2(T_1 K_1)}{(T_1 K_1)^{1/3}}\right), \text{ and } \epsilon_2 = O\left(\frac{T_1}{T} + \sqrt{\frac{D \cdot \log(T_1 K_1)}{T_1 K_1}}\right).[6]$$

The proof is given in Appendix C. This theorem makes explicit the fact that in order to simultaneously obtain approximate BIC and approximate revenue optimality, $T_1$ cannot be too small nor too large: for approximate revenue optimality we need $T_1 \ll T$ and for approximate BIC we need, e.g., $T_1 \gg (v^*)^2 \overline{m}^4 \log^6(v^* \overline{m})/K_1$ for MHR distributions, and $T_1 \gg D^{11} \overline{m}^5 \log^6(D\overline{m})/K_1$ for bounded distributions. When setting the parameters in this way, both $\epsilon_1$ and $\epsilon_2$ go to $0$ as $T \to \infty$.

## 2.3 Multi-Unit Extension

The auction in the exploitation phase can be generalized to a multi-unit Vickrey auction with anonymous reserve, where $k \geq 1$ identical units of an item are sold to $K_2$ unit-demand bidders and among those bidders whose bids are greater than the reserve price $p$, at most $k$ bidders with largest bids win the units and pay the maximum between $p$ and the $(k + 1)$-th largest bid. The multi-unit Vickrey auction with an anonymous reserve price is revenue-optimal when the value distribution is regular, and the optimal reserve price does not depend on $k$ or $K_2$ according to Myerson [32]. Thus the optimal reserve price can also be found by $\mathrm{ERM}^c$. All our results concerning truthfulness, e.g., Theorem 2.1, still hold for the multi-unit extension with any $k \geq 1$. Moreover, Theorem 2.2 also holds because we have already considered the multi-unit extension in its proof in Appendix C.

## 2.4 Two-Phase ERM Algorithm in Repeated Auctions

The two-phase model with ERM as the price learning function can be seen as a learning algorithm in the following setting of repeated auctions against strategic bidders: there are $T$ rounds of auctions, there are $K \geq 1$ bidders in each auction, and each bidder participates in at most $\overline{m}$ auctions. The algorithm, which we call "two-phase ERM", works as follows: in the first $T_1$ rounds, run any truthful, prior-independent auction $\mathcal{M}$ (e.g., the second price auction with no reserve); in the later $T_2 = T - T_1$ rounds, run second price auction with reserve $p = \mathrm{ERM}^c(b_1, \ldots, b_{T_1 K})$ where $b_1, \ldots, b_{T_1 K}$ are the bids from the first $T_1$ auctions. $T_1$ and $c$ are adjustable parameters.

In repeated games, one may also consider $\epsilon$-perfect Bayesian equilibrium ($\epsilon$-PBE) as the solution concept besides $\epsilon$-BNE. A formal definition is given in Appendix C.4 but roughly speaking, $\epsilon$-

PBE requires that the bidding of each bidder at each round of auction $\epsilon$-approximately maximizes the total expected utility in all future rounds, conditioning on any observed history of allocations and payments. Note that the history may leak some information about the historical bids of other buyers and these bids will affect the seller's choice of mechanisms in future rounds. Similar to the $\epsilon$-BNE notion, we can show that the two-phase ERM algorithm satisfies: (1) truthful bidding is an $O\left(\log^2(T_1 K)\sqrt[3]{\frac{D^{11}K\overline{m}}{T_1}}\right)$-PBE; (2) $\left(1 - O\left(\frac{T_1}{T} + \sqrt{\frac{D\log(T_1 K)}{T_1 K}}\right)\right)$ revenue optimality, for bounded distributions; and similar results for MHR distributions. Choosing $T_1 = \tilde{O}(T^{\frac{2}{3}})$, which maximizes the revenue, we obtain $\tilde{O}(T^{-\frac{2}{9}})$-truthfulness and $(1 - \tilde{O}(T^{-\frac{1}{3}}))$ revenue optimality, where we assume $D$, $\overline{m}$, and $K$ to be constant.[7]

Under the same setting, Liu et al. [29] and Abernethy et al. [1] design approximately truthful and revenue optimal learning algorithms using differential privacy techniques. We can compare two-phase ERM and their algorithms. Firstly, they make a similar assumption as ours that $\overline{m} = o(\sqrt{T/K})$, in order to obtain approximate truthfulness and revenue optimality at the same time. In terms of truthfulness notion, Liu et al. [29] assume that bidders play an exact PBE instead of $\epsilon$-PBE, so their quantitative result is incomparable with ours. Their notion of exact PBE is too strong to be practical because bidders need to collect a lot of information about other bidders and do a large amount of computation to find the exact equilibrium, while our notion guarantees bidders of approximately optimal utility as long as they bid truthfully. Although our truthfulness bound is worse than the bound of [1], which is $\tilde{O}(\frac{1}{\sqrt{T}})$, we emphasize that their $\epsilon$-truthfulness notion is *weaker* than ours: in their definition, each bidder cannot gain more than $\epsilon$ in current and future rounds if she deviates from truthful bidding *only in the current round*, given any fixed future strategy. But in our definition, each bidder cannot gain more than $\epsilon$ if she deviates *in current and all future rounds*. Our algorithm is easier to implement and more time-efficient than theirs, and works for unbounded distribution while theirs only support bounded distributions because they need to discretize the value space.

One may wonder whether we can do exploration and exploitation at the same time to improve the performance of ERM, compared to doing them in a two-phase manner. We argue that this is suitable for the BIC truthfulness notion but not for the PBE notion. For the BIC notion, we can change the two-phase algorithm which fixes the price in the second phase to continuously update the reserve price in the second phase; this improves revenue without affecting BIC truthfulness. But for the PBE notion, we cannot do exploration and exploitation at the same time. Consider this example: Suppose every auction has only one bidder. In the first round bidder A submits bid 1 and 1 is set to be the reserve price for the second round. In the second round, bidder B submits 0.9 and loses, so she knows that the first round bid is greater than 0.9. Suppose bidder B will join the next round, then she can strategize her bid based on the information "the first round bid is greater than 0.9". So the belief of bidder B on bidder A's value distribution is no longer $F$, and our argument, which relies on the assumption of i.i.d. value distribution, fails. Instead, doing exploration and exploitation separately preserves approximate incentive-compatibity for the PBE notion (see Appendix C.4).

## 3   A Second Application: Uniform-Price Auctions

The notion of an incentive-awareness measure (recall Definition 1.2) has implications regarding the classic uniform-price auction model, which we believe are of independent interest. In a static uniform-price auction we have $N$ copies of a good and $N$ unit-demand bidders with i.i.d. values $\boldsymbol{v}$ from $F$ that submit bids $\boldsymbol{b}$. The auctioneer then sets a price $p = P(\boldsymbol{b})$. Each bidder $i$ whose value $v_i$ is above or equal to $p$ receives a copy of the good and pays $p$, obtaining a utility of $v_i - p$; otherwise the utility is zero. Azevedo and Budish [4] show that this auction is "incentive-compatible in the large" which means that truthfulness is an $\epsilon$-BNE and $\epsilon$ goes to zero as $N$ goes to infinity. They assume bidders' value distribution has a finite support and their bids must be chosen from this finite support as well. They mention that allowing continuous supports and arbitrary bids is challenging.

In this context, taking $P = \mathrm{ERM}^c$ is very natural when the auctioneer aims to maximize revenue. Indeed, Goldberg et al. [20] suggest to use the uniform-price auction with $P = \mathrm{ERM}^c$, where $c = \frac{1}{N}$, as a revenue benchmark for evaluating other truthful auctions they design.

When the price function is $P = \text{ERM}^{c=\frac{1}{N}}$, our analysis of the incentive-awareness measure generalizes the result of [4] to bounded and to MHR distributions. Moreover, we generalize their result to the case where coalitions of at most $m$ bidders can coordinate bids and jointly deviate from truthfulness.

**Theorem 3.1.** *In the uniform-price auction, suppose that any $m$ bidders can jointly deviate from truthful bidding, then no bidder can obtain $\epsilon$ more utility (we call this $(m, \epsilon)$-group BIC), where,*

- *for any $P$ and any bounded $F$, $\epsilon = D\Delta_{N,m}^{\text{worst}}$, and*

- *for any MHR distribution $F$, if we fix $P = \text{ERM}^c$ with $\frac{m}{N} \leq c \leq \frac{1}{4e}$ and $m = o(\sqrt{N})$, then*
  $\epsilon = O\left(v^* \Delta_{N,m}^{\text{worst}}\right) + O\left(\frac{v^*}{\sqrt{N}}\right)$, *where $v^* = \arg\max_v\{v[1 - F(v)]\}$.*

*The constants in big O's are independent of $F$ and $c$.*

*Proof of Theorem 3.1 for bounded distributions.* Denote a coalition of $m$ bidders by an index set $I \subseteq \{1, \ldots, N\}$, and the true values of all bidders by $(v_I, v_{-I})$. When other bidders bid $v_{-I}$ truthfully, and the coalition bids $b_I$ instead of $v_I$, the reduction of price is at most

$$P(v_I, v_{-I}) - P(b_I, v_{-I}) \leq P(v_I, v_{-I})\delta_I(v_I, v_{-I}) \leq P(v_I, v_{-I})\delta_m^{\text{worst}}(v_{-I}) \leq D\delta_m^{\text{worst}}(v_{-I}),$$

by Definition 1.2 and by the fact that all values are upper-bounded by $D$. Then for each bidder $i \in I$, the increase of her utility by such a joint deviation is no larger than the reduction of price, i.e.

$$\mathbb{E}_{v_{-I}}\left[u_i(v_I, P(b_I, v_{-I})) - u_i(v_I, P(v_I, v_{-I}))\right] \leq \mathbb{E}_{v_{-I}}\left[P(v_I, v_{-I}) - P(b_I, v_{-I})\right]$$
$$\leq D\mathbb{E}_{v_{-I}}\left[\delta_m^{\text{worst}}(v_{-I})\right] = D\Delta_{N,m}^{\text{worst}}.$$

$\square$

The proof of this theorem for MHR distributions is similar to the proof of Theorem 2.1, thus omitted.

Combining with Theorem 1.3, we conclude that the uniform-price auction with $P = \text{ERM}^c$ (for the $c$'s mentioned there) is $(m, \epsilon)$-group BIC with $\epsilon$ converging to zero at a rate not slower than $O(m^{2/3}\frac{\log^2 N}{N^{1/3}})$ for bounded distributions and $O(m\frac{\log^3 N}{\sqrt{N}})$ for MHR distributions (constants in these big O's depend on distributions).

Theorem 3.1 also generalizes the result in [28] which is only for bounded distributions and $m = 1$.

## 4 More Discussions on Incentive-awareness Measures

### 4.1 Overview of the Proof for Upper Bounds on $\Delta_{N,m}^{\text{worst}}$

Here we provide an overview of the proof of Theorem 1.3. Details are in Appendix B.

Firstly, we show an important property of $\text{ERM}^c$: suppose $c \geq \frac{m}{N}$, for any $m$ values $v_I$, any $N - m$ values $v_{-I}$, and any $m$ values $\overline{v}_I$ that are greater than or equal to the maximum value in $v_{-I}$, we have $\text{ERM}^c(\overline{v}_I, v_{-I}) \geq \text{ERM}^c(v_I, v_{-I})$. As a consequence, $\delta_m^{\text{worst}}(v_{-I}) = \delta_I(\overline{v}_I, v_{-I})$.

Based on this property, we transfer the expectation in the incentive-awareness measure in the following way:

$$\Delta_{N,m}^{\text{worst}} = \mathbb{E}[\delta_m^{\text{worst}}(v_{-I})] = \mathbb{E}[\delta_I(\overline{v}_I, v_{-I})] = \int_0^1 \Pr[\delta_I(\overline{v}_I, v_{-I}) > \eta]d\eta$$

$$\leq \int_0^1 \left(\Pr[\delta_I(\overline{v}_I, v_{-I}) > \eta \mid \overline{\text{E}}]\Pr[\overline{\text{E}}] + \Pr[\text{E}]\right)d\eta = \int_0^1 \Pr[\delta_I(\overline{v}_I, v_{-I}) > \eta \wedge \overline{\text{E}}]d\eta + \Pr[\text{E}],$$

where E denotes the event that the index $k^* = \arg\max_{i>cN}\{iv_i\}$ (which is the index selected by $\text{ERM}^c$) satisfies $k^* \leq dN$, $\overline{\text{E}}$ denotes the complement of E, and the probability in $\Pr[\text{E}]$ is taken over the random draw of $N - m$ i.i.d. samples from $F$, with other $m$ samples fixed to be the upper bound (can be $+\infty$) of the distribution. For any value distribution, we prove that the first part

$$\int_0^1 \Pr[\delta_I(\overline{v}_I, v_{-I}) > \eta \wedge \overline{\text{E}}]d\eta \leq O\left(\sqrt[3]{\frac{m^2}{d^8}}\frac{\log^2 N}{\sqrt[3]{N}}\right),$$

with some constructions of auxiliary events and involved probabilistic argument. And we further tighten this bound to $O\left(\frac{m}{d^{7/2}}\frac{\log^3 N}{\sqrt{N}}\right)$ for MHR distribution by leveraging its properties.

The final part of the proof is to bound $\Pr[E]$. For bounded distribution, we choose $d = 1/D$. Since the support of the distribution is bounded by $[1, D]$, $N \cdot v_N \geq N$, while for any $k \leq dN$, $kv_k \leq (\frac{1}{D}N)D = N \leq Nv_N$. $\text{ERM}^c$ therefore never chooses an index $k \leq dN$ (recall that in case of a tie, $\text{ERM}^c$ picks the larger index). This implies $\Pr[E] = \Pr[k^* \leq dN] = 0$ for bounded distribution. For MHR distribution, we choose $d = \frac{1}{2e}$ and show $\Pr[E] = O\left(\frac{1}{N}\right)$. As the corresponding proof is quite complicated, we omit it here.

## 4.2 Lower Bounds on $\triangle_{N,m}^{\text{worst}}$ and on the Approximate BIC Parameter, $\epsilon_1$

Theorem 1.3 gives an upper bound on $\triangle_{N,m}^{\text{worst}}$ for bounded and MHR distributions and for a specific range of $c$'s. Here we briefly discuss the lower bound, with details given in Appendix F.

Lavi et al. [28] show that for the two-point distribution $v = 1$ and $v = 2$, each w.p. 0.5, $\triangle_{N,1}^{\text{worst}} = \Omega(N^{-1/2})$, when $c = 1/N$. We adopt their analysis to provide a similar lower bound for $[1, D]$-bounded distributions and the corresponding range of $c$'s. Let $F$ be a two-point distribution where for $X \sim F$, $\Pr[X = 1] = 1 - 1/D$ and $\Pr[X = D] = 1/D$.

**Theorem 4.1.** *For the above $F$, for any $c \in [\frac{m}{N}, \frac{1}{2D}]$, $\text{ERM}^c$ gives $\triangle_{N,m}^{\text{worst}} = \Omega(\frac{1}{\sqrt{N}})$ where the constant in $\Omega$ depends on $D$.*

Note that $\triangle_{N,m}^{\text{worst}}$ only upper bounds the $\epsilon_1$-BIC parameter $\epsilon_1$ in the two-phase model: a lower bound on $\triangle_{N,m}^{\text{worst}}$ does not immediately implies a lower bound on $\epsilon_1$. Still, a direct argument will show that the above distribution $F$ gives the same lower bound on $\epsilon_1$. For simplicity let $K_1 = K_2 = 2$ and suppose bidder $i$ participates in $m_1$ and $m_2$ auctions in the two phases, respectively. Let $N = T_1 K_1$ and assume $m_1 = o(\sqrt{N})$. Suppose the first-phase mechanism $\mathcal{M}$ is the second price auction with no reserve price. Then in the two-phase model with $\text{ERM}^c$, $\epsilon_1$ must be $\Omega(\frac{m_2}{\sqrt{N}})$ to guarantee $\epsilon_1$-BIC.

It remains open to prove a lower bound for MHR distributions, and to close the gap between our $O(N^{-1/3}\log^2 N)$ upper bound and the $\Omega(N^{-1/2})$ lower bound for bounded distributions.

## 4.3 Unbounded Regular Distributions

Theorem 2.2 shows that, in the two-phase model, approximate incentive-compatibility and revenue optimality can be obtained simultaneously for bounded (regular) distributions and for MHR distributions. A natural question would then be: what is the largest class of value distribution we can consider? Note that for non-regular distributions, Myerson [32] shows that revenue optimality cannot be guaranteed by anonymous reserve price, so ERM is not a correct choice. Thus we generalize our results to the class of regular distributions that are unbounded and not MHR. Here we provide a sketch, with details given in Appendix G.

Our results can be generalized to $\alpha$-strongly regular distributions with $\alpha > 0$. As defined in [13], a distribution $F$ with positive density function $f$ on its support $[A, B]$ where $0 \leq A \leq B \leq +\infty$ is $\alpha$-*strongly regular* if the virtual value function $\phi(x) = x - \frac{1-F(x)}{f(x)}$ satisfies $\phi(y) - \phi(x) \geq \alpha(y - x)$ whenever $y > x$ (or $\phi'(x) \geq \alpha$ if $\phi(x)$ is differentiable). As special cases, regular and MHR distributions are 0-strongly and 1-strongly regular distributions, respectively. For any $\alpha > 0$, we obtain bounds similar to MHR distributions on $\triangle_{N,m}^{\text{worst}}$ and on approximate incentive-compatibility in the two-phase model and the uniform-price auction. Specifically, if $F$ is $\alpha$-strongly regular then

$$\triangle_{N,m}^{\text{worst}} = O\left(m\frac{\log^3 N}{\sqrt{N}}\right), \text{ if } m = o(\sqrt{N}) \text{ and } \frac{m}{N} \leq \left(\frac{\log N}{N}\right)^{1/3} \leq c \leq \frac{\alpha^{1/(1-\alpha)}}{4}.$$

It remains an open problem for future research whether $\text{ERM}^c$ is incentive-compatible in the large for regular but not $\alpha$-strongly regular distributions for any $\alpha > 0$. For these distributions the choice of $c$ must be more sophisticated since it creates a clash between approximate incentive-compatibility and approximate revenue optimality. Intuitively, a large $c$ (for example, a constant) will hurt revenue optimality and a too small $c$ will hurt incentive-compatibility. In Appendix G, we provide examples and proofs to formally illustrate such a fact, and further discuss our conjecture that some intermediate $c$ can maintain the balance between incentive-compatibility and revenue optimality.

## Broader Impact

This work is mainly theoretical. It provides some intuitions and guidelines for potential practices, but does not have immediate societal consequences. A possible positive consequence is: the auction we consider uses an anonymous reserve price, while most of the related works on repeated auctions use unfair personalized prices. We do not see negative consequences.

## Acknowledgments and Disclosure of Funding

This work was supported by the NSFC-ISF joint research program (grant No. NSFC-ISF 61761146005), the ISF-NSFC joint research program (ISF grant No. 2560/17), and the Research Grant Council of Hong Kong (GRF Project No. 16215717 and 16243516). We would like to thank Changjun Wang for helpful discussion during the early stage of this work.

## Footnotes

[1]Lavi et al. [28] use the name "discount ratio" which we feel can be confused with the standard meaning of a discount ratio in repeated games.

[2]We use $a(n) = o(b(n))$ to denote $\lim_{n \to +\infty} \frac{a(n)}{b(n)} = 0$.

[3]It is well-known that when there are $n$ bidders with i.i.d. regular value distributions in one auction, a second price auction as a prior-independent mechanism is $(1 - 1/n)$-revenue optimal. But in our two-phase model, the numbers of bidders in each auction, $K_1$ and $K_2$, can be small, e.g., 1 or 2. With few bidders, prior-independent mechanisms do not have good revenue.

[4]One may consider transforming an $\epsilon$-BIC mechanism to an exact BIC mechanism using some current techniques (e.g., Cai et al. [11], Conitzer et al. [14]) in our model. However, current techniques need to know the value distribution, which is unavailable here. Even if we assume the distribution is available, those transformations result in an $O(n\sqrt{\epsilon})$ revenue loss; since $\epsilon$ is at least $\Omega(1/\sqrt{n})$ in our setting, the revenue loss does not converge to 0 as $n \to \infty$.

[5]One may take the optimal $\epsilon$-BIC auction rather than the exact BIC Myerson auction as the revenue benchmark. However, as shown by e.g., Lemma 1 of Brustle et al. [9], the revenue of the optimal $\epsilon$-BIC auction is at most $O(\epsilon)$ greater than that of Myerson auction; so all our revenue approximation results hold for this stronger benchmark except for an additive $O(\epsilon)$ term.

[6]The requirement that $F$ is regular in addition to being bounded comes from the fact that $\mathrm{ERM}^c$ approximates the optimal revenue in an auction with many bidders only for regular distributions. In fact, the sample complexity literature on $\mathrm{ERM}^c$ only studies the case of one bidder (which is, in our notation, $K_2 = 1$). In this case, i.e., if the second phase uses posted price auctions, we do not need the regularity assumption. To capture the case of general $K_2$, we make a technical observation that for regular distributions $(1 - \epsilon)$ revenue optimality for a single buyer implies $(1 - \epsilon)$ revenue optimality for many buyers (Lemma C.1). We do not know if this is true without the regularity assumption or if this observation – which may be of independent interest – was previously known.

[7]The $\tilde{O}$ notation omits polylogarithmic terms.

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
