[Supplementary Material]

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

 bidders [7, 28, 10, 9]. Existing works about non-myopic bidders focus on designing various learning algorithms to maximize revenue assuming bidders playing best responds [2, 3, 12, 18] or using no-regret learning algorithm [8]. We complement that line of works by showing that ERM, the most fundamental algorithm we believe, also has good performance in repeated auctions against strategic bidders.

Other works about incentive-aware learning (e.g., [13, 24, 6, 16]) consider settings different from ours. For example, [13] and [24] study repeated auctions where buyers' values are drawn from some distribution at first and then fixed throughout. The seller knows the distribution and tries to learn the exact values, which is different from our assumption that the distribution is unknown to the seller.

## 2 Main Application: A Two-Phase Model

Here we consider a *two-phase model* as a real-world scenario where strategic bidding affects ERM: the seller first runs second price auctions with no reserve for some time to collect samples, and then use these samples to set reserve prices by ERM in the second phase. This model can be regarded as an "exploration and exploitation" learning algorithm in repeated auctions, and we will show that this algorithm can be approximately incentive-compatible and revenue-optimal.

## 2.1 The Model

A two-phase model is denoted by $\text{TP}(\mathcal{M}, P; F, \boldsymbol{T}, \boldsymbol{m}, \boldsymbol{K}, S)$, where $\mathcal{M}$ is a truthful, prior-independent mechanism, $P$ is a price learning function, $\boldsymbol{T} = (T_1, T_2)$ are the numbers of auctions in the two phases, $\boldsymbol{m} = (m_1, m_2)$ are upper bounds on the number of auctions each bidder participates in, $\boldsymbol{K} = (K_1, K_2)$ are the number of bidders in auctions, $

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

# A    Useful Facts

 In this section we present some facts about $\mathrm{ERM}^c$ and incentive-awareness measures, some definitions about
 value distributions, and some useful lemmas that will be used throughout.

## A.1    Facts about $\mathrm{ERM}^c$ and Incentive-Awareness Measures

**Claim A.1.** *Let* $P = \mathrm{ERM}^c$*, where* $c \geq \frac{m}{N}$*. For any* $v_I \in \mathbb{R}_+^m$*,* $v_{-I} \in \mathbb{R}_+^{N-m}$*, let* $\overline{v}_I$ *denote* $m$ *values such that* $\min \overline{v}_I \geq \max v_{-I}$*. Then we have* $P(\overline{v}_I, v_{-I}) \geq P(v_I, v_{-I})$*.*

*Proof.* Let $\overline{v} := \max v_{-I}$ be the largest value in $v_{-I}$ and $\overline{v}_I$ be $m$ copies of $\overline{v}$. It suffices to show that $P(\overline{v}_I, v_{-I}) \geq P(v_I, v_{-I})$ since $P = \mathrm{ERM}^c$ ignores the largest $m$ samples, given $cN \geq m$. If $v_i \geq \overline{v}$ for each $i \in I$, then we have $P(\overline{v}_I, v_{-I}) = P(v_I, v_{-I})$ directly. If there exists some $i \in I$ such that $v_i < \overline{v}$, then we increase $v_i$ to $\overline{v}$ and show that for any such $i$ and $(v_i, v_{-i})$, $P(\overline{v}, v_{-i}) \geq P(v_i, v_{-i})$. Let $v' = P(v_i, v_{-i})$, then one can verify (assuming $\mathrm{ERM}^c$ picks the smaller value when there are ties) that (1) $P(v', v_{-i}) = P(v_i, v_{-i})$, and (2) $P(v', v_{-i}) \leq P(\overline{v}, v_{-i})$, implying $P(\overline{v}, v_{-i}) \geq P(v_i, v_{-i})$. $\square$

**Claim A.2.** *Let* $P = \mathrm{ERM}^c$*, where* $c \geq \frac{m}{N}$*. For any* $v_I \in \mathbb{R}_+^m$*,* $v_{-I} \in \mathbb{R}_+^{N-m}$*, let* $\overline{v}_I$ *be any* $m$ *values that are greater than or equal to the maximal value in* $v_{-I}$*. Then* $\delta_m^{\mathrm{worst}}(v_{-I}) = \delta_I(\overline{v}_I, v_{-I})$*.*

*Proof.* Recall the definition

$$\delta_m^{\mathrm{worst}}(v_{-I}) = \sup_{v_I \in \mathbb{R}_+^m} \delta_I(v_I, v_{-I}) = \sup_{v_I \in \mathbb{R}_+^m} \left\{ 1 - \frac{\inf_{b_I \in \mathbb{R}_+^m} \mathrm{ERM}^c(b_I, v_{-I})}{\mathrm{ERM}^c(v_I, v_{-I})} \right\}.$$

Claim A.1 immediately implies $\delta_m^{\mathrm{worst}}(v_{-I}) = \lim_{v_I \to +\infty} \delta_I(v_I, v_{-I})$. Moreover, since $\mathrm{ERM}^c$ ignores the highest $cN \geq m$ values, we have $\mathrm{ERM}^c(\overline{v}_I, v_{-I}) = \mathrm{ERM}^c(\overline{v}_I', v_{-I})$ as long as both $\overline{v}_I$ and $\overline{v}_I'$ are greater than or equal to $\max v_{-I}$, no matter what they are exactly. Thus $\delta_m^{\mathrm{worst}}(v_{-I}) = \delta_I(\overline{v}_I, v_{-I}) = \delta_I(\overline{v}_I', v_{-I})$. $\square$

Therefore, we will use $\overline{v}_I$ to denote any $m$ values that are greater than or equal to $\max v_{-I}$, for example, $m$ copies of $\max v_{-I}$ or $m$ copies of "$+\infty$". We always have $\delta_m^{\mathrm{worst}}(v_{-I}) = \delta_I(\overline{v}_I, v_{-I})$.

## A.2    Quantiles and Revenue Curves of Value Distributions

For a distribution $F(v)$, define the *quantile* $q(v) = 1 - F(v)$ as a mapping from value space to quantile space. Inversely, $v(q) = q^{-1}(v) = F^{-1}(1-q)$ is the mapping from quantile space to value space (i.e., w.p. $q$ a buyer's value will be at least $v(q)$). Define the *revenue curve* $R(q) = qv(q)$ as the expected revenue for the seller by posting price $v(q)$. Let $R^* = \max_q \{R(q)\}$ denote the optimal revenue the seller can obtain with one bidder, and $q^* = \arg\max_q R(q), v^* = v(q^*)$. When there are several i.i.d. bidders with a regular value distribution, $v^*$ is the optimal reserve price in a second price auction, and such an auction is revenue optimal [30]. Any bounded distribution satisfies $q^* \geq \frac{1}{D}$ because for any $q < \frac{1}{D}$, $qv(q) \leq qD < 1 \leq 1v(1) \leq R^*$. Any MHR distribution has a unique $q^*$ and $q^* \geq \frac{1}{e}$ [22].

## A.3    Concentration Inequality

For a distribution $F$, draw $N$ samples and sort them non-increasingly, $v_1 \geq v_2 \geq \cdots \geq v_N$. Let $q_j = q(v_j)$ denote their quantiles. The ratio $j/N$ is the *empirical quantile* of value $v_j$ since $j/N$ is the quantile of $v_j$ in the uniform distribution over $\{v_1, \ldots, v_N\}$. The following concentration inequality shows that for each value $v_j$, its empirical quantile $j/N$ is close to its true quantile $q_j$ with high probability, when $m$ samples are fixed to be $+\infty$ while other $N - m$ samples are i.i.d. drawn from $F$.

**Lemma A.3.** *Draw* $N - m$ *i.i.d. samples from a distribution* $F$*, and fix* $m$ *samples to be* $+\infty$*. Sort these samples non-increasingly:* $+\infty = v_1 = \cdots = v_m > v_{m+1} \geq \cdots \geq v_N$*. With probability at least* $1 - \delta$ *over the random draw of samples, we have for any* $j > m$*,*

$$\left| q_j - \frac{j}{N} \right| \leq \sqrt{\frac{2\ln(2(N-m)\delta^{-1})}{N-m}} + \frac{\ln(2(N-m)\delta^{-1})}{N-m} + \frac{m}{N}.$$

*Proof.* The value $v_j$ $(j > m)$ is the $(j - m)$th largest value in $N - m$ i.i.d. samples from $F$, by using Bernstein inequality (see e.g., Lemma 5 in Guo et al. [20]), we know that with probability at least $1 - \delta$, $\left| q_j - \frac{j-m}{N-m} \right| \leq \sqrt{\frac{2\ln(2(N-m)\delta^{-1})}{N-m}} + \frac{\ln(2(N-m)\delta^{-1})}{N-m}$. Also note that $|\frac{j}{N} - \frac{j-m}{N-m}| = \frac{(N-j)}{N-m}\frac{m}{N} < \frac{m}{N}$. By triangular inequality, $\left| q_j - \frac{j}{N} \right| \leq \sqrt{\frac{2\ln(2(N-m)\delta^{-1})}{N-m}} + \frac{\ln(2(N-m)\delta^{-1})}{N-m} + \frac{m}{N}$. $\square$

## B    Main Proof: Upper Bounds on Incentive-Awareness Measures

### B.1    Proof of Theorem 1.3

Recall the setting of Definition 1.2: we draw $N$ i.i.d. values $v_1, \ldots, v_N$ from $F$, and we have an additional parameter $m$ which is the number of bids that can be changed in the input of the price learning function. Theorem 1.3 then states an upper bound on $\Delta_{N,m}^{\text{worst}}$ for $P = \text{ERM}^c$. For bounded distributions, the theorem follows immediately from the next lemma which is our main technical lemma. Note that this lemma is useful in establishing the bound on $\Delta_{N,m}^{\text{worst}}$ not only for bounded distributions but also for all other distributions. Throughout, we assume that $v_1, \ldots, v_N$ are sorted, so that $v_1 \geq \cdots \geq v_N$.

**Lemma B.1** (Main Lemma)**.** *Suppose $m = o(\sqrt{N})$. Let $d$ be a constant, $0 < d < 1$. Suppose $\frac{m}{N} \leq c \leq \frac{d}{2}$. Let* E *be the event that the index $k^* = \arg\max_{i > cN}\{iv_i\}$ (which is the index selected by* $\text{ERM}^c$*) satisfies $k^* \leq dN$. For any non-negative distribution $F$,*

$$\Delta_{N,m}^{\text{worst}} \leq O\left( \sqrt[3]{\frac{m^2}{d^8}} \frac{\log^2 N}{\sqrt[3]{N}} \right) + \Pr[\text{E}],$$

*where the probability in* $\Pr[\text{E}]$ *is taken over the random draw of $N - m$ i.i.d. samples from $F$, with other $m$ samples fixed to be $+\infty$.*

To see that this lemma immediately implies the theorem for bounded distribution, choose $d = 1/D$. Since the support of the distribution is bounded by $[1, D]$, $N \cdot v_N \geq N$ while for any $k \leq dN$,

$$kv_k \leq (\frac{1}{D}N)D = N \leq Nv_N.$$

$\text{ERM}^c$ therefore never chooses an index $k \leq dN$ (recall that in case of a tie, $\text{ERM}^c$ picks the larger index). This implies $\Pr[\text{E}] = \Pr[k^* \leq dN] = 0$ and we have the bound in the theorem.

*Remark.* For MHR distributions, Lemma D.5 shows that $\Pr[\text{E}] = O\left(\frac{1}{N}\right)$ if we choose $d = \frac{1}{2e}$, so Lemma B.1 already gives an upper bound on $\Delta_{N,m}^{\text{worst}}$. However, we can use some additional properties of MHR distributions to strengthen the bound on the first term in the main lemma to $O\left(\frac{m}{d^{7/2}} \frac{\log^3 N}{\sqrt{N}}\right)$, as explained in Appendix D.3.

### B.2    Proof of the Main Lemma (Lemma B.1)

Let $\overline{v}_I$ be any $m$ values that are greater than the maximal value in $v_{-I}$. By Claim A.2, $\delta_m^{\text{worst}}(v_{-I}) = \delta_I(\overline{v}_I, v_{-I})$. Thus,

$$\Delta_{N,m}^{\text{worst}} = \mathbb{E}[\delta_m^{\text{worst}}(v_{-I})] = \mathbb{E}[\delta_I(\overline{v}_I, v_{-I})] = \int_0^1 \Pr[\delta_I(\overline{v}_I, v_{-I}) > \eta]\mathrm{d}\eta$$

$$\leq \int_0^1 \left( \Pr[\delta_I(\overline{v}_I, v_{-I}) > \eta \mid \overline{\text{E}}] \Pr[\overline{\text{E}}] + \Pr[\text{E}] \right) \mathrm{d}\eta$$

$$= \int_0^1 \Pr[\delta_I(\overline{v}_I, v_{-I}) > \eta \wedge \overline{\text{E}}]\mathrm{d}\eta + \Pr[\text{E}], \tag{3}$$

where $\overline{\text{E}}$ denotes the complement of E. Then the main effort is to upper-bound $\Pr[\delta_I(\overline{v}_I, v_{-I}) > \eta \wedge \overline{\text{E}}]$. After the random draw of $v_{-I}$, we sort all values non-increasingly, denoted by $\overline{v}_1 = \cdots = \overline{v}_m \geq v_{m+1} \geq \cdots \geq v_N$, and let $\overline{q}_1 = \cdots = \overline{q}_m \leq q_{m+1} \leq \cdots \leq q_N$ be their quantiles, where $q_j = q(v_j)$. We use a concentration inequality (Lemma A.3) to argue that for each value $v_j$, its empirical quantile $j/N$ should be close to its true quantile $q_j$ with high probability, as follows:

**Claim B.2.** *Define event* Conc*:*

$$\text{Conc} = \left[ \forall j > m, \left| q_j - \frac{j}{N} \right| \leq 2\sqrt{\frac{4\ln(2(N-m))}{N-m}} + \frac{m}{N} \right],$$

*then $\Pr[\overline{\text{Conc}}] \leq \frac{1}{N-m}$, where the probability is over the random draw of the $N-m$ samples $v_{-I}$.*

*Proof.*  Set $\delta = \frac{1}{N-m}$ in Lemma A.3. $\qquad\qquad\square$

Now, define $G(\eta) = \Pr[\delta_I(\overline{v}_I, v_{-I}) > \eta \wedge \overline{\text{E}} \wedge \text{Conc}]$ for $0 \leq \eta \leq 1$. We have

$$\Pr[\delta_I(\overline{v}_I, v_{-I}) > \eta \wedge \overline{\text{E}}] \leq G(\eta) + \frac{1}{N-m}. \tag{4}$$

490 **Lemma B.3.** *There exists a constant $C = \Theta\left(\frac{m \log^3 N}{d^4 \sqrt{N}}\right)$ such that $\eta > C^{2/3} \Rightarrow G(\eta) \leq \frac{C}{\eta^{3/2}}$.*

491 Finally we upper-bound the integral in (3):

$$
\int_0^1 \Pr[\delta_I(\overline{v}_I, v_{-I}) > \eta \wedge \overline{E}] d\eta \leq \int_0^1 \left( G(\eta) + \frac{1}{N-m} \right) d\eta \qquad \text{By (4)}
$$

$$
\leq \int_0^{C^{\frac{2}{3}}} 1 d\eta + \int_{C^{\frac{2}{3}}}^1 \frac{C}{\eta^{\frac{3}{2}}} d\eta + \frac{1}{N-m} \qquad \text{By Lemma B.3}
$$

$$
\leq 3C^{\frac{2}{3}} + \frac{1}{N-m} = O\left( \sqrt[3]{\frac{m^2 \log^6 N}{d^8 N}} \right) + \frac{1}{N-m} = O\left( \sqrt[3]{\frac{m^2}{d^8}} \frac{\log^2 N}{\sqrt[3]{N}} \right),
$$

492 which, together with (3), concludes the proof of Lemma B.1.

## B.3 Proof of Lemma B.3

494 Recall that we need to upper-bound $G(\eta) = \Pr[\delta_I(\overline{v}_I, v_{-I}) > \eta \wedge \overline{E} \wedge \text{Conc}]$ by $\Theta\left(\frac{m \log^3 N}{d^4 \sqrt{N}} \frac{1}{\eta^{3/2}}\right)$. We do
495 this via a union bound of $M+1$ events, where $M$ is a number to be chosen later. Each event is parameterized
496 by $\eta_t, \theta_t$ for $t = 0, \ldots, M$ which are chosen to satisfy the following conditions:

497 • $\eta_0 = \frac{1}{2}\eta, \eta_1 = \eta, \eta_2 = 2\eta$.

498 • For $t \geq 3$, $\eta_t$ can be chosen arbitrarily, as long as $\eta_2 < \eta_3 < \cdots < \eta_M < 1$.

499 • $\eta_{M+1} = 1$.

500 • $\theta_0 = 1$, and $\theta_t = \frac{\eta}{2\eta_{t+1}}$ for $t = 1, \ldots, M$.

501 Define the following $M+1$ events $\text{Bad}(\eta_t, \theta_t)$, where $t = 0, \ldots, M$:

$$
\text{Bad}(\eta_t, \theta_t) = \left[ \text{there exists } j \geq k^* \text{ such that } \left\{ \begin{array}{l} v_j \leq (1 - \eta_t) v_{k^*} \\ j v_j \geq k^* v_{k^*} - \frac{m}{\theta_t} v_{k^*} \end{array} \right. \right] \qquad (5)
$$

502 The next lemma shows that the union of these events contains the event $[\delta_I(\overline{v}_I, v_{-I}) > \eta] \wedge \overline{E}$.

503 **Lemma B.4.** *Suppose $\frac{2m}{dN} < \eta < 1$ and that the parameters $\eta_t, \theta_t$ satisfy the above conditions. If $\delta_I(\overline{v}_I, v_{-I}) >$*
504 *$\eta$ and $k^* > dN$, then there exists $t \in \{0, \ldots, M\}$ such that the event $\text{Bad}(\eta_t, \theta_t)$ holds.*

505 The proof of this lemma is given in Appendix B.4. Moreover, the next lemma upper-bounds the probability of
506 each of these bad events, when assuming that Conc holds as well.

507 **Lemma B.5.** *If $\eta_t$ and $\theta_t$ are at least $\Omega\left(\frac{m}{d}\sqrt{\frac{\log(N-m)}{N-m}}\right)$ (for some constant in $\Omega$ to be detailed in the proof),*

508 *then $\Pr[\text{Bad}(\eta_t, \theta_t) \wedge \overline{E} \wedge \text{Conc}] = O\left(\frac{m \log^2 N}{d^4 \theta_t \sqrt{\eta_t^3 N}}\right)$.*

509 The proof of Lemma B.5 is in Appendix B.5. Now,

$$
\Pr[\delta_I(\overline{v}_I, v_{-I}) > \eta \wedge \overline{E} \wedge \text{Conc}] \leq \sum_{t=0}^M \Pr[\text{Bad}(\eta_t, \theta_t) \wedge \overline{E} \wedge \text{Conc}] \qquad \text{Lemma B.4}
$$

$$
= \sum_{t=0}^M O\left( \frac{m \log^2 N}{d^4 \theta_t \sqrt{\eta_t^3 N}} \right) \qquad \text{Lemma B.5}
$$

$$
= O\left( \frac{m \log^2 N}{d^4 \sqrt{N}} \cdot \sum_{t=0}^M \frac{\eta_{t+1}}{\eta} \frac{1}{\eta_t^{3/2}} \right) \qquad \text{Definition of } \theta_t
$$

510 Note that because $\eta_t, \theta_t \geq \frac{\eta}{2}$, the condition of Lemma B.5 is satisfied under the assumption that $\eta \geq$
511 $\Theta\left(\left(\frac{m \log^3 N}{d^4 \sqrt{N}}\right)^{2/3}\right)$ in Lemma B.3. Finally, we choose a sequence of $\{\eta_t\}$ to make the above summation small
512 enough:

**Claim B.6.** *There exist an integer $M$ and parameters $\eta_3, \ldots, \eta_M$ that satisfy the conditions described above, such that*

$$\sum_{t=0}^{M} \frac{\eta_{t+1}}{\eta} \frac{1}{\eta_t^{3/2}} = O\left(\frac{\log\log(N-m)}{\eta^{3/2}}\right),$$

*assuming $\eta = \Omega\left(\frac{m}{d}\sqrt{\frac{\log(N-m)}{N-m}}\right)$.*

The proof of this claim is given in Appendix B.7. To conclude the proof,

$$\Pr[\delta_I(\overline{v}_I, v_{-I}) > \eta \,\wedge\, \overline{\mathrm{E}} \,\wedge\, \mathrm{Conc}] \leq O\left(\frac{m\log^2 N}{d^4\sqrt{N}} \cdot \frac{\log\log(N-m)}{\eta^{3/2}}\right) = O\left(\frac{m\log^3 N}{d^4\sqrt{N}} \frac{1}{\eta^{3/2}}\right).$$

*Remark.* This proof is inspired by a proof in Yao [31]. We improve upon that proof in two aspects: (1) Our definition of a sequence of bad events (Lemma B.4) improves upon similar single bad events defined in Yao [31] and Lavi et al. [26]; (2) Yao [31] only considers bounded and continuous distributions, while our proof works for arbitrary distributions. This is because Yao [31] works in the value space when upper-bounding the probability of bad events over the random draw of values $v_{-I}$ (Lemma B.5), but we work in the quantile space, which circumvents the boundedness assumption and deals with discontinuity. To argue in the quantile space, we need $\mathrm{Conc}$ to show that $q_j v_j$ approximates $j v_j$ in the proof of Lemma B.5.

## B.4 Proof of Lemma B.4

Suppose $\delta_I(\overline{v}_I, v_{-I}) > \eta$ and $k^* > dN$. By definition, there exist $m$ bids $b_I \in \mathbb{R}_+^m$ such that $\mathrm{ERM}^c(b_I, v_{-I}) < (1-\eta)v_{k^*}$. Without loss of generality, we can assume that all $m$ bids are identical, as shown in the following claim:

**Claim B.7.** *For any sorted $v = (v_1 \geq \cdots \geq v_N)$. Let $I = \{1, \ldots, m\}$, let*

$$b^* = \underset{b \in \mathbb{R}_+}{\arg\min}\, \mathrm{ERM}^c(\overbrace{b, \ldots, b}^{m\ copies}, v_{-I}).$$

*then $b^* = \mathrm{ERM}^c(b^*, \ldots, b^*, v_{-I}) = \min_{b_I \in \mathbb{R}_+^m} \mathrm{ERM}^c(b_I, v_{-I})$.*

*Proof.* We will show that for any vector of $m$ bids $b_I = (b_1, \ldots, b_m)$ that minimizes $\mathrm{ERM}^c(b_I, v_{-I})$, we can construct another vector $b'_I = (b, \ldots, b)$ such that $\mathrm{ERM}^c(b'_I, v_{-I}) = \mathrm{ERM}^c(b_I, v_{-I}) = b$. Because $b_I$ minimizes $\mathrm{ERM}^c(b_I, v_{-I})$, we can assume that there is a bid $b_{i^*}$ such that $\mathrm{ERM}^c(b_I, v_{-I}) = b_{i^*}$ (otherwise, we can decrease the bids in $b_I$ without increasing the price). Let $b = b_{i^*}$. For any $b_j > b$, decrease $b_j$ to $b$, then the price does not change. For any $b_j < b$, increase $b_j$ to $b$, then the price does not increase; and if the price decreases, then it contradicts the fact that $\mathrm{ERM}^c(b_I, v_{-I})$ is minimized. In this way, we change all bids in $b_I$ to $b$, without affecting the price. $\qquad\square$

By Claim B.7, there exists $b \in \mathbb{R}_+$ which equals $\mathrm{ERM}^c(b, \ldots, b, v_{-I})$ and satisfies

$$b < (1-\eta)v_{k^*}. \tag{6}$$

Choose index $i$ for which $v_i \geq b > v_{i+1}$. Assume for now $i \leq N-1$, we will postpone the analysis for $i = N$ to the end. Now we show that setting $j = i$ or $i+1$ will satisfy the lemma. Clearly $i \geq k^*$. The change of the bids vector caused by $b$ is:

$$(\overline{v}_1, \ldots, \overline{v}_m, v_{m+1}, \ldots, v_{k^*}, \ldots, v_i, v_{i+1}, \ldots, v_N) \to (v_{m+1}, \ldots, v_{k^*}, \ldots, v_i, \overbrace{b, \ldots, b}^{m\ times}, v_{i+1}, \ldots, v_N).$$

Note that $k^* - m > dN - m \geq cN$, so $v_{k^*}$ will not be ignored by $\mathrm{ERM}^c$ after the change of bids. Then in order for $b$ to be chosen by $\mathrm{ERM}^c$, we need:

$$i \cdot b \geq (k^* - m) \cdot v_{k^*} = k^* v_{k^*} - m v_{k^*}. \tag{7}$$

We will choose $j$ depending on how large $v_i$ is:

(a) If $v_i < (1 - \frac{1}{2}\eta)v_{k^*} = (1-\eta_0)v_{k^*}$, we set $j = i$ and $t = 0$. Clearly, $iv_i \geq ib \geq k^* v_{k^*} - m v_{k^*}$.

(b) If $v_i \geq (1 - \frac{1}{2}\eta)v_{k^*}$, then we set $j = i+1$ and choose the $t$ ($1 \leq t \leq M$) such that

$$(1 - \eta_t)v_{k^*} \geq v_{i+1} > (1 - \eta_{t+1})v_{k^*}. \tag{8}$$

To see why $(i+1)v_{i+1} \geq k^* v_{k^*} - \frac{2\eta_{t+1}}{\eta} m v_{k^*}$ holds, first we write $b$ as a convex combination of $v_i$ and $v_{i+1}$: $b = (1-\lambda)v_i + \lambda v_{i+1}$. From (6) and (7), we immediately get

$$(1-\eta)v_{k^*} > (1-\lambda)v_i + \lambda v_{i+1}, \tag{9}$$

$$(1-\lambda)iv_i + \lambda(i+1)v_{i+1} \geq k^* v_{k^*} - m v_{k^*}. \tag{10}$$

Equation (10) further implies $\lambda(i+1)v_{i+1} \geq k^* v_{k^*} - (1-\lambda)k^* v_{k^*} - m v_{k^*}$. Divide by $\lambda$,

$$(i+1)v_{i+1} \geq k^* v_{k^*} - \frac{m}{\lambda} v_{k^*}.$$

Then it remains to lower-bound $\lambda$ by $\theta_t$. Intuitively, since $v_i$ is larger than $(1-\frac{\eta}{2})v_{k^*}$ but $b < (1-\eta)v_{k^*}$, the coefficient of $v_{i+1}$ cannot be too small. Formally, from (9) and (8), we have:

$$(1-\eta)v_{k^*} > (1-\lambda)(1-\frac{1}{2}\eta)v_{k^*} + \lambda(1-\eta_{t+1})v_{k^*},$$

$$\implies 1-\eta > 1 - \frac{1}{2}\eta - \lambda(\eta_{t+1} - \frac{1}{2}\eta)$$

$$\implies \lambda > \frac{\frac{1}{2}\eta}{\eta_{t+1} - \frac{1}{2}\eta} \geq \frac{\eta}{2\eta_{t+1}} = \theta_t,$$

concluding the proof of this case.

Finally we return to the analysis for $i = N$. If $\frac{k^*}{N} < 1 - \frac{1}{2}\eta$, then $v_N \leq \frac{k^* v_{k^*}}{N} < (1-\frac{1}{2}\eta)v_{k^*}$, so the above argument (a) can be reused. Otherwise, from (6) and (7), we have

$$(1-\eta)v_{k^*} > b \geq \frac{(k^* - m)v_{k^*}}{N} \geq (1 - \frac{1}{2}\eta - \frac{m}{N})v_{k^*},$$

which contradicts the assumption that $\eta > \frac{2m}{dN}$.

## B.5 Proof of Lemma B.5

For convenience we drop the subscript $t$ and just write $\eta = \eta_t, \theta = \theta_t$. Recall that we need to upper-bound $\Pr[\mathrm{Bad}(\eta, \theta) \wedge \overline{\mathrm{E}} \wedge \mathrm{Conc}]$ where:

- $\mathrm{Bad}(\eta, \theta)$ implies that there exists $j \geq k^*$ such that $v_j \leq (1-\eta)v_{k^*}$ and $jv_j \geq k^* v_{k^*} - \frac{m}{\theta} v_{k^*}$.

- $\overline{\mathrm{E}}$ is $k^* \geq dN$.

- $\mathrm{Conc}$ requires that $|q_j - \frac{j}{N}| \leq 2\sqrt{\frac{4 \ln(2(N-m))}{N-m}} + \frac{m}{N}$ for any $j > m$.

Define

$$H = \frac{m}{d\theta - \frac{m}{N}} \qquad \text{and} \qquad h = \frac{1}{2}\left( d\eta - 4\sqrt{\frac{4\ln(2(N-m))}{N-m}} - \frac{4m}{N\theta} \right).$$

Assume $H, h > 0$, which can be satisfied when $\eta$ and $\theta$ are at least $\Omega\left( \frac{m}{d}\sqrt{\frac{\log(N-m)}{N-m}} \right)$.

**Claim B.8.** *The event* $[\mathrm{Bad}(\eta, \theta) \wedge \overline{\mathrm{E}} \wedge \mathrm{Conc}]$ *implies that there exists $j \geq k^*$ which satisfies:*

1. $jv_j \leq k^* v_{k^*} \leq (j+H)v_j$;

2. $q_j - q_{k^*} \geq 2h$.

*Proof of Claim B.8.* Choose the $j$ in $\mathrm{Bad}(\eta, \theta)$ which satisfies $jv_j \geq k^* v_{k^*} - \frac{m}{\theta} v_{k^*}$. To see why the first inequality holds, note that $dN v_{k^*} \leq k^* v_{k^*} \leq jv_j + \frac{m}{\theta} v_{k^*} \leq N v_j + \frac{m}{\theta} v_{k^*}$, subtracting the first and forth term, we get $(dN - \frac{m}{\theta})v_{k^*} \leq N v_j$, further implying $k^* v_{k^*} \leq jv_j + \frac{m}{\theta}\frac{N v_j}{(dN-m/\theta)}$, which is the first inequality.

Now consider the second inequality. Since $\mathrm{Bad}(\eta, \theta)$ requires $jv_j \geq k^* v_{k^*} - \frac{m}{\theta} v_{k^*}$ and $v_j \leq (1-\eta)v_{k^*}$, we have

$$j \geq \frac{k^* v_{k^*} - \frac{m}{\theta} v_{k^*}}{(1-\eta)v_{k^*}} = \frac{k^* - \frac{m}{\theta}}{1-\eta} \geq (k^* - \frac{m}{\theta})(1+\eta) = k^* + k^*\eta - \frac{m}{\theta}(1+\eta) \geq k^* + dN\eta - 2\frac{m}{\theta},$$

and dividing by $N$,

$$\frac{j}{N} - \frac{k^*}{N} \geq d\eta - 2\frac{m}{N\theta}.$$

Using the condition Conc on $j$ and $k^*$, we can derive the relationship between $q_j$ and $q_{k^*}$ by simple calculation:

$$q_j - q_{k^*} \geq \left(d\eta - 2\frac{m}{N\theta}\right) - 2\left(2\sqrt{\frac{4\ln(2(N-m))}{N-m}} + \frac{m}{N}\right) \geq 2h.$$

$\square$

Divide the quantile space $[0,1]$ into $[0, d/2]$ and $(1-d/2)/h$ equal-length intervals with length $h$,

$$[0,1] = [0, \frac{d}{2}] \cup I_1 \cup I_2 \cup \cdots \cup I_{\frac{1-d/2}{h}}, \tag{11}$$

where $I_l = (d/2 + (l-1)h, d/2 + lh]$. Thus a uniformly random draw of quantile falls into $I_l$ with probability $h$. Define $i_l^*$ and $i_{<(l+1)}^*$:

$$i_l^* = \arg\max_{i>cN}\left\{iv_i \mid q_i \in I_l\right\} \text{ or } i_l^* = \emptyset \text{ if there is no such } i.$$

$$i_{<(l+1)}^* = \arg\max_{i>cN}\left\{iv_i \mid q_i \in I_1 \cup \cdots \cup I_l\right\} \text{ or } i_{<(l+1)}^* = \emptyset \text{ if there is no such } i.$$

And $A_l \stackrel{\text{def}}{=} i_l^* v_{i_l^*}$, $A_{<(l+1)} \stackrel{\text{def}}{=} i_{<(l+1)}^* v_{i_{<(l+1)}^*}$. Moreover, define event $W_l$ for each $l$,

$$W_l = \left[\{i_{l+2}^* \neq \emptyset\} \wedge \{i_{<(l+1)}^* \neq \emptyset\} \wedge \{A_{l+2} \leq A_{<(l+1)} \leq A_{l+2} + H\widetilde{v}_{l+2}\}\right],$$

where $\widetilde{v}_{l+2} \stackrel{\text{def}}{=} v(d/2 + (l+1)h)$ is the upper bound on the values with quantiles in $I_{l+2}$. We argue that if the event $[\mathrm{Bad}(\eta, \theta) \wedge \overline{\mathrm{E}} \wedge \mathrm{Conc}]$ holds then there must exist an index $l$ such that $W_l$ holds. To see this, consider the index $j$ that is promised to exist in $\mathrm{Bad}(\eta, \theta)$ and choose the index $l$ such that $q_j \in I_{l+2}$. Note that $[\overline{\mathrm{E}} \wedge \mathrm{Conc}]$ implies $q_j \geq q_{k^*} > d/2$, so both $q_j$ and $q_{k^*}$ must fall in $I_1 \cup I_2 \cup \cdots$. To see why $W_l$ must hold, note that:

- $A_{l+2} \leq A_{<(l+1)}$ since $q_j - q_{k^*} > 2h$, implying $q_{k^*} \in I_{<(l+1)}$ and $A_{<(l+1)} = k^* v_{k^*}$.
- $A_{<(l+1)} \leq A_{l+2} + H\widetilde{v}_{l+2}$ since $k^* v_{k^*} \leq (j+H)v_j \leq i_{l+2}^* v_{i_{l+2}^*} + H\widetilde{v}_{l+2}$.

Therefore, a union bound over $\Pr[W_l]$ suffices to prove that $\Pr[\mathrm{Bad}(\eta, \theta) \wedge \overline{\mathrm{E}} \wedge \mathrm{Conc}]$ is small. The idea to bound $\Pr[W_l]$ is a refinement of Yao [31]: Note that there is an interval $I_{l+1}$ with length $h$ between $I_{l+2}$ and $I_{<(l+1)}$ and consider the number $X$ of quantiles falling into $I_{l+1}$. There is enough randomness in $X$ as its variance is $\Omega(hN)$, implying that the difference between the rankings of any pair of quantiles in $I_{l+2}$ and $I_{<(l+1)}$ varies broadly. As a result, it's unlikely that $A_{<(l+1)}$ will fall in the short interval $[A_{l+2}, A_{l+2} + H\widetilde{v}_{l+2}]$. Formally, we will prove that

**Lemma B.9.** *For any $l$*, $\Pr[W_l] \leq O(\frac{H\log^2 N}{\sqrt{h}d^3 N})$.

The proof of Lemma B.9 is in Appendix B.6. To conclude,

$$\Pr[\mathrm{Bad}(\eta, \theta) \wedge \overline{\mathrm{E}} \wedge \mathrm{Conc}] \leq \sum_{l=1}^{\frac{1-d/2}{h}} \Pr[W_l] \leq \frac{1}{h}O\left(\frac{H\log^2 N}{\sqrt{h}d^3 N}\right) = O\left(\frac{m\log^2 N}{d\theta\sqrt{(d\eta)^3 d^3 N}}\right),$$

where the last equality is because $H = O(\frac{m}{d\theta})$ and $h = \Omega(d\eta)$ under the assumption that $\eta$ and $\theta$ are at least $\Omega(\frac{m}{d}\sqrt{\frac{\log(N-m)}{N-m}})$.

## B.6 Proof of Lemma B.9

We need to upper-bound $\Pr[W_l]$ over the random draw of $v_{-I}$, or in quantile space, $q_{-I}$, which are $N-m$ i.i.d. random draws from $\mathrm{Uniform}[0,1]$. Let $N_L$ be the number of quantile draws that are in $L \stackrel{\text{def}}{=} [0, d/2] \cup I_1 \cup \cdots \cup I_{l+1}$. Suppose we draw the quantiles in the following procedure: first determine $N_L$, then draw $N - m - N_L$ quantiles that are not in $L$; finally draw $N_L$ quantiles that are in $L$.

Note that $N_L$ follows a binomial distribution, and a Chernoff bound implies that

$$\Pr[N_L \geq \frac{d}{4}(N-m)] \geq 1 - \exp\left(-\frac{d(N-m)}{16}\right). \tag{12}$$

We thus assume $N_L \geq d(N-m)/4$.

606  Then draw $N - m - N_L$ quantiles from $[0,1]\backslash L$, so $i^*_{l+2}, v_{i^*_{l+2}}$ and $A_{l+2}$ are determined. Suppose $i^*_{l+2} \neq \emptyset$;
607  otherwise, $W_l$ does not hold.

608  Now we draw $N_L$ quantiles, $q^{(1)}, \ldots, q^{(N_L)}$ from $L$. Consider the increment of $A_{<(l+1)}$, as a sequence
609  $A^{(t)}, t = 1, \ldots, N_L$. After the time $t - 1$ when $A^{(t-1)} \geq A_{l+2}$, the index $i^*_{<(l+1)}$ is no longer $\emptyset$. When one
610  more sample $q^{(t)}$ is generated, there are three cases:

611  　　1. If $q^{(t)} \in [0, d/2]$, $A^{(t-1)}$ increases by at least $\widetilde{v}_{l+2}$. This is because each term $iv_i^{(t-1)}$ increases to
612  　　　$(i+1)v_i^{(t)} \geq iv_i^{(t-1)} + \widetilde{v}_{l+2}$, for any $i$ such that $q_i \in I_1, \ldots, I_l$.

613  　　2. If $q^{(t)} \in I_1 \cup \cdots \cup I_l$, then $A^{(t-1)}$ does not decrease.

614  　　3. If $q^{(t)} \in I_{l+1}$, $A^{(t-1)}$ does not change.

615  Let $s$ be the number of quantiles that are not in $I_{l+1}$, and $A^{(t_1)}, \ldots, A^{(t_s)}$ be those steps, and write $B^{(i)} \stackrel{\text{def}}{=}$
616  $(A^{(t_i)} - A_{l+2})/\widetilde{v}_{l+2}$ for $i = 1, \ldots, s$. We have $A_{<(l+1)} = \widetilde{v}_{l+2}B^{(s)} + A_{l+2}$. Then our task is to analyze the
617  probability that $B^{(s)} \in [0, H]$.

618  We can think of the generation of $B^{(s)}$ as follows: regardless of $s$, first generate an infinite sequence
619  $B^{(1)}, B^{(2)}, \ldots$, where at each step $i$ the value $B^{(i)}$ is increased by 1 with probability at least $\Pr[q \in [0, d/2] \mid$
620  $q \in L] \geq d/2$. Then pick an index $s$ by a binomial distribution $Bin(N_L, 1 - \Pr[q \in I_{l+1} \mid q \in L])$. Then
621  the $s$-th value in the infinite sequence $\{B^{(i)}\}$ is chosen as $B^{(s)}$. Note that $Bin(N_L, 1 - \Pr[q \in I_{l+1} \mid q \in L])$
622  is dominated by $Bin(N_L, 1 - h)$, so the probability that $B^{(s)}$ takes on any one of the values in the sequence
623  $\{B^{(i)}\}$ is at most $\Pr[s = i] = O(1/\sqrt{hN_L})$.

624  Then we consider the length of the sub-sequence where $B^{(i)} \in [0, H]$. Intuitively, the expected number of steps
625  for $B^{(i)}$ to increase by $H$, is at most $H/(d/2)$. The probability that it takes more than $2H(\log N_L)^2/d$ steps
626  implies that the sum of $2H(\log N_L)^2/d$ i.i.d. Bernoulli variables whose success probability is at least $d/2$ does
627  not reach $H$, which can be bounded by a Chernoff bound:

$$\Pr[\text{Length} > \frac{2H(\log N_L)^2}{d}] \leq \Pr[Bin\left(\frac{2H(\log N_L)^2}{d}, \frac{d}{2}\right) < H]$$

$$\leq \exp\left(-\frac{1}{2}H(\log N_L)^2\left(1 - \frac{1}{(\log N_L)^2}\right)^2\right)$$

$$= O\left(\exp\left(-\frac{1}{8}(\log N_L)^2\right)\right)$$

$$= O\left(N_L^{-\frac{1}{8}\log N_L}\right).$$

628  Assuming Length $\leq (2H(\log N_L)^2)/d$, the probability that $B^{(s)} \in [0, H]$ can be bounded by a union bound:

$$\sum_{i:B^{(i)}\in[0,H]} \Pr[s = i] \leq \text{Length} \cdot O\left(\frac{1}{\sqrt{hN_L}}\right) \leq O\left(\frac{H(\log N_L)^2}{d\sqrt{hN_L}}\right).$$

629  Therefore,

$$\Pr[W_l] \leq O\left(\frac{H(\log N_L)^2}{d\sqrt{hN_L}}\right) + O\left(N_L^{-\frac{1}{8}\log N_L}\right)$$

$$\leq O\left(\frac{H(\log N)^2}{d\sqrt{hdN}}\right) + \left(\frac{d}{4}(N-m)\right)^{-\frac{1}{8}\log\frac{d}{4}(N-m)} + \exp\left(-\frac{d(N-m)}{16}\right) \qquad \text{By (12)}$$

$$= O\left(\frac{H(\log N)^2}{\sqrt{hd^3N}}\right).$$

## B.7  Proof of Claim B.6

631  We need to show that there exist an integer $M$ and parameters $\eta_0 = \frac{1}{2}\eta < \eta_1 = \eta < \eta_2 = 2\eta < \eta_3 < \cdots <$
632  $\eta_M < \eta_{M+1} = 1$, such that

$$\sum_{t=0}^{M} \frac{\eta_{t+1}}{\eta}\frac{1}{\eta_t^{3/2}} = O\left(\frac{\log\log(N-m)}{\eta^{3/2}}\right).$$

We start with:

$$\sum_{t=0}^{M} \frac{\eta_{t+1}}{\eta} \frac{1}{\eta_t^{3/2}} = \frac{1}{\eta^{3/2}} \sum_{t=0}^{M} \frac{\eta_{t+1}/\eta}{(\eta_t/\eta)^{3/2}} = \frac{1}{\eta^{3/2}} \left( O(1) + \sum_{t=2}^{M} \frac{\eta_{t+1}/\eta}{(\eta_t/\eta)^{3/2}} \right) \tag{13}$$

Let $\eta_{t+1}/\eta = (\eta_t/\eta)^{3/2}$ for any $t \geq 2$. We can recursively compute $\eta_t$ until the maximum step $t = M$ which satisfies $\eta_M < 1$. Then (13) is upper-bounded by $\frac{1}{\eta^{3/2}}(O(1) + M)$. By our construction of $\{\eta_t\}$, we have

$$\frac{\eta_M}{\eta} = (\frac{\eta_2}{\eta})^{\frac{3}{2}M-2} = 2^{\frac{3}{2}M-2} < \frac{1}{\eta}.$$

Thus,

$$M < \log_{3/2} \log_2 \frac{1}{\eta} + 2 = O(\log\log\frac{1}{\eta}) = O(\log\log(N-m)),$$

where the last equality follows from the assumption that $\eta = \Omega\left(\frac{m}{d}\sqrt{\frac{\log(N-m)}{N-m}}\right)$. Thus, the summation (13) becomes

$$\sum_{t=0}^{M} \frac{\eta_{t+1}}{\eta} \frac{1}{\eta_t^{3/2}} = O\left(\frac{\log\log(N-m)}{\eta^{3/2}}\right).$$

as required.

# C   Missing Proofs From Section 2

## C.1   Proof of Theorem 2.1 (for Bounded Distributions)

*Proof of Theorem 2.1 for bounded Distributions.* First consider the reduction of reserve price caused by the deviation of bidder $i$. The true values of all bidders in the first phase are $(v_I, v_{-I})$, where bidder $i$'s true values are $v_I \in \mathbb{R}_+^{m_{i,1}}$. When other bidders bid $v_{-I}$ truthfully, and bidder $i$ bids $b_I$ instead, the reserve price $p$ changes from $P(v_I, v_{-I})$ to $P(b_I, v_{-I})$ and the change is at most

$$P(v_I, v_{-I}) - P(b_I, v_{-I}) \leq P(v_I, v_{-I})\delta_I(v_I, v_{-I}) \leq P(v_I, v_{-I})\delta_{m_{i,1}}^{\text{worst}}(v_{-I}) \leq D\delta_{m_1}^{\text{worst}}(v_{-I}),$$

by Definition 1.2 and by the fact that all values are upper-bounded by $D$. Consider the increase of utility in the second phase. We claim that for any two possible reserve prices $p_2 \leq p_1$, for any $v \in \mathbb{R}_+$, for any $K_2 \geq 1$, we have

$$u^{K_2}(v, p_2) - u^{K_2}(v, p_1) \leq p_1 - p_2. \tag{14}$$

To see this, first re-write $u^{K_2}(v, p)$ in (2) as

$$u^{K_2}(v, p) = \mathbb{E}_{X_2,\dots,X_{K_2} \sim F}\left[(v - \max\{p, X^*\})^+\right],$$

where $X^* \stackrel{\text{def}}{=} \max\{X_2, \dots, X_{K_2}\}$ and $(x)^+ \stackrel{\text{def}}{=} \max\{x, 0\}$. Note that $(x)^+ - (y)^+ \leq |x-y|$, thus

$$u^{K_2}(v, p_2) - u^{K_2}(v, p_1) = \mathbb{E}\left[(v - \max\{p_2, X^*\})^+ - (v - \max\{p_1, X^*\})^+\right]$$
$$\leq \mathbb{E}\left[|\max\{p_1, X^*\} - \max\{p_2, X^*\}|\right] \leq \mathbb{E}\left[|p_1 - p_2|\right] = p_1 - p_2.$$

For the first phase we have $U_i^{\mathcal{M}}(\boldsymbol{v}_i, b_I, v_{-I}) \leq U_i^{\mathcal{M}}(\boldsymbol{v}_i, v_I, v_{-I})$ since $\mathcal{M}$ is truthful. Thus, by (1) and (14) the difference in interim utilities is at most

$$\mathbb{E}_{\boldsymbol{v}_{-i}}\left[U_i^{\text{TP}}(\boldsymbol{v}_i, b_I, v_{-I}) - U_i^{\text{TP}}(\boldsymbol{v}_i, v_I, v_{-I})\right]$$
$$\leq \mathbb{E}_{\boldsymbol{v}_{-i}}\left[U_i^{\mathcal{M}}(\boldsymbol{v}_i, b_I, v_{-I}) - U_i^{\mathcal{M}}(\boldsymbol{v}_i, v_I, v_{-I})\right]$$
$$+ \mathbb{E}_{\boldsymbol{v}_{-i}}\left[\sum_{t=m_{i,1}+1}^{m_{i,1}+m_{i,2}}\left[u^{K_2}\left(v_{i,t}, P(b_I, v_{-I})\right) - u^{K_2}\left(v_{i,t}, P(v_I, v_{-I})\right)\right]\right]$$
$$\leq 0 + \mathbb{E}_{\boldsymbol{v}_{-i}}\left[m_2(P(v_I, v_{-I}) - P(b_I, v_{-I}))\right] \leq \mathbb{E}_{\boldsymbol{v}_{-i}}\left[m_2 D\delta_{m_1}^{\text{worst}}(v_{-I})\right] = m_2 D\Delta_{T_1 K_1, m_1}^{\text{worst}},$$

which indicates that truthful bidding is an $\epsilon$-BNE, where $\epsilon = m_2 D\Delta_{T_1 K_1, m_1}^{\text{worst}}$. This concludes the proof for bounded distributions. $\qquad\square$

*Remark.* The proof for MHR distributions is trickier since the difference $P(v_I, v_{-I}) - P(b_I, v_{-I})$ can be unbounded. Intuitively, the probability that $P(v_I, v_{-I})$ will be higher than $(1 + o(1))v^*$ ($v^*$ is defined in the statement of the lemma) is exponentially small, and the main effort is to show that the expected difference $P(v_I, v_{-I}) - P(b_I, v_{-I})$ multiplied by this exponentially small probability is negligible. Full details are given Appendix D.2.

## C.2 Proof of Theorem 2.2

The bound on approximate truthfulness, i.e., $\epsilon_1$, follows from Theorem 2.1 and Theorem 1.3, where we first obtain the bound on $\Delta_{T_1 K_1, m_1}^{\text{worst}}$ from Theorem 1.3 by setting $N = T_1 K_1$ and $m = m_1$ and then replace $m_2 m_1$ with $O(\overline{m}^2)$ for MHR distribution and replacing $m_2 m_1^{2/3}$ with $O(\overline{m}^{5/3})$ for bounded distribution.

It remains to consider revenue, where we will use sample complexity results to obtain the convergence rate of the revenue loss, i.e., $\epsilon_2$. Let $rev_1, rev_2$ be the expected revenues of the two phases in $\text{TP}(\mathcal{M}, P; \boldsymbol{T}, \boldsymbol{m}, \boldsymbol{K}, S)$, and $rev^*$ be the revenue obtained by using Myerson's auction in all rounds, i.e., $rev^* = T_1 \text{Mye}^{K_1} + T_2 \text{Mye}^{K_2}$ where $\text{Mye}^K$ is the revenue of Myerson's auction with $K$ i.i.d. bidders from $F$. For $rev_1$, we only have $rev_1 \geq 0$ since we do not any revenue guarantee for the arbitrary first-phase mechanism $\mathcal{M}$. Now consider $rev_2$, let $r^{K_2}(p)$ denote the expected revenue of a second price auction with reserve price $p$. Since the values in the two phases are independent, we have

$$rev_2 = T_2 \cdot \mathbb{E}_{v_1, \ldots, v_{T_1 K_1} \sim F} \left[ r^{K_2}(\text{ERM}^c(v_1, \ldots, v_{T_1 K_1})) \right].$$

We need to compare $r^{K_2}(\text{ERM}^c(v_1, \ldots, v_{T_1 K_1}))$ with $\text{Mye}^{K_2}$. Since bidders have i.i.d. regular value distributions, Myerson's auction is exactly the second price auction with reserve price $p = v^*$. When $K_2 = 1$, Myerson's auction becomes a post-price auction. Let $\epsilon^{sample}(\cdot)$ be the inverse function of the required number of samples for $\text{ERM}^c$ to guarantee $(1 - \epsilon^{sample})$-optimal revenue (as obtained in Huang et al. [23]) in the posted-price auction, i.e., the expected revenue of a one-bidder auction with a posted price $p$ determined by $\text{ERM}^c$ with $N$ samples is at least $(1 - \epsilon^{sample}(N))$ times the optimal expected revenue. Then for the one-bidder case, we have

$$rev_2 = T_2(1 - \epsilon^{sample}(T_1 K_1))\text{Mye}^1.$$

For general $K_2$, while the sample complexity literature does not analyze the revenue of the same reserve price $p = \text{ERM}^c(v_1, \ldots, v_{T_1 K_1})$ in a second price auction with $K_2 \geq 2$ bidders, we are able to generalize the existing revenue guarantee to the case of multiple bidders (and multiple units) under the assumption that the distribution is regular. The generalization is made by the following lemma, which we believe is of independent interest:

**Lemma C.1.** *For any regular distribution $F$, if the expected revenue of a posted price auction with price $p$ and with one bidder whose value is drawn from $F$ is $(1 - \epsilon)$-optimal, then the revenue of a Vickrey auction with reserve price $p$ selling at most $k \geq 1$ units of a item to $K \geq 2$ i.i.d. unit-demand bidders with values from $F$ is also $(1 - \epsilon)$-optimal.*

The proof is in Appendix C.3. Thus for $K_2 \geq 2$, we also have: $rev_2 = T_2(1 - \epsilon^{sample}(T_1 K_1))\text{Mye}^{K_2}$.

Finally,

$$
\begin{aligned}
1 - \epsilon_2 = \frac{rev_1 + rev_2}{rev^*} &\geq \frac{0 + T_2 \text{Mye}^{K_2} \cdot (1 - \epsilon^{sample}(T_1 K_1))}{T_1 \text{Mye}^{K_1} + T_2 \text{Mye}^{K_2}} \\
&\geq 1 - \frac{T_1 \text{Mye}^{K_1}}{T_1 \text{Mye}^{K_1} + T_2 \text{Mye}^{K_2}} - \epsilon^{sample}(T_1 K_1) \\
&\geq 1 - \frac{T_1}{T} - \epsilon^{sample}(T_1 K_1) \qquad\qquad (\text{Mye}^{K_2} \geq \text{Mye}^{K_1} \text{ since } K_2 \geq K_1)
\end{aligned}
$$

From Huang et al. [23], we know that for bounded distributions, $\epsilon^{sample}(N) = O(\sqrt{\frac{D \cdot \log N}{N}})$ when $c \leq \frac{1}{2D}$, and for MHR distributions (MHR implies regularity), $\epsilon^{sample}(N) = O([\frac{\log N}{N}]^{\frac{2}{3}})$ when $c \leq \frac{1}{4e}$. This implies the bounds on $\epsilon_2$ as stated in the theorem, and concludes the proof.

## C.3 Proof of Lemma C.1

It's more convenient to work in the quantile space. Let $v(q)$, $R(q) = qv(q)$ be the value curve and revenue curve of $F$. It's well-known that the derivative $R'(q)$ equals to the virtual value $\phi(v(q)) = v - \frac{1 - F(v)}{f(v)}$ and by Myerson's Lemma, the expected revenue with allocation rule $x(\cdot)$ equals to the virtual surplus:

$$rev = \sum_{i=1}^{K} \mathbb{E}[R'(q) x_i(q)].$$

Let $x^{K,k}(q)$ be the allocation to (the probability of winning of) a bidder whose value has quantile $q$ in a Vickrey auction selling $k$ units to $K$ bidders without reserve price. Specifically, $x^{1,1}(q) = 1$; for general $K$,

693  $x^{K,1}(q) = (1-q)^{K-1}$; for general $K, k$, $x^{K,k}(q) = \sum_{i=0}^{k-1}\binom{K-1}{i}q^i(1-q)^{K-1-i}$. With reserve price $p_0$, let
694  $q_0 = q(p_0)$, then the allocation becomes $x_{q_0}^{K,k}(q) = x^{K,k}(q)$ for $q < q_0$ and $x_{q_0}^{K,k}(q) = 0$ otherwise. So the
695  revenue of $p_0$ is

$$rev(K,k) = K\int_0^{q_0} R'(q)x^{K,k}(q)\mathrm{d}q,$$

696  and the optimal revenue is:

$$rev^*(K,k) = K\int_0^{q^*} R'(q^*)x^{K,k}(q)\mathrm{d}q,$$

697  where $q^*$ satisfies: $R'(q) \geq 0, \forall q \leq q^*$ and $R'(q) \geq 0, \forall q \geq q^*$. And define:

$$loss(K,k) = rev^*(K,k) - rev(K,k) = K\int_{q_0}^{q^*} R'(q)x^{K,k}(q)\mathrm{d}q.$$

698  Since $p_0$ is $(1-\epsilon)$-optimal with $K=1, k=1$ (the posted-price auction), we have:

$$loss(1,1) \leq \epsilon \cdot rev^*(1,1).$$

699  Now for general $K, k$:

700  • If $q_0 > q^*$. The loss:

$$loss(K,k) = K\int_{q^*}^{q_0} -R'(q)x^{K,k}(q)\mathrm{d}q$$

$$\leq K\int_{q^*}^{q_0} -R'(q)x^{K,k}(q^*)\mathrm{d}q$$

$$= Kx^{K,k}(q^*)\int_{q^*}^{q_0} -R'(q)\mathrm{d}q = Kx^{K,k}(q^*)loss(1,1),$$

701  since $x^{K,k}(q)$ is non-increasing in $q$ (actually, the monotonicity of $x^{K,k}(q)$ is the only property that is
702  used throughout the proof), and the optimal revenue:

$$rev^*(K,k) \geq K\int_0^{q^*} R'(q)x^{K,k}(q^*)\mathrm{d}q$$

$$= Kx^{K,k}(q^*)\int_0^{q^*} R'(q)\mathrm{d}q = Kx^{K,k}(q^*)rev^*(1,1),$$

703  which gives: $\frac{loss(K,k)}{rev^*(K,k)} \leq \frac{loss(1,1)}{rev^*(1,1)} \leq \epsilon$.

704  • If $q_0 < q^*$. The loss:

$$loss(K,k) = K\int_{q_0}^{q^*} R'(q)x^{K,k}(q)\mathrm{d}q \leq Kx^{K,k}(q_0)loss(1,1),$$

705  and the optimal revenue:

$$rev^*(K,k) = K\int_0^{q_0} R'(q)x^{K,k}(q)\mathrm{d}q + K\int_{q_0}^{q^*} R'(q)x^{K,k}(q)\mathrm{d}q$$

$$\geq K\int_0^{q_0} R'(q)x^{K,k}(q_0)\mathrm{d}q + K\int_{q_0}^{q^*} R'(q)x^{K,k}(q)\mathrm{d}q$$

$$= Kx^{K,k}(q_0)rev(1,1) + loss(K,k),$$

706  which gives:

$$\frac{loss(K,k)}{rev^*(K,k)} \leq \frac{loss(K,k)}{Kx^{K,k}(q_0)rev(1,1) + loss(K,k)} = \frac{1}{\frac{Kx^{K,k}(q_0)rev(1,1)}{loss(K,k)} + 1}$$

$$\leq \frac{1}{\frac{rev(1,1)}{loss(1,1)} + 1} \leq \frac{1}{\frac{1-\epsilon}{\epsilon} + 1} = \epsilon.$$

## C.4  Perfect Bayesian Equilibrium

708  Here we consider the setting of $T$-round repeated auctions where each auction contains $K \geq 1$ bidders and each
709  bidder participates in at most $\overline{m}$ rounds of auctions. We use $\boldsymbol{v}_i = (v_i^t)$ to denote bidder $i$'s profile of values,

710  where $v_i^t$ is her value at round $t$ if she participates in that round. Similarly denote by $\boldsymbol{b}_i = (b_i^t)$ the bids of
711  bidder $i$. Values are i.i.d. samples from some distribution $F$.

712  In repeated auctions, the seller can adjust the mechanism dynamically based on the bidding history of buyers,
713  and buyers can use historical information to adjust their bidding strategies. The solution concept of an $\epsilon$-perfect
714  Bayesian equilibrium ($\epsilon$-PBE) captures this dynamic nature. For each bidder $i$, we use $h_i^t$ to denote the history
715  she can observe at the start of round $t$. For example, $h_i^t$ includes her bid $b_i^{t'}$, whether she receives the item, how
716  much she pays, etc, at round $t' < t$ if she participates in round $t'$. We assume that bidder $i$ cannot observe the
717  bids in the auctions she does not participate in. We allow bidder $i$ to anticipate her values in future rounds, so she
718  can make decision on her entire value profile $\boldsymbol{v}_i = (v_i^t)$. Bidder $i$'s strategy is thus denoted by $\sigma_i = (\sigma_i^t)$ where
719  $\sigma_i^t$ maps $\boldsymbol{v}_i$ and $h_i^t$ to a bid $b_i^t = \sigma_i^t(\boldsymbol{v}_i, h_i^t)$. Let $U_i^{[t:T]}(\sigma; \boldsymbol{v}_i, h_i^t)$ be the total expected utility of bidder $i$ in
720  rounds $t, t+1, \ldots, T$, given her value profile $\boldsymbol{v}_i$, the history $h_i^t$ at round $t$, and bidders playing $\sigma = (\sigma_i, \sigma_{-i})$.

721  **Definition C.2.** *A profile of strategy $\sigma = (\sigma_i, \sigma_{-i})$ is an $\epsilon$-perfect Bayesian equilibrium ($\epsilon$-PBE) if for each*
722  *bidder $i$, each round $t$, any history $h_i^t$, any values $\boldsymbol{v}_i$, the strategy $\sigma_i$ approximately maximizes bidder $i$'s*
723  *expected utility from round $t$ to round $T$ up to $\epsilon$ error, i.e., $U_i^{[t:T]}(\sigma; \boldsymbol{v}_i, h_i^t) \geq U_i^{[t:T]}(\sigma_i', \sigma_{-i}; \boldsymbol{v}_i, h_i^t) - \epsilon$ for*
724  *any alternative strategy $\sigma_i'$.*

725  **Definition C.3.** *The seller's mechanism (or auction learning algorithm) is $\epsilon$-perfect Bayesian incentive-*
726  *compatible ($\epsilon$-PBIC) if truthful bidding (i.e., $\sigma_i^t(\boldsymbol{v}_i, h_i^t) = v_i^t$) is an $\epsilon$-PBE.*

727  We emphasize that the expected utility $U_i^{[t:T]}(\sigma; \boldsymbol{v}_i, h_i^t)$ is conditioned on $h_i^t$. This is because, the history $h_i^t$
728  which includes the allocation of item and the payment of bidder $i$ can leak information about other bidders' bids
729  (or values). Other bidders' bids will influence the mechanism the seller will use in future rounds. Thus, based on
730  this information, bidder $i$ can update her belief about other bidders' bids and the seller's choice of mechanisms
731  by Bayesian rule, then she can compute her expected utility on her updated belief.

732  **PBIC of the two-phase ERM algorithm.**  As discussed, the two-phase ERM algorithm is a learning
733  algorithm that learns approximately revenue-optimal auctions in an approximately incentive-compatible way
734  against strategic bidders in repeated auctions. The algorithm, obtained by adopting the two-phase model with
735  ERM as the price learning function and setting $K_1 = K_2 = K$, works as follows:

736  - in the first $T_1$ rounds, run any truthful, prior-independent auction $\mathcal{M}$, e.g., the second price auction
737    with no reserve;

738  - in the later $T_2 = T - T_1$ rounds, run second price auction with reserve $p = \text{ERM}^c(b_1, \ldots, b_{T_1 K})$
739    where $b_1, \ldots, b_{T_1 K}$ are the bids from the first $T_1$ auctions.

740  $T_1$ and $c$ are adjustable parameters of the two-phase ERM algorithm.

741  **Theorem C.4.** *The two-phase ERM algorithm is $\epsilon$-PBIC, where,*

742  - *for any bounded $F$, $\epsilon = \overline{m} D \Delta_{T_1 K, \overline{m} K}^{\text{worst}}$, and*

743  - *for any MHR $F$, if $\frac{\overline{m}}{T_1} \leq c \leq \frac{1}{4e}$ and $\overline{m} K = o(\sqrt{T_1 K})$, then $\epsilon = O\left(\overline{m} v^* \Delta_{T_1 K, \overline{m} K}^{\text{worst}}\right) + O\left(\frac{\overline{m} v^*}{\sqrt{T_1 K}}\right)$,*
744    *where $v^* = \arg\max_v\{v[1 - F(v)]\}$.*

745  *The constants in big O's are independent of $F$ and $c$.*

746  Combining with the bounds on $\Delta_{N,m}^{\text{worst}}$ in Theorem 1.3, we have

747  **Corollary C.5.** *The two-phase ERM algorithm is $\epsilon_1$-PBIC, where,*

748  - *for any bounded $F$,*

$$\epsilon_1 = O\left(\log^2(T_1 K) \sqrt[3]{\frac{D^{11} K \overline{m}^5}{T_1}}\right)$$

749  *if $\frac{\overline{m}}{T_1} \leq c \leq \frac{1}{2D}$ and $\overline{m} K = o(\sqrt{T_1 K})$;*

750  - *for any MHR $F$,*

$$\epsilon_1 = O\left(\log^3(T_1 K) v^* \overline{m}^2 \sqrt{\frac{K}{T_1}},\right)$$

751  *if $\frac{\overline{m}}{T_1} \leq c \leq \frac{1}{4e}$ and $\overline{m} K = o(\sqrt{T_1 K})$.*

752  *The constants in big O's are independent of $F$ and $c$.*

753  *The guarantee of $(1 - \epsilon_2)$ revenue optimality is the same as Theorem 2.2:*

- *For bounded and regular distribution, $\epsilon_2 = O\left(\frac{T_1}{T} + \sqrt{\frac{D \cdot \log(T_1 K)}{T_1 K}}\right)$.*

- *For MHR distribution, $\epsilon_2 = O\left(\frac{T_1}{T} + \left[\frac{\log(T_1 K)}{T_1 K}\right]^{\frac{2}{3}}\right)$.*

In the rest of this section we prove Theorem C.4 for bounded distributions. The proof is similar to that of Theorem 2.1 except that we need to consider the effect of history $h_i^t$ on the conditional distribution of the values of other bidders when considering perfect Bayesian equilibrium. The extension to MHR distributions is similar to the extension of Theorem 2.1 to MHR distributions (discussed in Appendix D.2) and hence omitted.

*Proof of Theorem C.4 for bounded distributions.* Assume that other bidders bid truthfully and bidder $i$ deviates from truthful bidding to other strategy, consider the increase of bidder $i$'s total expected utility from round $t$ to $T$, for each $t$, given any history $h_i^t$ and any values $\boldsymbol{v}_i$. If $t > T_1$, then the auctions in $t$ and later rounds are in the second phase and never change due to the deviation of bidder $i$, thus deviation does not increase her utility.

Then we consider $t \le T_1$. Strategic bidding does not increase bidder $i$'s utility in rounds $t, t+1, \ldots, T_1$ because these rounds are in the first phase and the mechanism in the first phase is truthful. Thus, strategic bidding can increase her utility only in the second phase. Let $t' > T_1$ be a second-phase round in which she participates. The auction at round $t'$ is a second-price auction with reserve price determined by $\mathrm{ERM}^c$ from bids in rounds 1 to $T_1$. Denote by $\boldsymbol{v}^{[1:T_1]}$ the values of all bidders in rounds 1 to $T_1$. If bidder $i$ bid truthfully, then the reserve price at round $t'$ is $p_1 = \mathrm{ERM}^c(\boldsymbol{v}^{[1:T_1]})$. Let $A \subseteq [1:T_1]$ be the set of rounds in which bidder $i$ participates from round 1 to round $T_1$. Then we can partition $\boldsymbol{v}^{[1:T_1]}$ into two parts: $\boldsymbol{v}^A$ and $\boldsymbol{v}^{[1:T_1]\backslash A}$, where $\boldsymbol{v}^A$ denotes bidders' values in the rounds in $A$, and $\boldsymbol{v}^{[1:T_1]\backslash A}$ denotes the values in the rounds not in $A$. There are $|A|K$ values in $\boldsymbol{v}^A$, $|A|$ of which are bidder $i$'s values. By deviating, bidder $i$ can change her values in $\boldsymbol{v}^A$ to some arbitrary bids. We denote by $\boldsymbol{b}^A$ the bids of bidder $i$ and the values of other bidders in $\boldsymbol{v}^A$. After deviation, the reserve price is changed to $p_2 = \mathrm{ERM}^c(\boldsymbol{b}^A, \boldsymbol{v}^{[1:T_1]\backslash A})$. By (14), the increase of bidder $i$'s utility due to the change of reserve price is at most $p_1 - p_2$, which is further upper-bounded by

$$
\begin{aligned}
p_1 - p_2 &= \mathrm{ERM}^c(\boldsymbol{v}^A, \boldsymbol{v}^{[1:T_1]\backslash A}) - \mathrm{ERM}^c(\boldsymbol{b}^A, \boldsymbol{v}^{[1:T_1]\backslash A}) \\
&\le \mathrm{ERM}^c(\boldsymbol{v}^A, \boldsymbol{v}^{[1:T_1]\backslash A}) \cdot \delta_I(\boldsymbol{v}^A, \boldsymbol{v}^{[1:T_1]\backslash A}) \\
&\le \mathrm{ERM}^c(\boldsymbol{v}^A, \boldsymbol{v}^{[1:T_1]\backslash A}) \cdot \delta_{|A|K}^{\mathrm{worst}}(\boldsymbol{v}^{[1:T_1]\backslash A}) \\
&\le D \cdot \delta_{|A|K}^{\mathrm{worst}}(\boldsymbol{v}^{[1:T_1]\backslash A}).
\end{aligned}
$$

We then argue that given any history $h_i^t$, $\boldsymbol{v}^{[1:T_1]\backslash A}$ are still i.i.d. samples from $F$, from bidder $i$'s perspective. Note that bidder $i$ does not participate in the auctions in rounds $[1:T_1]\backslash A$, and the auctions she does participate in before round $t$ is prior-independent, which implies that the allocation of item and the payments of bidders in any round depend only on the bids of bidders in that round but not on any information like bids from other rounds. Moreover, other bidders' values across different rounds are independent. Therefore, the auctions bidder $i$ participates in leaks no information about other bidders' values in rounds $[1:T_1]\backslash A$.

Therefore, the increase of bidder $i$'s expected utility at round $t'$ is at most

$$
\begin{aligned}
\mathbb{E}\left[p_1 - p_2 \mid h_i^t, \boldsymbol{v}_i\right] &\le \mathbb{E}\left[D \cdot \delta_{|A|K}^{\mathrm{worst}}(\boldsymbol{v}^{[1:T_1]\backslash A}) \mid h_i^t, \boldsymbol{v}_i\right] \\
&= \mathbb{E}_{\boldsymbol{v}^{[1:T_1]\backslash A} \sim F}\left[D \cdot \delta_{|A|K}^{\mathrm{worst}}(\boldsymbol{v}^{[1:T_1]\backslash A})\right] \\
&= D \cdot \Delta_{T_1 K, |A|K}^{\mathrm{worst}} \\
&\le D \cdot \Delta_{T_1 K, \overline{m}K}^{\mathrm{worst}},
\end{aligned}
$$

where the last inequality is because $|A| \le \overline{m}$ and $\Delta_{N, m_1}^{\mathrm{worst}} \ge \Delta_{N, m_2}^{\mathrm{worst}}$ for $m_1 \ge m_2$.

Since bidder $i$ participates in at most $\overline{m}$ auctions, the sum of increases of expected utility from round $t$ to $T$ is at most $\overline{m} D \Delta_{T_1 K, \overline{m}K}^{\mathrm{worst}}$. $\qquad\square$

# D  Analysis for MHR Distributions

Recall that a distribution $F$ is MHR if its hazard rate $\frac{f(x)}{1 - F(x)}$ is monotone non-decreasing.

## D.1  Properties of MHR Distributions

Recall that $R(q) = qv(q)$ is the revenue curve of distribution $F$, where $q(v) = 1 - F(v)$. And $q^* = \arg\max_q R(q)$ is the quantile of the optimal reserve price $v^* = \arg\max_v [1 - F(v)]v = v(q^*)$.

791 For MHR distributions, we first introduce a lemma which says that $q^*$ is bounded away from 0 by a constant.

792 **Lemma D.1** (Hartline et al. [22]). *Any MHR distribution has a unique $q^*$, and $q^* \geq \frac{1}{e}$.*

793 Moreover, the revenue curve decreases quadratically from $q^*$.

794 **Lemma D.2** (Huang et al. [23], Lemma 3.3). *For any MHR $F$, for any $0 \leq q \leq 1$, $R(q^*) - R(q) \geq$*
795 $\frac{1}{4}(q^* - q)^2 R(q^*)$.

796 The following lemma shows that samples from an MHR distribution are rarely too large.

797 **Lemma D.3.** *Let $F$ be an MHR distribution. Let $X = \max\{v_1, \ldots, v_N\}$ where $v_1, \ldots, v_N$ are $N$ i.i.d. samples*
798 *from $F$. For any $x \geq v^*$, we have $\Pr[X > x] \leq N e^{-x/v^* + 1}$.*

799 *Proof.* Note that $1 - F(x) = \exp\left\{-\int_0^x \frac{f(v)}{1 - F(v)} \mathrm{d}v\right\} \leq \exp\left\{-\int_{v^*}^x \frac{f(v)}{1 - F(v)} \mathrm{d}v\right\}$. By the definition of $v^*$ we
800 know $(v^*[1 - F(v^*)])' = 0$, or $\frac{f(v^*)}{1 - F(v^*)} = \frac{1}{v^*}$. By the definition of MHR, we have $\frac{f(x)}{1 - F(x)} \geq \frac{1}{v^*}$ for any
801 $x \geq v^*$, thus

$$1 - F(x) \leq \exp\left\{-\int_{v^*}^x \frac{1}{v^*} \mathrm{d}v\right\} = \exp\left\{-\frac{x - v^*}{v^*}\right\}.$$

802 Then the lemma follows from a simple union bound:

$$\Pr[X > x] = \Pr[\exists i, v_i > x] \leq N[1 - F(x)] \leq N \exp\left\{-\frac{x}{v^*} + 1\right\}.$$

803 $\qquad\qquad\qquad\qquad\qquad\qquad\qquad\qquad\qquad\qquad\qquad\qquad\qquad\qquad\qquad\qquad\qquad\qquad\qquad\qquad\qquad\qquad\qquad\qquad$ $\square$

804 We will use above lemmas to prove some further lemmas which characterize the behavior of $\mathrm{ERM}^c$ on samples
805 from a MHR distribution, where $c$ can be any value between $m/N$ and $1/(2e)$. The samples we consider consist
806 of $m$ copies of $+\infty$, denoted by $v_I$, and $N - m$ random draws from $F$. We sort the samples non-increasingly
807 and use

$$v_{-I} = (v_{m+1} \geq v_{m+2} \geq \cdots \geq v_N)$$

808 to denote the random draws. Let $q_{m+1} \leq q_{m+2} \leq \cdots \leq q_N$ denote their quantiles where $q_j = q(v_j)$.

809 **Lemma D.4.** *Let $F$ be an MHR distribution. Suppose $m = o(\sqrt{N})$. Fix $m$ values $v_I$ to be $+\infty$, and randomly*
810 *draw $N - m$ values $v_{-I}$ from $F$. Let $k^* = \arg\max_{i > cN}\{iv_i\}$, i.e., the index selected by $\mathrm{ERM}^c$, where*
811 $\frac{m}{N} \leq c \leq \frac{1}{2e}$. *Then we have*

$$R(q_{k^*}) \geq \left(1 - O\left(\sqrt{\frac{\log N}{N}}\right)\right) R(q^*),$$

812 *with probability at least $1 - O\left(\frac{1}{N}\right)$.*

813 *Proof.* Let $\gamma \overset{\text{def}}{=} 2\sqrt{\frac{4 \ln(2(N-m))}{N-m}} + \frac{m}{N} = O\left(\sqrt{\frac{\log N}{N}}\right)$ as in Claim B.2. We have $|q_j - \frac{j}{N}| \leq \gamma$ for any
814 $j > m$ with probability at least $1 - \frac{1}{N-m}$. We thus assume $|q_j - \frac{j}{N}| \leq \gamma$.

815 The intuition is follows: The product $jv_j$ divided by $N$ approximates $R(q_j) = q_j v_j$ up to an $O(\gamma)$ error. Our
816 proof consists of three steps: The first step is to show that with high probability, there must be some sample
817 with quantile $q_i$ that is very close to $q^*$ so its revenue $R(q_i) \approx R(q^*) \approx \frac{i}{N} v_i$. The second step is to argue
818 that all samples with quantile $q_j < \frac{1}{2e}$ are unlikely to be chosen by $\mathrm{ERM}^c$ because $q_j$ is too small and the gap
819 between $q^*$ and $\frac{1}{2e}$ leads to a large loss in revenue, roughly speaking, $\frac{j}{N} v_j \approx R(q_j) < (1 - \frac{1}{4}(\frac{1}{2e})^2) R(q^*) \approx$
820 $(1 - \Omega(1)) \frac{i}{N} v_i$. The final step is to show that if a quantile $q_j > \frac{1}{2e}$ is to be chosen by $\mathrm{ERM}^c$, then it must have
821 equally good revenue as $q_i$.

822 Formally:

823     1. Firstly, consider the quantile interval $[q^* - \gamma, q^*]$. Each random draw $q_i$, if falling into this interval,
824         will satisfy:

$$\frac{i}{N} v_i \geq (q_i - \gamma) v_i \geq (q^* - 2\gamma) v_i \geq (q^* - 2\gamma) v^* \geq (1 - 2e\gamma) q^* v^*, \qquad (15)$$

825         where the last but one inequality is because $q_i \leq q^*$ and the last one follows from $q^* \geq \frac{1}{e}$. The
826         probability that no quantile falls into $[q^* - \gamma, q^*]$ is at most

$$(1 - \gamma)^{N-m} = \left(1 - O\left(\sqrt{\frac{\log N}{N}}\right)\right)^{N-m} = o(\frac{1}{N}).$$

2. For the second step, first note that the $q_i \in [q^* - \gamma, q^*]$ in the first step will be considered by $\mathrm{ERM}^c$ since $i \geq (q_i - \gamma)N \geq (q^* - 2\gamma)N \geq (\frac{1}{e} - 2\gamma)N > cN$. Then suppose $\mathrm{ERM}^c$ chooses another quantile $q_j$ instead of $q_i$, we must have

$$\frac{j}{N}v_j \geq \frac{i}{N}v_i. \tag{16}$$

We will show that such probability is small if $q_j < \frac{1}{2e} + \gamma$. Pick a threshold quantile $\frac{1}{T}$ where $T = N^{1/4}$. Consider two cases:

- If $0 \leq q_j < \frac{1}{T}$. We argue that $\mathrm{ERM}^c$ picks $q_j$ with probability at most $o(\frac{1}{N})$. Note that

$$\frac{j}{N}v_j \leq (q_j + \gamma)v_j \leq (\frac{1}{T} + \gamma)v_j, \tag{17}$$

together with (16) and (15), we obtain $(\frac{1}{T} + \gamma)v_j \geq (1 - 2e\gamma)q^*v^*$, implying

$$v_j \geq \frac{1 - 2e\gamma}{e}\frac{Tv^*}{1 + T\gamma} = \Omega(Tv^*),$$

since $T\gamma = O\left(\frac{\sqrt{\log N}}{N^{1/4}}\right) \to 0$. According to Lemma D.3, the probability that there exists $v_j > \Omega(Tv^*)$ is at most

$$N \exp\left\{-\frac{\Omega(Tv^*)}{v^*} + 1\right\} = o(\frac{1}{N}).$$

- If $\frac{1}{T} \leq q_j < \frac{1}{2e} + \gamma$. We argue that $\mathrm{ERM}^c$ will never choose such $q_j$. Note that

$$\frac{j}{N}v_j \leq (q_j + \gamma)v_j \leq (1 + T\gamma)q_jv_j, \tag{18}$$

together with (16) and (15), we obtain $(1 + T\gamma)q_jv_j > (1 - 2e\gamma)q^*v^*$. Then by Lemma D.2,

$$\frac{1 - 2e\gamma}{1 + T\gamma} \leq \frac{q_jv_j}{q^*v^*} \leq 1 - \frac{1}{4}(q_j - q^*)^2 \leq 1 - \frac{1}{4}(\frac{1}{2e} - \gamma)^2. \tag{19}$$

However, the left hand side of (19) approaches to 1 since $\gamma$ and $T\gamma$ approach 0 while the right hand side is strictly less than 1, a contradiction. So this case never happens.

3. Finally, if $q_j \geq \frac{1}{2e} + \gamma$. We argue that if $\mathrm{ERM}^c$ picks $q_j$ instead of $q_i$, then $R(q_j)$ approximates $R(q_i)$ well, satisfying the conclusion in the lemma. This is because

$$R(q_j) = q_jv_j \geq (\frac{j}{N} - \gamma)v_j$$

$$\geq (1 - 2e\gamma)\frac{j}{N}v_j \qquad\qquad \frac{j}{N} \geq q_j - \gamma \geq \frac{1}{2e}$$

$$\geq (1 - 2e\gamma)\frac{i}{N}v_i \qquad\qquad \text{Eq. (16)}$$

$$\geq (1 - 2e\gamma)(1 - 2e\gamma)q^*v^* \qquad\qquad \text{Eq. (15)}$$

$$= (1 - O(\gamma))R(q^*).$$

Combining above three steps and the event in the beginning of the proof, we have $R(q_{k^*}) \geq (1 - O(\sqrt{\frac{\log N}{N}}))R(q^*)$ except with probability at most

$$\frac{1}{N - m} + o(\frac{1}{N}) + o(\frac{1}{N}) = O(\frac{1}{N}).$$

$\square$

**Lemma D.5.** *Let $F$ be an MHR distribution. Suppose $m = o(\sqrt{N})$. Fix $m$ values $v_I$ to be $+\infty$, and randomly draw $N - m$ values $v_{-I}$ from $F$. Let $k^* = \arg\max_{i>cN}\{iv_i\}$, i.e., the index selected by $\mathrm{ERM}^c$, where $\frac{m}{N} \leq c \leq \frac{1}{2e}$. Let $\epsilon = \sqrt[4]{\frac{\log N}{N}}$. Then with probability at least $1 - O\left(\frac{1}{N}\right)$, the following inequalities hold:*

1. $q_{k^*} \geq q^* - O(\epsilon)$;

2. $k^* \geq [q^* - O(\epsilon)]N > \frac{1}{2e}N$;

3. $v_{k^*} \leq [1 + O(\epsilon)]v^*$.

*Proof.* For inequality (1), by Lemma D.2 and Lemma D.4, with probability at least $1 - O(\frac{1}{N})$, we have

$$\frac{1}{4}(q_{k^*} - q^*)^2 \leq \frac{R(q^*) - R(q_{k^*})}{R(q^*)} \leq O\left(\sqrt{\frac{\log N}{N}}\right).$$

Taking the square root, we obtain $q_{k^*} \geq q^* - O\left(\sqrt[4]{\frac{\log N}{N}}\right)$.

Assume that (1) holds. To prove (2), note that by Claim B.2, we have $\frac{k^*}{N} \geq q_{k^*} - O\left(\sqrt{\frac{\log N}{N}}\right) \geq q^* - O(\epsilon)$

except with probability at most $O(\frac{1}{N})$, and $q^* > \frac{1}{e}$.

Finally, inequality (3) follows from

$$\frac{v_{k^*}}{v^*} = \frac{R(q_{k^*})}{q_{k^*}} \frac{q^*}{R(q^*)} \leq 1 \cdot \frac{q^*}{q_{k^*}} \leq \frac{q^*}{q^* - O(\epsilon)} = 1 + \frac{O(\epsilon)}{q^* - O(\epsilon)} \leq 1 + O(e\epsilon).$$

$\square$

## D.2 Detailed Proof of Theorem 2.1 for MHR Distributions

Let $\Delta U(\boldsymbol{v}_i, b_I, v_{-I}) = U_i^{\mathrm{TP}}(\boldsymbol{v}_i, b_I, v_{-I}) - U_i^{\mathrm{TP}}(\boldsymbol{v}_i, v_I, v_{-I})$. Similar to the proof for bounded distributions, we have for any $\boldsymbol{v}_i, b_I, v_{-I}$,

$$\Delta U(\boldsymbol{v}_i, b_I, v_{-I}) \leq m_2 \cdot \left(\mathrm{ERM}^c(v_I, v_{-I}) - \mathrm{ERM}^c(b_I, v_{-I})\right) \leq m_2 \cdot \mathrm{ERM}^c(v_I, v_{-I}) \cdot \delta_{m_1}^{\mathrm{worst}}(v_{-I}).$$

By Claim A.2, we have $\mathrm{ERM}^c(v_I, v_{-I}) \leq \mathrm{ERM}^c(\overline{v}_I, v_{-I})$ where $\overline{v}_I$ can be any $m_1$ values (e.g., $+\infty$) that are greater than the maximal value in $v_{-I}$, when $c \geq \frac{m_1}{T_1 K_1}$.

Let $N = T_1 K_1$, define two threshold prices $T_1 = \sqrt{N}v^*$ and $T_2 = [1 + O(\epsilon)]v^*$ where $\epsilon = \sqrt[4]{\frac{\log N}{N}}$ as in Lemma D.5. Note that for sufficiently large $N$, $T_1 > T_2$. With the random draw of $v_{-I}$ from $F$, denote the random variable $\mathrm{ERM}^c(\overline{v}_I, v_{-I})$ by $P$, we have:

$$\begin{aligned}
\mathbb{E}_{\boldsymbol{v}_{-i}}\left[\Delta U(\boldsymbol{v}_i, b_I, v_{-I})\right] &= \mathbb{E}_{v_{-I}}\left[\Delta U(\boldsymbol{v}_i, b_I, v_{-I}) \mid P \leq T_2\right] \cdot \Pr[P \leq T_2] \\
&\quad + \mathbb{E}_{v_{-I}}\left[\Delta U(\boldsymbol{v}_i, b_I, v_{-I}) \mid T_2 < P \leq T_1\right] \cdot \Pr[T_2 < P \leq T_1] \\
&\quad + \mathbb{E}_{v_{-I}}\left[\Delta U(\boldsymbol{v}_i, b_I, v_{-I}) \mid P > T_1\right] \cdot \Pr[P > T_1] \\
&\overset{\text{def}}{=} \mathbb{E}_1 + \mathbb{E}_2 + \mathbb{E}_3.
\end{aligned} \tag{20}$$

1. For the first term $\mathbb{E}_1$,

$$\begin{aligned}
\mathbb{E}_1 &= \mathbb{E}_{v_{-I}}\left[\Delta U(\boldsymbol{v}_i, b_I, v_{-I}) \mid P \leq T_2\right] \cdot \Pr[P \leq T_2] \\
&\leq \mathbb{E}_{v_{-I}}\left[m_2 \cdot P \cdot \delta_{m_1}^{\mathrm{worst}}(v_{-I}) \mid P \leq T_2\right] \cdot \Pr[P \leq T_2] \\
&\leq m_2 \cdot T_2 \cdot \mathbb{E}_{v_{-I}}\left[\delta_{m_1}^{\mathrm{worst}}(v_{-I}) \mid P \leq T_2\right] \cdot \Pr[P \leq T_2] \\
&\leq m_2 \cdot [1 + O(\epsilon)]v^* \cdot \mathbb{E}_{v_{-I}}\left[\delta_{m_1}^{\mathrm{worst}}(v_{-I})\right] \\
&= O\left(m_2 \cdot v^* \cdot \Delta_{N,m_1}^{\mathrm{worst}}\right).
\end{aligned}$$

2. For the second term, we claim that $\mathbb{E}_2 = O(\frac{m_2 v^*}{\sqrt{N}})$.

By Lemma D.5, we have $\Pr[P > [1 + O(\epsilon)]v^*] \leq O(\frac{1}{N})$. Therefore,

$$\begin{aligned}
\mathbb{E}_2 &= \mathbb{E}_{v_{-I}}\left[\Delta U(\boldsymbol{v}_i, b_I, v_{-I}) \mid T_2 < P \leq T_1\right] \cdot \Pr[T_2 < P \leq T_1] \\
&\leq \mathbb{E}_{v_{-I}}\left[m_2 \cdot P \cdot 1 \mid T_2 < P \leq T_1\right] \cdot \Pr[T_2 < P \leq T_1] \\
&\leq m_2 \cdot T_1 \cdot \Pr[P > T_2] \\
&\leq m_2 \cdot \sqrt{N}v^* \cdot O\left(\frac{1}{N}\right) \\
&= O\left(\frac{m_2 v^*}{\sqrt{N}}\right).
\end{aligned}$$

3. For the third term, we claim that $\mathbb{E}_3 = o(\frac{m_2 v^*}{N})$.

Let $B$ be the upper bound on the support of $F$ ($B$ can be $+\infty$). Let $F_P(x)$ be the distribution of $P$. For convenience, suppose it is continuous and has density $f_P(x)$. We have:

$$\begin{aligned}
\mathbb{E}_3 &= \mathbb{E}_{v_{-I}}\left[\Delta U(\boldsymbol{v}_i, b_I, v_{-I}) \mid P > T_1\right] \cdot \Pr[P > T_1] \\
&\leq \mathbb{E}_{v_{-I}}\left[m_2 \cdot P \cdot 1 \mid P > T_1\right] \cdot \Pr[P > T_1] \\
&= m_2 \cdot \mathbb{E}_{v_{-I}}\left[P \mid P > T_1\right] \cdot \Pr[P > T_1] \\
&= m_2 \cdot \int_{T_1}^{B} x f_P(x) \mathrm{d}x \\
&= m_2 \cdot \left(\int_{T_1}^{B} [1 - F_P(x)]\mathrm{d}x + T_1[1 - F_P(T_1)]\right).
\end{aligned}$$

Let $\max\{v_{-I}\}$ denote the maximum value in the $N - m_1$ samples $v_{-I}$. By Lemma D.3, we have for any $x \geq v^*$,

$$1 - F_P(x) = \Pr[P > x] \leq \Pr[\max\{v_{-I}\} > x] \leq N e^{-\frac{x}{v^*} + 1}.$$

Thus,

$$\begin{aligned}
\int_{T_1}^{B} [1 - F_P(x)]\mathrm{d}x + T_1[1 - F_P(T_1)] &\leq v^* N e^{-\frac{T_1}{v^*} + 1} + T_1 N e^{-\frac{T_1}{v^*} + 1} \\
&= v^* N(1 + \sqrt{N})e^{-\sqrt{N} + 1} \\
&= o(\frac{v^*}{N}),
\end{aligned}$$

as desired.

Combining the three items,

$$\mathbb{E}_{\boldsymbol{v}_{-i}}\left[\Delta U(\boldsymbol{v}_i, b_I, v_{-I})\right] = O\left(m_2 v^* \Delta_{N,m_1}^{\text{worst}}\right) + O\left(\frac{m_2 v^*}{\sqrt{N}}\right).$$

## D.3 An Improved Bound on Incentive-Awareness Measure for MHR Distributions

Here we improve the upper bound on $\Delta_{N,m}^{\text{worst}}$ for MHR distributions by proving:

**Lemma D.6** (Tigher bound for MHR distributions). *Moreover, if $F$ is MHR, let $d = \frac{1}{2e}$, and suppose $\frac{m}{N} \leq c \leq \frac{1}{4e}$, we have*

$$\Delta_{N,m}^{\text{worst}} \leq O\left(\frac{m}{d^{7/2}} \frac{\log^3 N}{\sqrt{N}}\right) + \Pr[\mathrm{E}].$$

The main idea is to limit the range of the quantile $q_j$ of the "bad value" $v_j$ in $\text{Bad}(\eta_t, \theta_t)$ in Lemma B.5. Recall that in the proof of Lemma B.5 we assume $q_j$ can take any value in $[0, 1]$, divide $[0, 1]$ into $O(1/h)$ intervals (as in (11)), and take a union bound to upper-bound the probability that a bad $q_j$ exists. For MHR distributions, however, we will show that $q_j v_j$ is a $(1 - O(\sqrt{\frac{\log N}{N}}))$ approximation to $R(q^*)$, thus we can use Lemma D.2 to reduce the possible range of $q_j$ from 1 to $O(\sqrt[4]{\frac{\log N}{N}})$.

### D.3.1 Proof of Lemma D.6

We repeat the argument for Lemma B.1 until Claim B.2, before which we have:

$$\Delta_{N,m}^{\text{worst}} \leq \int_0^1 \Pr[\delta_I(\overline{v}_I, v_{-I}) > \eta \wedge \overline{\mathrm{E}}]\mathrm{d}\eta + \Pr[\mathrm{E}]. \tag{21}$$

Let $\gamma = O(\sqrt{\frac{\log N}{N}})$ be the upper bound on $|q_j - \frac{j}{N}|$ in Conc. With the random draw of $N - m$ samples $v_{-I}$ from $F$ (and assume other $m$ samples $\overline{v}_I$ are equal to $\max\{v_{-I}\}$), we have $|q_j - \frac{j}{N}| < \gamma$ for any $j > m$ with probability at least $1 - \frac{1}{N-m}$. Moreover, by Lemma D.4 and Lemma D.5, with probability at least $1 - O(\frac{1}{N})$ we have $R(q_{k^*}) = q_{k^*} v_{k^*} \geq (1 - \epsilon)R(q^*)$ and $q_{k^*} \geq q^* - O(\sqrt{\epsilon}) > \frac{1}{2e}$, where $\epsilon = O(\sqrt{\frac{\log N}{N}})$. Combine the above two inequalities with Conc and denote the combined event by Conc', i.e.,

$$\text{Conc}' \overset{\text{def}}{=} \text{Conc} \wedge [R(q_{k^*}) \geq (1 - \epsilon)R(q^*)] \wedge \left[q_{k^*} \geq q^* - O(\sqrt{\epsilon}) > \frac{1}{2e}\right].$$

892    We have $\Pr[\overline{\text{Conc}'}] \le O(\frac{1}{N})$. Re-define $G(\eta) = \Pr[\delta_I(\overline{v}_I, v_{-I}) > \eta \wedge \overline{\text{E}} \wedge \text{Conc}']$, and re-write (4):

$$\Pr[\delta_I(\overline{v}_I, v_{-I}) > \eta \wedge \overline{\text{E}}] \le G(\eta) + O\left(\frac{1}{N}\right). \tag{22}$$

893    The following steps of bounding $G(\eta) = \Pr[\delta_I(\overline{v}_I, v_{-I}) > \eta \wedge \overline{\text{E}} \wedge \text{Conc}']$ are the same as before (in
894    particular, Lemma B.4 in Lemma B.3), until upper-bounding $\Pr[\text{Bad}(\eta, \theta) \wedge \overline{\text{E}} \wedge \text{Conc}']$ (Lemma B.5), where
895    we improve the bound by a factor of $\sqrt[4]{\frac{\log N}{N}}$.

896    **Lemma D.7** (Improved Lemma B.5 for MHR distributions). *Let $d = \frac{1}{2e}$. If $\eta$ and $\theta$ are at least*
897    $\Omega\left(\frac{m}{d}\sqrt{\frac{\log(N-m)}{N-m}}\right)$, *then* $\Pr[\text{Bad}(\eta, \theta) \wedge \overline{\text{E}} \wedge \text{Conc}'] = O\left(\sqrt[4]{\frac{\log N}{N}} \frac{m \log^2 N}{d^4 \theta \sqrt{\eta^3 N}}\right)$.

898    *Proof.* The proof is the same as that of Lemma B.5 (in Appendix B.5), except that before dividing the quantile
899    space $[0, 1]$, we argue that the space to be divided can be shortened to $[q^* - O(\sqrt{\epsilon}), q^* + O(\sqrt{\epsilon})]$.

900    Consider the index $j$ that is promised to exist in $\text{Bad}(\eta, \theta)$,

$$R(q_j) = q_j v_j$$

$$\begin{aligned}
&\ge (\frac{j}{N} - \gamma)v_j &&\text{Conc}' \\
&\ge \frac{k^*}{N}v_{k^*} - \frac{m}{N\theta}v_{k^*} - \gamma v_{k^*} &&jv_j \ge k^* v_{k^*} - \frac{m}{\theta}v_{k^*} \text{ and } v_{j^*} \le v_{k^*} \\
&\ge (q_{k^*} - \gamma)v_{k^*} - \frac{m}{N\theta}v_{k^*} - \gamma v_{k^*} &&\text{Conc}' \\
&= \left[q_{k^*} - (2\gamma + \frac{m}{N\theta})\right]v_{k^*} \\
&= \left[1 - \frac{2\gamma + \frac{m}{N\theta}}{q_{k^*}}\right]R(q_k^*) \\
&\ge \left[1 - 2e(2\gamma + \frac{m}{N\theta})\right](1 - \epsilon)R(q^*) &&q_{k^*} \ge \frac{1}{2e} \text{ and } R(q_{k^*}) \ge (1-\epsilon)R(q^*) \text{ in Conc}' \\
&= \left[1 - O\left(\sqrt{\frac{\log N}{N}}\right)\right]R(q^*) &&\text{Definition of } \gamma \text{ and } \epsilon, \text{ and } \theta = \Omega\left(m\sqrt{\frac{\log N}{N}}\right).
\end{aligned}$$

901    By Lemma D.2, we have

$$|q_j - q^*| \le 2\sqrt{O\left(\sqrt{\frac{\log N}{N}}\right)} = O\left(\sqrt[4]{\frac{\log N}{N}}\right).$$

902    Now we modify the analysis after Claim B.8. Consider those intervals $I_l$'s with length $h$ in (11) that intersect
903    with

$$I_{q_j} = \left[q^* - O\left(\sqrt[4]{\frac{\log N}{N}}\right), q^* + O\left(\sqrt[4]{\frac{\log N}{N}}\right)\right].$$

904    There are at most $O\left(\frac{1}{h}\sqrt[4]{\frac{\log N}{N}}\right)$ such intervals and we denote the set of (indices of) those intervals by $\mathcal{L}$. The
905    definitions of $i_l^*$, $i_{<(l+1)}^*$, $A_l$, $A_{<(l+1)}$, and $W_l$ remain unchanged. By choosing the index $l$ such that $q_j \in I_{l+2}$,
906    we know that if the event $[\text{Bad}(\eta, \theta) \wedge \overline{\text{E}} \wedge \text{Conc}']$ holds then $W_l$ must hold for some $l$ such that $l + 2 \in \mathcal{L}$.
907    By Lemma B.9 we have $\Pr[W_l] \le O(\frac{H \log^2 N}{\sqrt{hd^3 N}})$. Taking a union bound over $l$, we obtain

$$\Pr[\text{Bad}(\eta, \theta) \wedge \overline{\text{E}} \wedge \text{Conc}'] \le O\left(\frac{1}{h}\sqrt[4]{\frac{\log N}{N}} \cdot \frac{H \log^2 N}{\sqrt{hd^3 N}}\right) = O\left(\sqrt[4]{\frac{\log N}{N}} \frac{m \log^2 N}{d\theta \sqrt{(d\eta)^3 d^3 N}}\right),$$

908    where the last equality is because $H = O(\frac{m}{d\theta})$ and $h = \Omega(d\eta)$ under the assumption that $\eta$ and $\theta$ are at least
909    $\Omega(\frac{m}{d}\sqrt{\frac{\log(N-m)}{N-m}})$.                       □

910    Then we improve Lemma B.3 based on Lemma D.7.

911    **Lemma D.8** (Improved Lemma B.3). *If $\eta$ is at least $\Omega\left(\frac{m}{d}\sqrt{\frac{\log(N-m)}{N-m}}\right)$, then* $G(\eta) = O\left(\frac{m \log^3 N}{d^4 N^{3/4}} \frac{1}{\eta^{3/2}}\right)$.

*Proof.* Modify the end of Appendix B.3,

$$\Pr[\delta_I(\overline{v}_I, v_{-I}) > \eta \wedge \overline{E} \wedge \mathrm{Conc}'] \leq \sum_{t=0}^{M} \Pr[\mathrm{Bad}(\eta_t, \theta_t) \wedge \overline{E} \wedge \mathrm{Conc}'] \qquad \text{Lemma B.4}$$

$$= \sum_{t=0}^{M} O\left( \sqrt[4]{\frac{\log N}{N}} \frac{m \log^2 N}{d^4 \theta_t \sqrt{\eta_t^3 N}} \right) \qquad \text{Lemma D.7}$$

$$= \sum_{t=0}^{M} O\left( \sqrt[4]{\frac{\log N}{N}} \frac{m \log^2 N}{d^4 \sqrt{N}} \frac{\eta_{t+1}}{\eta \sqrt{\eta_t^3}} \right) \qquad \text{Definition of } \theta_t$$

$$= O\left( \sqrt[4]{\frac{\log N}{N}} \frac{m \log^2 N}{d^4 \sqrt{N}} \cdot \sum_{t=0}^{M} \frac{\eta_{t+1}}{\eta} \frac{1}{\eta_t^{3/2}} \right)$$

Note that because $\eta_t, \theta_t \geq \frac{\eta}{2}$, the condition of Lemma D.7 is satisfied when $\eta = \Omega(\frac{m}{d}\sqrt{\frac{\log(N-m)}{N-m}})$.

By Claim B.6, we can choose a sequence of $\{\eta_t\}$ such that

$$\sum_{t=0}^{M} \frac{\eta_{t+1}}{\eta} \frac{1}{\eta_t^{3/2}} = O\left( \frac{\log\log(N-m)}{\eta^{3/2}} \right).$$

Therefore,

$$\Pr[\delta_I(\overline{v}_I, v_{-I}) > \eta \wedge \overline{E} \wedge \mathrm{Conc}'] \leq O\left( \frac{m \log^{2+1/4} N}{d^4 N^{1/2+1/4}} \cdot \frac{\log\log(N-m))}{\eta^{3/2}} \right) = O\left( \frac{m \log^3 N}{d^4 N^{3/4}} \frac{1}{\eta^{3/2}} \right).$$

$\square$

We finish the proof of Lemma D.6 by computing the integral in (21). Let $C = \Theta\left( \frac{m \log^3 N}{d^4 N^{3/4}} \right)$ be the bound on $G(\eta)$ in Lemma D.8, and let $A = \Theta(\frac{m}{d}\sqrt{\frac{\log(N-m)}{N-m}}) = \Theta(\frac{m}{d}\sqrt{\frac{\log N}{N}})$ be the condition on the lower bound on $\eta$ in Lemma D.8. Then $G(\eta) < \frac{C}{\eta^{3/2}}$ when $\eta > \max\{C^{2/3}, A\}$. If $C^{2/3} > A$, then we have

$$\int_0^1 \Pr[\delta_I(\overline{v}_I, v_{-I}) > \eta \wedge \overline{E}] \mathrm{d}\eta \leq \int_0^1 \left( G(x) + O\left( \frac{1}{N} \right) \right) \mathrm{d}x \qquad \text{by (22)}$$

$$\leq \int_0^{C^{\frac{2}{3}}} 1 \mathrm{d}x + \int_{C^{\frac{2}{3}}}^1 \frac{C}{x^{\frac{3}{2}}} \mathrm{d}x + O\left( \frac{1}{N} \right)$$

$$= C^{\frac{2}{3}} + \frac{C}{-\frac{1}{2}} - \frac{C}{-\frac{1}{2}} C^{-\frac{1}{3}} + O\left( \frac{1}{N} \right)$$

$$\leq 3C^{\frac{2}{3}} + O\left( \frac{1}{N} \right)$$

$$= O\left( \frac{m^{2/3}}{d^{8/3}} \frac{\log^2 N}{\sqrt{N}} \right) + O\left( \frac{1}{N} \right)$$

$$= O\left( \frac{m^{2/3}}{d^{8/3}} \frac{\log^2 N}{\sqrt{N}} \right).$$

If $A > C^{2/3}$, then we have:

$$\int_0^1 \Pr[\delta_I(\overline{v}_I, v_{-I}) > \eta \wedge \overline{E}] \mathrm{d}\eta \leq \int_0^1 \left( G(x) + O\left( \frac{1}{N} \right) \right) \mathrm{d}x \qquad \text{by (22)}$$

$$\leq \int_0^A 1 \mathrm{d}x + \int_A^1 \frac{C}{x^{\frac{3}{2}}} \mathrm{d}x + O\left( \frac{1}{N} \right)$$

$$= A + \frac{C}{-\frac{1}{2}} - \frac{C}{-\frac{1}{2}} A^{-\frac{1}{2}} + O\left( \frac{1}{N} \right)$$

$$\leq \Theta\left( \frac{m}{d}\sqrt{\frac{\log N}{N}} \right) + \Theta\left( \frac{m^{1-1/2} \log^{3-1/4} N}{d^{4-1/2} N^{3/4-1/4}} \right) + O\left( \frac{1}{N} \right)$$

$$= O\left( \frac{m}{d^{7/2}} \frac{\log^3 N}{\sqrt{N}} \right),$$

which, together with (21), concludes the proof of Lemma D.6.

## E  Analysis for $\alpha$-Strongly Regular Distributions

### E.1  Useful Lemmas

**Lemma E.1** (Cole and Roughgarden [11]). *Any $\alpha$-strongly regular distribution has a unique $q^*$, and $q^* \geq \alpha^{\frac{1}{1-\alpha}}$.*

**Lemma E.2** (Huang et al. [23], Lemma 3.5). *For any $\alpha$-strongly regular distribution $F$, for any $0 \leq q \leq 1$, $R(q^*) - R(q) \geq \frac{\alpha}{3}(q^* - q)^2 R(q^*)$.*

**Lemma E.3.** *Let $F$ be an $\alpha$-strongly regular distribution. Let $X = \max\{v_1, \ldots, v_N\}$ where $v_1, \ldots, v_N$ are $N$ i.i.d. samples from $F$. For any $x \geq v^*$, we have $\Pr[X > x] \leq N\left(\frac{v^*}{(1-\alpha)x + \alpha v^*}\right)^{\frac{1}{1-\alpha}}$.*

*Proof.* Note that $1 - F(x) = \exp\left\{-\int_0^x \frac{f(v)}{1-F(v)}dv\right\} \leq \exp\left\{-\int_{v^*}^x \frac{f(v)}{1-F(v)}dv\right\}$. By the definition of $v^*$ we know $(v^*[1 - F(v^*)])' = 0$, or $\frac{f(v^*)}{1-F(v^*)} = \frac{1}{v^*}$. By the definition of $\alpha$-strong regularity, we have

$$\left(\frac{1-F(x)}{f(x)}\right)' = 1 - \frac{d\phi}{dx} \leq 1 - \alpha$$

and

$$\frac{1-F(x)}{f(x)} \leq \frac{1-F(v^*)}{f(v^*)} + (1-\alpha)(x - v^*).$$

Thus

$$\int_{v^*}^x \frac{f(v)}{1-F(v)} \geq \int_{v^*}^x \frac{1}{\frac{1-F(v^*)}{f(v^*)} + (1-\alpha)(v - v^*)}dv = \frac{1}{1-\alpha}\left[\ln\left(v^* + (1-\alpha)(x - v^*)\right) - \ln v^*\right]$$

and

$$1 - F(x) \leq \exp\left\{-\frac{1}{1-\alpha}\ln\frac{v^* + (1-\alpha)(x - v^*)}{v^*}\right\} = \left(\frac{v^*}{(1-\alpha)x + \alpha v^*}\right)^{\frac{1}{1-\alpha}}.$$

Then the lemma follows from a simple union bound:

$$\Pr[X > x] = \Pr[\exists i, v_i > x] \leq N[1 - F(x)] \leq N\left(\frac{v^*}{(1-\alpha)x + \alpha v^*}\right)^{\frac{1}{1-\alpha}}.$$

$\square$

**Claim E.4.** *(Improved Claim B.2) Define event* Conc*:*

$$\text{Conc} = \left[\forall j > m, \left|q_j - \frac{j}{N}\right| \leq 2\sqrt{\frac{3}{\alpha}\frac{\ln(2(N-m))}{N-m}} + \frac{m}{N}\right],$$

*then $\Pr[\overline{\text{Conc}}] \leq \frac{1}{(N-m)^{\frac{3-2\alpha}{2\alpha}}}$, where the probability is over the random draw of the $N - m$ samples $v_{-I}$.*

*Proof.* Set $\delta = \frac{1}{(N-m)^{\frac{3-2\alpha}{2\alpha}}}$ in Lemma A.3. $\square$

**Lemma E.5.** *Let $F$ be an $\alpha$-strongly regular distribution. Suppose $m = o(\sqrt{N})$. Fix $m$ values $v_I$ to be $+\infty$, and randomly draw $N - m$ values $v_{-I}$ from $F$. Let $k^* = \arg\max_{i>cN}\{iv_i\}$, i.e., the index selected by $\mathrm{ERM}^c$, where $\left(\frac{\log N}{N}\right)^{\frac{1}{3}} \leq c \leq \frac{\alpha^{1/(1-\alpha)}}{2}$. Then we have*

$$R(q_{k^*}) \geq \left(1 - O\left(\sqrt{\frac{\log N}{N}}\right)\right) R(q^*),$$

*with probability at least $1 - O\left(\frac{1}{N^{\frac{3-2\alpha}{2\alpha}}}\right)$. The constants in the big O's depend on $\alpha$.*

*Proof.* Let $\gamma \overset{\text{def}}{=} 2\sqrt{\frac{3}{\alpha}\frac{\log(2(N-m))}{N-m}} + \frac{m}{N} = O\left(\sqrt{\frac{\log N}{N}}\right)$ as in Claim E.4. We have $\left|q_j - \frac{j}{N}\right| \leq \gamma$ for any $j > m$ with probability at least $1 - \frac{1}{(N-m)^{\frac{3-2\alpha}{2\alpha}}}$. We thus assume $\left|q_j - \frac{j}{N}\right| \leq \gamma$. For simplicity, we define $e(\alpha) = \alpha^{1/(1-\alpha)}$, and Lemma E.1 implies $q^* \geq e(\alpha)$.

The intuition is follows: The product $jv_j$ divided by $N$ approximates $R(q_j) = q_j v_j$ up to an $O(\gamma)$ error. Our proof consists of three steps: The first step is to show that with high probability, there must be some sample with quantile $q_i$ that is very close to $q^*$ so its revenue $R(q_i) \approx R(q^*) \approx \frac{i}{N} v_i$. The second step is to argue that all samples with quantile $q_j < \frac{e(\alpha)}{2}$ are unlikely to be chosen by $\mathrm{ERM}^c$ because $q_j$ is too small and the gap between $q^*$ and $\frac{e(\alpha)}{2}$ leads to a large loss in revenue, roughly speaking, $\frac{j}{N} v_j \approx R(q_j) < (1 - \frac{\alpha}{3}(\frac{e(\alpha)}{2})^2) R(q^*) \approx (1 - \Omega(1)) \frac{i}{N} v_i$. The final step is to show that if a quantile $q_j > \frac{e(\alpha)}{2}$ is to be chosen by $\mathrm{ERM}^c$, then it must have equally good revenue as $q_i$.

Formally:

1. Firstly, consider the quantile interval $[q^* - \gamma, q^*]$. Each random draw $q_i$, if falling into this interval, will satisfy:

$$\frac{i}{N} v_i \geq (q_i - \gamma) v_i \geq (q^* - 2\gamma) v_i \geq (q^* - 2\gamma) v^* \geq (1 - 2\gamma/e(\alpha)) q^* v^*, \qquad (23)$$

where the last but one inequality is because $q_i \leq q^*$ and the last one follows from $q^* \geq e(\alpha)$. The probability that no quantile falls into $[q^* - \gamma, q^*]$ is at most

$$(1 - \gamma)^{N-m} = \left( 1 - O\left( \sqrt{\frac{\log N}{N}} \right) \right)^{N-m} = o(\frac{1}{N^{\frac{3-2\alpha}{2\alpha}}}).$$

2. For the second step, first note that the $q_i \in [q^* - \gamma, q^*]$ in the first step will be considered by $\mathrm{ERM}^c$ since $i \geq (q_i - \gamma) N \geq (q^* - 2\gamma) N \geq (e(\alpha) - 2\gamma) N > cN$. Then suppose $\mathrm{ERM}^c$ chooses another quantile $q_j$ instead of $q_i$, we must have

$$\frac{j}{N} v_j \geq \frac{i}{N} v_i. \qquad (24)$$

We will show that such $q_j$ does not exist.

Suppose $\mathrm{ERM}^c$ chooses $q_j$, then $j$ must satisfy $j/N > c > \left( \frac{\log N}{N} \right)^{\frac{1}{3}}$, and as a result, $q_j > \left( \frac{\log N}{N} \right)^{\frac{1}{3}} - \gamma$.

If $\left( \frac{\log N}{N} \right)^{\frac{1}{3}} - \gamma < q_j < \frac{e(\alpha)}{2} + \gamma$, note that

$$\frac{j}{N} v_j \leq (q_j + \gamma) v_j \leq \left( 1 + \frac{\gamma}{\left( \frac{\log N}{N} \right)^{\frac{1}{3}} - \gamma} \right) q_j v_j, \qquad (25)$$

together with (24) and (23), we obtain $\left( 1 + \frac{\gamma}{\left( \frac{\log N}{N} \right)^{\frac{1}{3}} - \gamma} \right) q_j v_j > (1 - 2\gamma/e(\alpha)) q^* v^*$. Then by Lemma E.2,

$$\frac{1 - 2\gamma/e(\alpha)}{1 + \frac{\gamma}{\left( \frac{\log N}{N} \right)^{\frac{1}{3}} - \gamma}} \leq \frac{q_j v_j}{q^* v^*} \leq 1 - \frac{\alpha}{3}(q_j - q^*)^2 \leq 1 - \frac{\alpha}{3}(\frac{e(\alpha)}{2} - \gamma)^2. \qquad (26)$$

However, the left hand side of (26) approaches to 1 while the right hand side is strictly less than 1, a contradiction. So this case never happens.

3. Finally, if $q_j \geq \frac{e(\alpha)}{2} + \gamma$. We argue that if $\mathrm{ERM}^c$ picks $q_j$ instead of $q_i$, then $R(q_j)$ approximates $R(q_i)$ well, satisfying the conclusion in the lemma. This is because

$$R(q_j) = q_j v_j \geq (\frac{j}{N} - \gamma) v_j$$

$$\geq (1 - \frac{2\gamma}{e(\alpha)}) \frac{j}{N} v_j \qquad\qquad \frac{j}{N} \geq q_j - \gamma \geq \frac{e(\alpha)}{2}$$

$$\geq (1 - \frac{2\gamma}{e(\alpha)}) \frac{i}{N} v_i \qquad\qquad \text{Eq. (24)}$$

$$\geq (1 - \frac{2\gamma}{e(\alpha)})(1 - \frac{2\gamma}{e(\alpha)}) q^* v^* \qquad \text{Eq. (23)}$$

$$= (1 - O(\gamma)) R(q^*).$$

Combining above three steps and the event in the beginning of the proof, we have $R(q_{k^*}) \geq (1 - O(\sqrt{\frac{\log N}{N}}))R(q^*)$ except with probability at most

$$\frac{1}{(N-m)^{\frac{3-2\alpha}{2\alpha}}} + o(\frac{1}{N^{\frac{3-2\alpha}{2\alpha}}}) = O(\frac{1}{N^{\frac{3-2\alpha}{2\alpha}}}).$$

$\square$

**Lemma E.6.** *Let $F$ be an $\alpha$-strongly regular distribution. Suppose $m = o(\sqrt{N})$. Fix $m$ values $v_I$ to be $+\infty$, and randomly draw $N - m$ values $v_{-I}$ from $F$. Let $k^* = \arg\max_{i > cN}\{iv_i\}$, i.e., the index selected by $\mathrm{ERM}^c$, where $\left(\frac{\log N}{N}\right)^{\frac{1}{3}} \leq c \leq \frac{\alpha^{1/(1-\alpha)}}{2}$. Let $\epsilon = \sqrt[4]{\frac{\log N}{N}}$. Then with probability at least $1 - O\left(\frac{1}{N^{\frac{3-2\alpha}{2\alpha}}}\right)$, the following three inequalities hold:*

*1. $q_{k^*} \geq q^* - O(\epsilon)$;*

*2. $k^* \geq [q^* - O(\epsilon)]N > \frac{\alpha^{1/(1-\alpha)}}{2}N$;*

*3. $v_{k^*} \leq [1 + O(\epsilon)]v^*$.*

*The constants in the big $O$'s depend on $\alpha$.*

*Proof.* Define $e(\alpha) = \alpha^{1/(1-\alpha)}$. We have $q^* \geq e(\alpha)$.

For inequality (1), by Lemma E.2 and Lemma E.5, with probability at least $1 - O(\frac{1}{N^{\frac{3-2\alpha}{2\alpha}}})$, we have

$$\frac{\alpha}{3}(q_{k^*} - q^*)^2 \leq \frac{R(q^*) - R(q_{k^*})}{R(q^*)} \leq O\left(\sqrt{\frac{\log N}{N}}\right).$$

Taking the square root, we obtain $q_{k^*} \geq q^* - O\left(\sqrt[4]{\frac{\log N}{N}}\right)$.

To prove (2), note that by Claim E.4, we have $\frac{k^*}{N} \geq q_{k^*} - O\left(\sqrt{\frac{\log N}{N}}\right) \geq q^* - O(\epsilon)$.

Finally, (3) follows from

$$\frac{v_{k^*}}{v^*} = \frac{R(q_{k^*})}{q_{k^*}}\frac{q^*}{R(q^*)} \leq 1 \cdot \frac{q^*}{q_{k^*}} \leq \frac{q^*}{q^* - O(\epsilon)} = 1 + \frac{O(\epsilon)}{q^* - O(\epsilon)} \leq 1 + O\left(\frac{\epsilon}{e(\alpha)}\right).$$

$\square$

## E.2 Detailed Proof of Theorem 2.1 for $\alpha$-Strongly Regular Distributions

Let $\Delta U(\boldsymbol{v}_i, b_I, v_{-I}) = U_i^{\mathrm{TP}}(\boldsymbol{v}_i, b_I, v_{-I}) - U_i^{\mathrm{TP}}(\boldsymbol{v}_i, v_I, v_{-I})$. Similar to the proof for bounded distributions, we have for any $\boldsymbol{v}_i, b_I, v_{-I}$,

$$\Delta U(\boldsymbol{v}_i, b_I, v_{-I}) \leq m_2 \cdot \left(\mathrm{ERM}^c(v_I, v_{-I}) - \mathrm{ERM}^c(b_I, v_{-I})\right) \leq m_2 \cdot \mathrm{ERM}^c(v_I, v_{-I}) \cdot \delta_{m_1}^{\mathrm{worst}}(v_{-I}).$$

By Claim A.2, we have $\mathrm{ERM}^c(v_I, v_{-I}) \leq \mathrm{ERM}^c(\overline{v}_I, v_{-I})$ where $\overline{v}_I$ can be any $m_1$ values (e.g., $+\infty$) that are greater than the maximal value in $v_{-I}$, when $c \geq \frac{m_1}{T_1 K_1}$.

Let $N = T_1 K_1$, define two threshold prices $T_1 = N^{\frac{3(1-\alpha)}{2\alpha}}v^*$ and $T_2 = [1 + O(\epsilon)]v^*$ where $\epsilon = \sqrt[4]{\frac{\log N}{N}}$ as in Lemma E.6.

Note that for sufficiently large $N$, $T_1 > T_2$. With the random draw of $v_{-I}$ from $F$, denote the random variable $\mathrm{ERM}^c(\overline{v}_I, v_{-I})$ by $P$, we have:

$$\begin{aligned}
\mathbb{E}_{\boldsymbol{v}_{-i}}\left[\Delta U(\boldsymbol{v}_i, b_I, v_{-I})\right] &= \mathbb{E}_{v_{-I}}\left[\Delta U(\boldsymbol{v}_i, b_I, v_{-I}) \mid P \leq T_2\right] \cdot \Pr[P \leq T_2] \\
&+ \mathbb{E}_{v_{-I}}\left[\Delta U(\boldsymbol{v}_i, b_I, v_{-I}) \mid T_2 < P \leq T_1\right] \cdot \Pr[T_2 < P \leq T_1] \\
&+ \mathbb{E}_{v_{-I}}\left[\Delta U(\boldsymbol{v}_i, b_I, v_{-I}) \mid P > T_1\right] \cdot \Pr[P > T_1] \\
&\overset{\mathrm{def}}{=} \mathbb{E}_1 + \mathbb{E}_2 + \mathbb{E}_3.
\end{aligned} \tag{27}$$

1. For the first term $\mathbb{E}_1$,

$$\begin{aligned}
\mathbb{E}_1 &= \mathbb{E}_{v_{-I}}\left[\Delta U(\boldsymbol{v}_i, b_I, v_{-I}) \mid P \le T_2\right] \cdot \Pr[P \le T_2] \\
&\le \mathbb{E}_{v_{-I}}\left[m_2 \cdot P \cdot \delta_{m_1}^{\text{worst}}(v_{-I}) \mid P \le T_2\right] \cdot \Pr[P \le T_2] \\
&\le m_2 \cdot T_2 \cdot \mathbb{E}_{v_{-I}}\left[\delta_{m_1}^{\text{worst}}(v_{-I}) \mid P \le T_2\right] \cdot \Pr[P \le T_2] \\
&\le m_2 \cdot [1 + O(\epsilon)]v^* \cdot \mathbb{E}_{v_{-I}}\left[\delta_{m_1}^{\text{worst}}(v_{-I})\right] \\
&= O\left(m_2 \cdot v^* \cdot \Delta_{N,m_1}^{\text{worst}}\right).
\end{aligned}$$

2. For the second term, we claim that $\mathbb{E}_2 = O(\frac{m_2 v^*}{\sqrt{N}})$.

   By Lemma E.6, we have $\Pr[P > [1 + O(\epsilon)]v^*] \le O(\frac{1}{N^{\frac{3-2\alpha}{2\alpha}}})$. Therefore,

$$\begin{aligned}
\mathbb{E}_2 &= \mathbb{E}_{v_{-I}}\left[\Delta U(\boldsymbol{v}_i, b_I, v_{-I}) \mid T_2 < P \le T_1\right] \cdot \Pr[T_2 < P \le T_1] \\
&\le \mathbb{E}_{v_{-I}}\left[m_2 \cdot P \cdot 1 \mid T_2 < P \le T_1\right] \cdot \Pr[T_2 < P \le T_1] \\
&\le m_2 \cdot T_1 \cdot \Pr[P > T_2] \\
&\le m_2 \cdot N^{\frac{3(1-\alpha)}{2\alpha}} v^* \cdot O\left(\frac{1}{N^{\frac{3-2\alpha}{2\alpha}}}\right) \\
&= O\left(\frac{m_2 v^*}{\sqrt{N}}\right).
\end{aligned}$$

3. For the third term, we claim that $\mathbb{E}_3 = o(\frac{m_2 v^*}{\sqrt{N}})$.

   Let $B$ be the upper bound on the support of $F$ ($B$ can be $+\infty$). Let $F_P(x)$ be the distribution of $P$. For convenience, suppose it is continuous and has density $f_P(x)$. We have:

$$\begin{aligned}
\mathbb{E}_3 &= \mathbb{E}_{v_{-I}}\left[\Delta U(\boldsymbol{v}_i, b_I, v_{-I}) \mid P > T_1\right] \cdot \Pr[P > T_1] \\
&\le \mathbb{E}_{v_{-I}}\left[m_2 \cdot P \cdot 1 \mid P > T_1\right] \cdot \Pr[P > T_1] \\
&= m_2 \cdot \mathbb{E}_{v_{-I}}\left[P \mid P > T_1\right] \cdot \Pr[P > T_1] \\
&= m_2 \cdot \int_{T_1}^{B} x f_P(x)\mathrm{d}x \\
&= m_2 \cdot \left(\int_{T_1}^{B} [1 - F_P(x)]\mathrm{d}x + T_1[1 - F_P(T_1)]\right).
\end{aligned}$$

   Let $\max\{v_{-I}\}$ denote the maximum value in the $N - m_1$ samples $v_{-I}$. By Lemma E.3, we have for any $x \ge v^*$,

$$1 - F_P(x) = \Pr[P > x] \le \Pr[\max\{v_{-I}\} > x] \le N\left(\frac{v^*}{(1-\alpha)x + \alpha v^*}\right)^{\frac{1}{1-\alpha}}.$$

   Thus,

$$\begin{aligned}
&\int_{T_1}^{B} [1 - F_P(x)]\mathrm{d}x + T_1[1 - F_P(T_1)] \\
&\le \int_{T_1}^{B} N\left(\frac{v^*}{(1-\alpha)x + \alpha v^*}\right)^{\frac{1}{1-\alpha}} \mathrm{d}x + T_1 N\left(\frac{v^*}{(1-\alpha)T_1 + \alpha v^*}\right)^{\frac{1}{1-\alpha}} \\
&\le \frac{N}{\alpha} \cdot \frac{(v^*)^{\frac{1}{1-\alpha}}}{[(1-\alpha)T_1 + \alpha v^*]^{\frac{\alpha}{1-\alpha}}} + T_1 N\left(\frac{v^*}{(1-\alpha)T_1 + \alpha v^*}\right)^{\frac{1}{1-\alpha}} \\
&= \frac{N}{\alpha} \cdot (T_1 + \alpha v^*)\left(\frac{v^*}{(1-\alpha)T_1 + \alpha v^*}\right)^{\frac{1}{1-\alpha}} \\
&= O\left(\frac{v^*}{\sqrt{N}}\right),
\end{aligned}$$

   as desired.

Combining the three items,

$$\mathbb{E}_{v_{-I}}\left[\Delta U(v_I, b_I, v_{-I})\right] = O\left(m_2 v^* \Delta_{N,m_1}^{\text{worst}}\right) + O\left(\frac{m_2 v^*}{\sqrt{N}}\right),$$

where the constants in $O$'s depend on $\alpha$.

# F  Lower Bounds

## F.1  Discussion

**A lower bound on $\Delta_{N,m}^{\text{worst}}$.** Theorem 1.3 gives an upper bound on $\Delta_{N,m}^{\text{worst}}$ for a specific range of $c$'s. When one considers respective lower bounds, a preliminary question would be: how does the choice of $c$ affect the possible lower bound? The following result shows that it is enough to prove a lower bound for one specific $c$ in the range of allowed $c$'s. The same lower bound will then hold for all $c$'s in that range.

**Proposition F.1.** *Let $\Delta_{N,m}^{\text{worst}}(c)$ denote the worst-case incentive-awareness measure of* $\mathrm{ERM}^c$. *Suppose $m = o(\sqrt{N})$.*

- *For bounded distributions, $\Delta_{N,m}^{\text{worst}}(c) = \Delta_{N,m}^{\text{worst}}(\frac{m}{N})$, for any $c \in [\frac{m}{N}, \frac{1}{2D}]$.*

- *For MHR distributions, $\Delta_{N,m}^{\text{worst}}(c)$ is bounded by $\Delta_{N,m}^{\text{worst}}(\frac{m}{N}) \pm O(\frac{1}{N})$, for any $c \in [\frac{m}{N}, \frac{1}{4e}]$.*

*Proof.* Fix any $d \in (0,1)$ and any $c \in [\frac{m}{N}, \frac{d}{2}]$. Recall that $\Delta_{N,m}^{\text{worst}}(c) = \mathbb{E}_{v_{-I} \sim F}[\delta_m^{\text{worst}}(v_{-I}, c)]$, where, letting $\overline{v}_I$ be a vector of $m$ identical values that are equal to $\max v_{-I}$, then by Claim A.2,

$$\delta_m^{\text{worst}}(v_{-I}, c) = 1 - \frac{\inf_{b_I \in \mathbb{R}_+^m} P(b_I, v_{-I}, c)}{P(\overline{v}_I, v_{-I}, c)}.$$

Let $k^*(v, c) = \arg\max_{i > cN}\{iv_{(i)}\}$ where $v = (\overline{v}_I, v_{-I})$, i.e. the index of $P(v, c)$ in $v$. We show that, if $k^*(v, c) > dN$ for $c = \frac{m}{N}$, then

- $P(\overline{v}_I, v_{-I}, c') = P(\overline{v}_I, v_{-I}, \frac{m}{N})$ for any $c' \in [\frac{m}{N}, d]$. To see this, note that

$$k^*(v, \frac{m}{N}) = \arg\max_{i > \frac{m}{N}N}\{iv_{(i)}\} = \arg\max_{i > dN}\{iv_{(i)}\} = \arg\max_{i > c'N}\{iv_{(i)}\} = k^*(v, c'),$$

 where the second equality follow from our assumption that $k^*(v, \frac{m}{N}) > dN > m$ and the third equality is because $\frac{m}{N} \leq c' \leq d$.

- $\inf_{b_I \in \mathbb{R}_+^m} P(b_I, v_{-I}, c') = \inf_{b_I \in \mathbb{R}_+^m} P(b_I, v_{-I}, \frac{m}{N})$ for any $c' \in [\frac{m}{N}, \frac{d}{2}]$. Fix $c = \frac{m}{N}$ and consider any $c' \in [\frac{m}{N}, \frac{d}{2}]$. Let $b_I \in \mathbb{R}_+^m$ be any bids such that $P(b_I, v_{-I}, c) < P(\overline{v}_I, v_{-I}, c)$. Let $v^b = (b_I, v_{-I})$. Consider $k^*(v^b, c)$, we have $v^b_{(k^*(v^b, c))} < v_{(k^*(v, c))}$, so $k^*(v^b, c)$ must be greater than the index of $v_{(k^*(v, c))}$ in $v^b$. The index of $v_{(k^*(v, c))}$ in $v^b$ is at least $k^*(v, c) - m$, thus $k^*(v^b, c) > k^*(v, c) - m > dN - m \geq \frac{d}{2}N$. We claim that $P(b_I, v_{-I}, c') = P(b_I, v_{-I}, c)$. To see this, note that

$$k^*(b_I, v_{-I}, \frac{m}{N}) = \arg\max_{i > \frac{m}{N}N}\{iv^b_{(i)}\} = \arg\max_{i > \frac{d}{2}N}\{iv^b_{(i)}\} = \arg\max_{i > c'N}\{iv^b_{(i)}\} = k^*(b_I, v_{-I}, c'),$$

 where the second equality is because $k^*(v^b, \frac{m}{N}) > \frac{d}{2}N$, and the third equality follows from $\frac{m}{N} \leq c' \leq \frac{d}{2}$.

Thus, $\delta_m^{\text{worst}}(v_{-I}, c) = \delta_m^{\text{worst}}(v_{-I}, \frac{m}{N})$ for any $c \in [\frac{m}{N}, \frac{d}{2}]$. Define $\mathrm{E}(c) = [k^* \leq dN]$, then

$$\begin{aligned}
\Delta_{N,m}^{\text{worst}}(c) &= \mathbb{E}[\delta_m^{\text{worst}}(v_{-I}, c)] \\
&= \mathbb{E}[\delta_m^{\text{worst}}(v_{-I}, c) \mid \overline{\mathrm{E}(c)}] \cdot \Pr[\overline{\mathrm{E}(c)}] + \mathbb{E}[\delta_m^{\text{worst}}(v_{-I}, c) \mid \mathrm{E}(c)] \cdot \Pr[\mathrm{E}(c)] \\
&= \mathbb{E}[\delta_m^{\text{worst}}(v_{-I}, \frac{m}{N}) \mid \overline{\mathrm{E}(c)}] \cdot \Pr[\overline{\mathrm{E}(c)}] + \mathbb{E}[\delta_m^{\text{worst}}(v_{-I}, c) \mid \mathrm{E}(c)] \cdot \Pr[\mathrm{E}(c)].
\end{aligned}$$

For bounded distributions, consider $d = \frac{1}{D}$ and $c \in [\frac{m}{N}, \frac{1}{2D}]$. We have proved in Theorem 1.3 that $\Pr[\mathrm{E}(c)] = 0$ for any $c \in [\frac{m}{N}, \frac{d}{2}]$. Thus $\Delta_{N,m}^{\text{worst}}(c) = \mathbb{E}[\delta_m^{\text{worst}}(v_{-I}, \frac{m}{N}) \mid \overline{\mathrm{E}(c)}] \cdot \Pr[\overline{\mathrm{E}(c)}] = \Delta_{N,m}^{\text{worst}}(\frac{m}{N})$.

For MHR distributions, let $d = \frac{1}{2e}$, as proved in Lemma D.5, $\Pr[\mathrm{E}] = O(\frac{1}{N})$ for $c \in [\frac{m}{N}, \frac{1}{4e}]$. Then $0 \leq \mathbb{E}[\delta_m^{\text{worst}}(v_{-I}, c) \mid \mathrm{E}(c)] \cdot \Pr[\mathrm{E}(c)] \leq 1 \cdot \Pr[\mathrm{E}(c)] = O(\frac{1}{N})$. Thus $\left|\Delta_{N,m}^{\text{worst}}(c) - \Delta_{N,m}^{\text{worst}}(\frac{m}{N})\right| \leq O(\frac{1}{N})$ for any $c \in [\frac{m}{N}, \frac{1}{4e}]$.

$\square$

Lavi et al. [26] show a lower bound that can be compared to our upper bounds in Theorem 1.3. Specifically, they show that for the two-point distribution $v = 1$ and $v = 2$, each w.p. 0.5, $\Delta_{N,1}^{\text{worst}} = \Omega(1/\sqrt{N})$. It is easy to adopt their analysis to any two-point distribution with $v_1 = 1$ and $v_2 > 1$. Since this is a bounded distribution, we obtain the following corollary:

**Corollary F.2.** *For the class of bounded distribution with support in $[1, D]$, and any choice of $\frac{m}{N} \leq c \leq \frac{1}{2D}$,* ERM$^c$ *gives $\Delta_{N,m}^{\text{worst}}(c) = \Omega(\frac{1}{\sqrt{N}})$ where the constant in $\Omega$ depends on $D$.*

It remains open to prove other lower bounds on $\Delta_{N,m}^{\text{worst}}$, especially for MHR distributions.

**A lower bound on the approximate truthfulness parameter, $\epsilon_1$.** Since $\Delta_{N,m}^{\text{worst}}$ only upper bounds the approximate truthfulness parameter $\epsilon_1$, a lower bound on $\Delta_{N,m}^{\text{worst}}$ does not immediately implies a lower bound on $\epsilon_1$. However, an argument similar to above shows the same lower bound directly on $\epsilon_1$. Consider the two-point distribution $F$ where for $X \sim F$, $\Pr[X = 1] = 1 - \frac{1}{D}$ and $\Pr[X = D] = \frac{1}{D}$. For simplicity let $K_1 = K_2 = 2$ and suppose bidder $i$ participates in $m_1$ and $m_2$ auctions in the two phases, respectively. Let $N = T_1 K_1$ and assume $m_1 = o(\sqrt{N})$. Suppose the first-phase mechanism $\mathcal{M}$ is the second price auction with no reserve price. Then,

**Proposition F.3.** *In the above setting, $\epsilon_1 = \Omega\left(\frac{m_2}{\sqrt{N}}\right)$ for any $c \in [\frac{m_1}{N}, \frac{1}{2D}]$, where the constant in $\Omega$ depends on $D$.*

The proof is in Appendix F.2. Once again, it remains open (and interesting, we believe) to prove a lower bound for MHR distributions, and to close the gap between our upper bound for bounded distributions which is $O(N^{-1/3} \log^2 N)$.

## F.2 Proof of Proposition F.3: Lower Bound for the Two-Phase Model

Consider the two-point distribution $F$ where for $X \sim F$, $\Pr[X = 1] = 1 - \frac{1}{D}$ and $\Pr[X = D] = \frac{1}{D}$. For simplicity let $K_1 = K_2 = 2$, and suppose bidder $i$ participates in $m_1$ and $m_2$ auctions in the two phases, respectively. Let $N = T_1 K_1$ and assume $m_1 = o(\sqrt{N})$. Suppose the first-phase mechanism $\mathcal{M}$ is the second price auction with no reserve price. We argue that to satisfy $\epsilon_1$-approximate truthfulness, $\epsilon_1$ must be

$$\Omega\left(\frac{m_2(D-1)^2}{\sqrt{(D-1)N}}\right), \text{ for } c \in [\frac{m}{N}, \frac{1}{2D}].$$

Suppose the values of bidder $i$ across two phases are all $D$'s, i.e.,

$$\boldsymbol{v}_i = (\overbrace{D, \ldots, D}^{m_1}, \overbrace{D, \ldots, D}^{m_2}),$$

and bidder $i$ bids $m_1$ $(D - \epsilon)$'s with $\epsilon = \frac{D^2}{N}$ in the first phase (assume $N \gg D^2$),

$$b_I = (\overbrace{D - \epsilon, \ldots, D - \epsilon}^{m_1}).$$

Recall the definition of the interim utility of bidder $i$:

$$\mathbb{E}_{v_{-I}}\left[U_i^{\text{TP}}(\boldsymbol{v}_i, b_I, v_{-I})\right] = \mathbb{E}_{v_{-I}}\left[U_i^{\mathcal{M}}(\boldsymbol{v}_i, b_I, v_{-I}) + m_2 u^{K_2}(D, P(b_I, v_{-I}))\right].$$

- First consider the increase of interim utility in the second phase. If the reserve price is $D$, then bidder $i$'s utility is

$$u^{K_2}(D, D) = 0. \tag{28}$$

  If the reserve price is 1, then her utility becomes

$$u^{K_2}(D, 1) = (1 - \frac{1}{D}) \cdot (D - 1) + \frac{1}{D} \cdot 0 = \frac{(D-1)^2}{D}. \tag{29}$$

  We then consider the probability that the reserve price is decreased from $D$ to 1 because bidder $i$ deviates from $D$ to $D - \epsilon$. This probability is over the random draw of $N - m_1$ values $v_{-I}$. Suppose there are exactly $(\frac{1}{D}N + 1 - m_1)$ $D$'s in $v_{-I}$. Then when bidder $i$ bids truthfully, there are $(\frac{1}{D}N + 1)$ $D$'s in $(v_I, v_{-I})$ in total, which results in $P(v_I, v_{-I}) = D$ because $(\frac{1}{D}N + 1) \cdot D > N \cdot 1$. However, if bidder $i$ deviates to $b_I$, then $P(b_I, v_{-I})$ becomes 1, because

$$(\frac{1}{D}N + 1) \cdot (D - \epsilon) = N(1 + \frac{D}{N})(1 - \frac{D}{N}) < N \cdot 1, \quad \text{and} \quad (\frac{1}{D}N + 1 - m_1) \cdot D \leq N \cdot 1.$$

  Thus, the reserve price is decreased from $D$ to 1 with probability at least:

$$\Pr[Bin(N - m_1, \frac{1}{D}) = \frac{1}{D}N + 1 - m_1] = \Omega\left(\frac{1}{\sqrt{\frac{1}{D}(1 - \frac{1}{D})N}}\right). \tag{30}$$

Combining (30), (29) and (28), we obtain

$$\mathbb{E}_{v_{-I}}\left[m_2 u^{K_2}(D, P(b_I, v_{-I})) - m_2 u^{K_2}(D, P(v_I, v_{-I}))\right]$$

$$\geq \Omega\left(\frac{1}{\sqrt{\frac{1}{D}(1-\frac{1}{D})N}}\right) m_2 \left(\frac{(D-1)^2}{D} - 0\right) = \Omega\left(\frac{m_2(D-1)^2}{\sqrt{(D-1)N}}\right). \quad (31)$$

- Then we upper bound the utility loss due to non-truthful bidding in the first phase for bidder $i$. Note that since $\mathcal{M}$ is the second price auction with no reserve price, no matter bidder $i$ bids $D$ or $D - \epsilon > 1$, her interim utility is the same:

$$\mathbb{E}_{v_{-I}}\left[U_i^{\mathcal{M}}(\boldsymbol{v}_i, v_I, v_{-I})\right] = m_1\left((1 - \frac{1}{D})(D-1) + \frac{1}{D}\cdot 0\right) = \frac{m_1(D-1)^2}{D},$$

$$\mathbb{E}_{v_{-I}}\left[U_i^{\mathcal{M}}(\boldsymbol{v}_i, b_I, v_{-I})\right] = m_1\left((1 - \frac{1}{D})(D-1) + \frac{1}{D}\cdot 0\right) = \frac{m_1(D-1)^2}{D}.$$

Thus

$$\mathbb{E}_{v_{-I}}\left[U_i^{\mathcal{M}}(\boldsymbol{v}_i, b_I, v_{-I}) - U_i^{\mathcal{M}}(\boldsymbol{v}_i, v_I, v_{-I})\right] = 0. \quad (32)$$

Finally, by (31) and (32), we have

$$\mathbb{E}_{v_{-I}}\left[U_i^{\text{TP}}(\boldsymbol{v}_i, b_I, v_{-I}) - U_i^{\text{TP}}(\boldsymbol{v}_i, v_I, v_{-I})\right] \geq \Omega\left(\frac{m_2(D-1)^2}{\sqrt{(D-1)N}}\right),$$

which gives a lower bound on $\epsilon_1$.

# G  Unbounded Regular Distributions

## G.1  Discussion

Theorem 2.2 shows that approximate truthfulness and revenue optimality can be obtained simultaneously for bounded (regular) distributions and for MHR distributions. A natural question would then be: what is the largest class of value distribution that we can consider? If $K_1 > 1$ or $K_2 > 1$ (i.e., each auction includes multiple bidders, at least 2), then running a second price auction with an anonymous reserve price may not be optimal if the distribution is non-regular [30]. Moreover, even in the one-bidder case, the sample complexity literature analyzes the ERM algorithm only for regular or for non-regular and bounded distributions. For other classes of distributions, ERM does not seem to be the correct choice. Thus, the class of general unbounded regular distributions is the largest class we can consider. Still, our results do not cover this entire class since MHR distributions is a strict sub-class of regular distributions and for regular but non-MHR distributions we assume boundedness.

Our results can be generalized to the class of $\alpha$-strongly regular distributions with $\alpha > 0$. As defined in [11], a distribution $F$ with positive density function $f$ on its support $[A, B]$ where $0 \leq A \leq B \leq +\infty$ is $\alpha$-*strongly regular* if the virtual value function $\phi(x) = x - \frac{1-F(x)}{f(x)}$ satisfies $\phi(y) - \phi(x) \geq \alpha(y-x)$ whenever $y > x$ (or $\phi'(x) \geq \alpha$ if $\phi(x)$ is differentiable). As special cases, regular and MHR distributions are 0-strongly and 1-strongly regular distributions, respectively. For $\alpha > 0$, we obtain bounds similar to MHR distributions on $\Delta_{N,m}^{\text{worst}}$ and approximate incentive-compatibility in the two-phase model and the uniform-price auction. Specifically, Theorem 1.3 can be extended to any $\alpha$-strongly regular distribution with $\alpha > 0$ as follows:

**Theorem G.1.** *If $F$ is $\alpha$-strongly regular for $0 < \alpha \leq 1$, then $\Delta_{N,m}^{\text{worst}} = O\left(m\frac{\log^3 N}{\sqrt{N}}\right)$, when $m = o(\sqrt{N})$ and $\frac{m}{N} \leq \left(\frac{\log N}{N}\right)^{1/3} \leq c \leq \frac{\alpha^{1/(1-\alpha)}}{4}$. The constants in $O$ and $o$ depend on $\alpha$.*

Note that this bound holds only for large enough $N$'s since $\alpha^{1/(1-\alpha)} \to 0$ as $\alpha \to 0$. However, for any fixed $\alpha > 0$ there exists a large enough $N$ such that the relevant range for appropriate $c$'s will be non-empty.[5] The proof of the upper bound on $\Delta_{N,m}^{\text{worst}}$ is similar to the proof for MHR distributions (Lemma D.6), except that, we need $c \geq \left(\frac{\log N}{N}\right)^{1/3}$ to guarantee $\Pr[\text{E}] = O(\frac{1}{N})$ (Lemma E.6); thus we omit the proof.

Similarly, both of Theorem 2.1 which says that the two-phase model is $(O(m_2 v^* \Delta_{T_1 K_1, m_1}^{\text{worst}}) + O(\frac{m_2 v^*}{\sqrt{T_1 K_1}}))$-BIC and Theorem 3.1 which says that the uniform-price auction is $(m, (O(v^* \Delta_{N,m}^{\text{worst}}) + O(\frac{v^*}{\sqrt{N}})))$-group BIC, hold for any $\alpha$-strongly regular distribution with $\alpha > 0$. The proof of the former is in Appendix E.2 and the latter is omitted.

Figure 1: For equal-revenue distribution, $\Delta_{N,1}^{\mathrm{worst}}$ (in black) on the $y$-axis as a function of $N$ on the $x$-axis, with $c = 1/N$. Three other functions are plotted for reference.

Figure 2: For equal-revenue distribution, $\Delta_{N,1}^{\mathrm{worst}}$ (in black) on the $y$-axis as a function of $N$ on the $x$-axis, with $c = \Theta((\log N/N)^{1/3})$. Three other functions are plotted for reference.

It remains an open problem for future research whether $\mathrm{ERM}^c$ is incentive-aware in the large for regular distributions that are not $\alpha$-strongly regular for any $\alpha > 0$. For these distributions additional technical challenges exist since the choice of $c$ in $\mathrm{ERM}^c$ creates a clash between approximate truthfulness and approximate revenue optimality. Unlike MHR and bounded regular distributions for which we can fix $c = m/N$ to obtain approximate truthfulness and revenue optimality, for arbitrary unbounded distribution we have to choose $c$ more carefully. If $c$ is too large, for example, a positive constant, then we cannot obtain nearly optimal revenue.

Specifically, to obtain close-to-optimal revenue for all bounded distributions on $[1, D]$ it is easy to verify that we need $c \leq 1/D$. Since the class of unbounded regular distributions contains all bounded regular distributions for all $D \in \mathbb{R}_+$, it follows that $c$ cannot be a constant. We therefore need to consider a non-constant $c(N)$. In fact, it has been shown in [23] that if $c(N) \to 0$ as $N \to \infty$ then approximate revenue optimality can be satisfied. However, if $c$ is too small, truthfulness will be violated, as discussed in the following two examples (assume $m = 1$ for simplicity).

**Example G.2** (Small $c$ hurts truthfulness). *Suppose we choose $c(N) = \frac{1}{N}$, that is, $\mathrm{ERM}^c$ ignores only the largest sample. Consider the equal-revenue distribution $F(v) = 1 - \frac{1}{v}$ for $v \in [1, +\infty)$. Note that this is a 0-strongly regular distribution but not $\alpha$-strongly regular for any $\alpha > 0$ since for any $x$, $\phi(x) = 0$. Similarly to Yao [31], we prove in Appendix G.2 that $\Delta_{N,1}^{\mathrm{worst}}$ does not go to 0 as $N \to +\infty$. This is also visible in Fig. 1, which shows in black $\Delta_{N,1}^{\mathrm{worst}}$ (on the y-axis) as a function of $N$ (on the x-axis). This was obtained via simulation, for $c = 1/N$. Three other functions are plotted in other colors, for reference. However, whether $\Delta_{N,1}^{\mathrm{worst}} \to 0$ crucially depends on the choice of $c$, as can be seen in Fig. 2, where $\Delta_{N,1}^{\mathrm{worst}}$ seems to converge to zero with $c = \Theta((\log N/N)^{1/3})$.*

**Example G.3** (Does an intermediate $c$ hurt truthfulness as well?). *Now assume $c = \Theta((\log N/N)^{1/3})$, and consider the "triangular" distribution $F(v) = 1 - \frac{1}{v+1}$ for $v \in [0, +\infty)$. This distribution can be seen as the limit of a series of bounded regular distributions whose upper bounds and optimal reserve prices both tend to $+\infty$. Note that it is a regular distribution but not $\alpha$-strongly regular for any $\alpha > 0$. We do not know whether $\Delta_{N,1}^{\mathrm{worst}} \to 0$ as $N \to \infty$ (see Fig. 3). In particular, our main lemma (Lemma B.1) may not suffice to analyze this distribution as $\Pr[\mathrm{E}]$ (as defined in that lemma) is unlikely to go to zero as $N$ goes to infinity (see Fig. 4 for simulation results).*

## G.2 Proof of Example G.2: $\Delta_{N,1}^{\mathrm{worst}} \not\to 0$ for the Equal-Revenue Distribution and $c = 1/N$

We show that when $F$ is the equal-revenue distribution, $F(v) = 1 - \frac{1}{v}$, ($v \in [1, +\infty)$), and $c = 1/N$, $\Delta_{N,1}^{\mathrm{worst}}$ does not go to 0 as $N \to +\infty$.

Recall Definition 1.2, $\Delta_{N,1}^{\mathrm{worst}} = \mathbb{E}_{v_{-I}}[\delta_{N,1}^{\mathrm{worst}}(v_{-I})]$: we draw $N - 1$ i.i.d. values $v_{-I} = \{v_2, \ldots, v_N\}$ from $F$, and a bidder can change any value $v_I = v_1$ to any non-negative bid $b_1$. Let the other values $v_{-I} = \{v_2, \ldots, v_N\}$ be sorted as $v_{(2)} \geq \cdots \geq v_{(N)}$. Let $\lambda = 10, \lambda' = 1$, and let $T_{N-1}$ be the event that $v_{(2)} > \lambda(N-1)$ and $v_{(3)} < \lambda'(N-1)$. Let $V_{N-1}$ be the event that $\max_{3 \leq i \leq N}\{i \cdot v_{(i)}\} \leq \lambda(N-1)$.

When $v_{-I}$ satisfies event $T_{N-1} \wedge V_{N-1}$, then

$$\delta_m^{\mathrm{worst}}(v_{-I}) \geq 1 - \frac{\lambda'}{\lambda}. \tag{33}$$

Figure 3: $\Delta_{N,1}^{\text{worst}}$ for the triangular distribution, with $c = \Theta((\log N/N)^{1/3})$

Figure 4: $\Pr[k^* < 2cN]$ as a function of $N$ for the triangular distribution, with $c = (\log N/N)^{1/3}/5$. Our simulations show that $\Pr[E] \geq \Pr[k^* < 2cN] \geq 0.815$.

This is because when $T_{N-1} \wedge V_{N-1}$ happens, there exists some $v_1 > \lambda(N-1)$, resulting $P(v_I, v_{-I}) > \lambda(N-1)$, while the bidder can strategically bid $b_1 < \lambda'(N-1)$ and change the price to $P(b_I, v_{-I}) < \lambda'(N-1)$.

Moreover, we will show that the probability that the event $T_{N-1} \wedge V_{N-1}$ happens satisfies

$$\Pr[T_{N-1} \wedge V_{N-1}] \geq 0.9 \cdot \frac{1}{\lambda} e^{-\frac{2}{\lambda'}}. \tag{34}$$

Combining (33) and (34), we know

$$\Delta_m^{\text{worst}} \geq (1 - \frac{\lambda'}{\lambda}) \cdot (0.9 \cdot \frac{1}{\lambda} e^{-\frac{2}{\lambda'}}) > 0.$$

The proof of (34) is separated into two parts. Firstly,

$$\Pr[T_{N-1}] = \binom{N-1}{1} \frac{1}{\lambda(N-1)} (1 - \frac{1}{\lambda'(N-1)})^{N-2} \geq \frac{1}{\lambda} e^{-\frac{2}{\lambda'}}.$$

Then we show that

$$\Pr[\overline{V_{N-1}} \mid T_{N-1}] < 0.1.$$

Let $z_3, \dots, z_N$ be i.i.d. random draws according to $F$ conditioning on $z < \lambda'(N-1)$, i.e., for any $t \in [1, N-1]$, recalling that $\lambda' = 1$,

$$\Pr_{z \sim F}[z > t \mid z < \lambda'(N-1)] = \frac{1}{1 - \frac{1}{N-1}} (\frac{1}{t} - \frac{1}{N-1}).$$

Let $Y_{N-1}^{max} = \max_{3 \leq i \leq N} \{i \cdot z_{(i)}\}$ where $z_{(3)} \geq \cdots \geq z_{(N)}$ is the sorted list of $z_i$'s. Clearly,

$$\Pr[\overline{V_{N-1}} \mid T_{N-1}] = \Pr[Y_{N-1}^{max} \geq \lambda(N-1)].$$

For any $t \geq 1$, let $M_t$ be the number of $z_i$'s ($3 \leq i \leq N$) satisfying $z_i \geq t$, and $B_t$ be the event that $t \cdot (M_t + 2) \geq \frac{\lambda(N-1)}{2}$. Let $t_k = \frac{N-1}{2^k}$ for $1 \leq k \leq \lfloor \log_2(N-1) \rfloor$. As the event $Y_{N-1}^{max} \geq \lambda(N-1)$ implies $\bigvee_{1 \leq k \leq \lfloor \log_2(N-1) \rfloor} B_{t_k}$, we have

$$\Pr[Y_{N-1}^{max} \geq \lambda(N-1)] \leq \sum_{k=1}^{\lfloor \log_2(N-1) \rfloor} \Pr[B_{t_k}]. \tag{35}$$

Note that $\mathbb{E}[M_t] = \frac{N-1}{t} - 1$. Using Chernoff's bound, we have

$$\Pr[B_t] = \Pr[M_t \geq \frac{\lambda(N-1)}{2t} - 2] \leq e^{-\frac{1}{3}(\frac{N-1}{t} - 1)(\frac{\lambda}{2} - 1)^2}. \tag{36}$$

Combining (35) and (36), we know

$$\Pr[\overline{V_{N-1}} \mid T_{N-1}] = \Pr[Y_{N-1}^{max} \geq \lambda(N-1)] \leq \sum_{k=1}^{\lfloor \log_2(N-1) \rfloor} e^{-\frac{1}{3}(2^k - 1)(\frac{\lambda}{2} - 1)^2} < \sum_{k=1}^{+\infty} e^{-\frac{1}{3}k(\frac{\lambda}{2} - 1)^2} < 0.1,$$

and this completes the proof of (34).