[Reviews · NeurIPS 2020]

Review 1

Summary and Contributions: Please see the comments to authors section

Strengths: Please see the comments to authors section

Weaknesses: Please see the comments to authors section

Correctness: Please see the comments to authors section

Clarity: Please see the comments to authors section

Relation to Prior Work: Please see the comments to authors section

Reproducibility: Yes

Additional Feedback: This paper studies mechanisms that use an agent's current round bids to influence his future round utility and ask how bad would truthful bidding be for an agent in such an auction. The motivation for this is that often Bayesian mechanism design uses distributional knowledge to design good mechanisms, and such distributional samples. These samples in turn are not available from a stand alone source, but are rather generated from prior interactions of the agent with the mechanism. Naturally, when an agent realizes that the mechanism could use his current round bids against himself in the future he would not be willing to report his true valuation (even if the current round mechanism is truthful). There has been a good amount of prior work on this problem, and most of them assume that the agent has control over all the samples that will be used to design the mechanism he will face in the future. Accordingly, one paper shows that when buyers don't discount their future utility, then the seller is going to have a linear revenue regret --- i.e., compared with the revenue the seller could have achieved against a buyer who always bids truthfully, the revenue achievable is much smaller. Given this, the subsequent papers resorted to studying what happens when the buyer discounts the future more than the seller does, and here they establish what is the smallest revenue regret achievable by the seller. The main point of departure of this work, and some very recent prior work is that not all samples are in the control of any one buyer for him to go and influence them as he likes. The samples are distributed across numerous buyers. The question then is how much influence does a buyer have as a function of the number of samples he controls? A recent paper by Lavi et al introduced the notion of incentive-awareness-measure (or as they call it, discount ratio) to characterize the worst-case percentage reduction in reserve price that would occur for a given mechanism if the buyer were to control some number samples. That paper and a subsequent paper by Yao studied the case of m=1 and showed that as the number of samples goes to infinity, the reduction percentage goes to 0. While this is somewhat expected, the true question then is what is the rate at which this goes to zero --- this is the central question studied in this paper and the paper has a closed form expression for the incentive-awareness-measure as a function of the number m of samples the buyer controls and the total number N, for two large classes of distributions. The paper applies this incentive awareness measure to study the loss in utility an agent would suffer in the Empirical Revenue Maximization mechanism by bidding truthfully. Essentially it computes the epsilon for which the mechanism is epsilon-BNE. It has another application on Uniform Price Auctions where the paper generalizes an earlier result to arbitrary bids and continuous support for distribution as well. Overall I quite like the approach taken by this paper and the recent couple of papers that breathe fresh air into the space of mechanisms using agents' bid against themselves in the future. The earlier results painted a pessimistic picture and showed that good revenue can be obtained by the seller only when one settles for a somewhat unsatisfying feature of increased future discounting by the buyer than the seller. These recent papers aim to get us out of this pessimistic world, and this paper in particular studies one of the central questions to be addressed here. I think it would be a good fit for NeurIPS. After author rebuttal: Thanks for the rebuttal. I am reducing the score a bit based on the discussion that followed, although I remain enthusiastic about the paper.


Review 2

Summary and Contributions: Optimal mechanism design requires the designer to know the distribution of valuations of the bidders. If the designer doesn't already, it's natural to try to learn it from the bids. The problem is that if bidders are aware that this is happening then it distorts their incentives in bidding -- e.g., they may want to bid lower in order to get lower prices later. The authors study asymptotically how strong the incentives are to bid differently are and how close we get to optimal revenue.

Strengths: The problem under study is both difficult and important. The results seem technically nice. I like the idea of "let's take the simple idea, we know there's some issue with it, but let's evaluate how bad it actually gets, maybe it's not even that bad."

Weaknesses: My main concern is with approximate incentive compatibility. The authors claim that "if a mechanism is \epsilon-BIC and \lim_{n \to \infty} \epsilon = 0, then each bidder knows that if all other bidders are bidding truthfully then the gain from any deviation from truthful bidding is negligible for her." In my opinion there are several problems with this view. First of all, just that it goes to 0 in the limit does not mean that the incentives are in any practical sense negligible in the finite real world. Second, the statement correctly conditions "if all other bidders are bidding truthfully" but this could also start falling apart -- what if there's some suspicion that some other bidder may not bid truthfully, maybe that's enough to dramatically change the incentives, then everything falls apart -- it could be very fragile. There's also the issue that if we relax BIC to epsilon-BIC, then the optimal revenue will grow above that of the Myerson auction, which is used as "OPT". There are some recent techniques out there for turning approximately BIC into exactly BIC mechanisms, at some loss to revenue. Can those be applied here? If so what would be the result on revenue? That could alleviate all these concerns. Another issue is that I don't have a great understanding of what these asymptotic bounds mean in practice. At some point *for a different paper* on auctions with asymptotic bounds I went through the exercise of trying to figure out what the actual numbers would be for a reasonable approximation and it turned out to require many millions of bids which seemed unrealistic. I'm not saying that that is the case here, but there are a lot of parameters and I'd like to get some more intuition. There are a few figures late in the appendix but they do not totally clear this up for me.

Correctness: I did not see anything that stood out as false.

Clarity: A few English issues but generally very legible.

Relation to Prior Work: I did not see problems with this but someone more closely involved with this specific line of research would know better.

Reproducibility: Yes

Additional Feedback: As I said I do like the idea of "let's take the simple idea, we know there's some issue with it, but let's evaluate how bad it actually gets, maybe it's not even that bad." The only problem is that knowing that it's epsilon-BIC in my view doesn't really tell us how bad it gets because we still don't know how bidders will actually react. But this work could be a stepping stone to understanding that. I also appreciate that this is a theoretical paper and I'm not asking to see evidence of real-world deployment or anything like that. But I would like to get a better conceptual understanding. ***********After rebuttal: I think the authors generally responded well. I'd still like to see at least a discussion of turning an approximately IC mechanism into an exact one. While in practical application the designer may not know the distribution, when conducting the experiments *we* know the distribution and can still use that to obtain a useful comparison, also in regard to whether benchmarks are reasonable.


Review 3

Summary and Contributions: The authors study the problem of strategic behavior in learning reserve prices in auctions. More specifically, the paper looks at ERM as a way to learn optimal reserve price in auctions, and consider strategic agents that may manipulate their reports to the ERM in order to affect (and lower) the output reserve price. The main contributions of the paper are the following: i) First, the authors characterize the robustness of ERM^c (a simple variant of ERM that restricts the range of prices that can be selected) to strategic behavior ii) The authors consider an application of this result to two settings. The first one is a two-phase model where i) the seller runs an exploration phase in the form of second price auctions with no reserve, ii) the seller uses the collected data to decide on a reserve price to use in the second phase. The second one is uniform price auctions (where a single price is posted for all bidders, and bidders with values of the price receive the item and pay the price).

Strengths: One of the main strength of the paper is to provide bounds on the incentive-awareness measure when m > 1. First, it seems only the case of m = 1 was understood in previous work, and even then, convergence was shown without providing rates. As such, the paper makes a significant contribution compared to previous work. I also think the applications, in particular the two-phased one, significantly strengthen the paper. Beyond bounding the incentive-awareness measure, an important question is to understand how these bounds help mechanism design, and translate into truthfulness and revenue optimality guarantees. The authors do so in a natural explore then exploit type of setting, where the mechanism designer first wants to learn an optimal reserve price, then use this price to maximize his revenue. There, the authors show how the bounds on incentive-awareness measure lead to approximate truthfulness guarantees in the first phase, and how to balance the first and the second phase to get truthfulness and near-revenue optimality (if the first phase is too short, it is easy for agents to manipulate and obtain a lower reserve price, which hurts the second phase; if the first phase is too long, the mechanism designer may miss out on a lot of revenue). I think such a result provides useful guidelines on how to balance exploration and exploitation in such mechanism design settings, in the presence of agents that anticipate future reserve prices. Finally, I do like that the paper provides results under different assumptions on the valuation distribution F. In particular, the authors look at two assumptions that are common in mechanism design, monotone hazard rate and regularity, and provide bounds and discussions for both of these assumptions.

Weaknesses: The main weakness of the paper comes in the fact that there is a large range of parameters for which the incentive-awareness measure bounds are lacking. First, the bounds only apply when m = O(\sqrt{N}), effectively ruling out situations in which there are few bidders but each bidder controls a constant fraction of the bids (so, in particular, repeated auctions with few bidders). Second, when m is close to \sqrt{N}, the rate at which the upper bound decreases is very slow. In turn, it seems that the current framework can only handle situations in which m is relatively small/converges relatively fast to 0 compared to \sqrt{N}. This may not be possible; however, since the lower bound does not have a dependency on m, there is a significant gap between the upper and lower bounds for m far from constant that is not resolved, and it would be nice to tighten said lower bound/understand the optimal dependency on m, which is the central novel aspect of the paper. A question that seems to be left open by the paper (and may possibly lead to better guarantees) is that of using other rules than ERM^c that explicitly aim to incentivize truthful behavior and provide better truthfulness guarantees, rather than taking ERM as a given and bounding how untruthful it is. It seems in particular that one could maybe try to apply one of the techniques known to reduce the dependency of the output on the input (for example, differential privacy would be a very natural notion to look at here; how would it compare to the current results?)

Correctness: I have not checked the details of the proofs carefully. However, the proof sketches provided by the author in the main body of the paper seem to hold at a high level.

Clarity: The paper is well written and easy to follow. The paper provides proper motivation of the problem, clear discussions of the results and the corresponding parameters, and an entire section dedicated to giving an overview of the proofs and techniques used in the paper (that are deferred to the full version, attached with the supplementary material), which helps build intuition on why their results hold.

Relation to Prior Work: Yes. In particular, the authors make clear that the idea of ERM^c was proposed in previous work, and that their contribution is novel in the sense that only the case when m= 1 was studied in previous work, and previous work was only establishing convergence but not providing rates for the incentive-awareness measures.

Reproducibility: Yes

Additional Feedback: Do the authors know how doing exploration and exploitation at the same time, rather than having first an explore phase then an exploit phase (as in the two-phased model), affects the results? Post-rebuttal: thanks to the authors for their response! I have lowered my score a bit, as I agree with some of the concerns brought up by reviewer 4. Since the paper requires m to be of the order of \sqrt{N} at most, it seems to me that the number of bidders has to grow with N in the regimes in which the results of the paper hold, so I am not 100% convinced that the constant of number bidders justification used in the feedback directly applies to the results of this paper. That said, I still think the paper should be accepted. I don't think the fact that good mechanisms already exist make the paper uninteresting; I think there is still significant value in studying how ERM does, as this seems (to me) to be a fairly natural algorithm to run.


Review 4

Summary and Contributions: This paper considers an auction setting where a seller runs a two stage model. First run a second price auction with no reserve for T_1 rounds, and then, for T_2 rounds, use the first phase bids to set reserve prices by ERM (the optimal reserves for the empirical wrt the past bids) and run a second price with reserves. The authors prove that this auction format is approximately strategyproof (\eps - BIC) and approximately optimal in terms of revenue, when bidders are iid and have MHR and/or bounded distributions. The authors also give various extensions (multi-unit, uniform price). All results crucially use an upper bound on the “incentive-awareness measure”, a measure on how much the behavior of a single bidder can affect the change in the reserve price (when ERM is used).

Strengths: I think the research question is very interesting and timely. Now that the community understands how to use samples to design good auctions, it is very important to ask to what extend the incentives are distorted by the sample collection. The incentive awareness measure (not introduced in this paper) is a very clean way to measure this effect, and this paper does give an interesting bound (Thm 1.3).

Weaknesses: In terms of applications, since the agents are iid and MHR, we know very good prior-independent, truthful and simple approximations (namely, a second price auction is 1-1/n, even). How does this compare to the epsilon from Theorem 2.1? This point needs to be discussed. In light of this observation, I can't say I was extremely impressed by any of the applications.

Correctness: I didn't thoroughly check the appendix, but as far as I can tell the statements seem theoretically sound.

Clarity: In terms of writing, the theorem statements are very dense and hard to parse and there's definitely a lot of room for improvement there.

Relation to Prior Work: The authors do a good job comparing to the literature around the incentive awareness measure, but since the setting is iid and mhr there should be a bit more discussion around the prior-independent results.

Reproducibility: Yes

Additional Feedback: UPDATE: I updated my score to reflect my opinion after the discussion and rebuttal. I am more positive towards this paper.

[Author Response · NeurIPS 2020]

Thanks for your comments!

Review 1: Thanks for your appreciation!

Review 2:

Figure 1: $\Delta_{N,m}^{\text{worst}}$

**Justification for approximate incentive compatibility.** It is true that
$\epsilon$ going to 0 in the limit doesn't mean the incentives are negligible in
practice. But this is exactly why we study the convergence rate of $\epsilon$.

Indeed, $\epsilon$-BIC is fragile in a game with a few players. But in our model,
there is large pool of bidders, so it is unrealistic for a bidder to find the
exact best response: she has to collect a lot of information about other
bidders' strategies, try many solutions, do a large amount of computation,
etc. When the cost of searching for a better response exceeds $\epsilon$, the bidder is better-off bidding truthfully. Even if the
bidder suspects that some other bidders may bid non-truthfully, our incentive-awareness measure $\Delta_{N,m}^{\text{worst}}$ still guarantees
that, as long as the total number of bids of all non-truthful bidders doesn't exceed $m$, no bidder can benefit a lot from
lying. So we think $\epsilon$-BIC is appropriate here. Also, we don't see the possibility of a (nontrivial) exact BIC mechanism
when a bidder can affect her future reserve price; no BIC pricing function exists even in a two-round auction.

**Revenue.** Surely, the optimal revenue will be higher than Myerson if we relax BIC to $\epsilon$-BIC. But to our knowledge, no
work has studied how much higher that will be. We conjecture that it will be $O(\epsilon)$, and our $(1 - \epsilon_2)$-revenue guarantee
will hold against this stronger benchmark with an $\epsilon_2$ that is larger than the current $\epsilon_2$, but still goes to 0.

**Transforming $\epsilon$-BIC to exact BIC mechanisms.** The current techniques cannot be applied here for two reasons: (1)
they require some knowledge of the type distribution, either the full distribution or samples, which is unavailable in our
model; (2) they are for one-shot games, but our two-phase model is a repeated game.

**Asymptotic bounds.** Fig. 1 shows $\Delta_{N,m}^{\text{worst}}$ for the uniform$[1,5]$ distribution with $m = 1, 2, 3$. $\Delta_{N,1}^{\text{worst}}$ goes below 0.05
already when $N = 150$, so a bidder controls a few bids cannot reduce the price by much in a medium sized market.
$\Delta_{N,m}^{\text{worst}}$ grows linearly in $m$ roughly, which agrees with our theorem.

Review 3:

**Range of $m$.** In the most extreme case where $m = N$ (there's only one bidder), Amin et al (2013) show that it's
impossible to learn a strategic bidder's value distribution to obtain near-Myerson revenue in repeated auctions unless the
bidder discounts her future utility; similarly, in our model, when $m > cN$, $\Delta_{N,m}^{\text{worst}}$ always equals 1, which means the
incentive property of ERM$^c$ is very bad. So we are mainly interested in the case where $m$ is a relatively small fraction
of $N$, i.e., $o(\sqrt{N})$. We don't know what happens when $\Omega(\sqrt{N}) \le m \le cN$; the current framework cannot handle that.
For the lower bound, thanks for suggesting considering the dependency on $m$! We were focused on $N$ previously.

**Rules other than ERM$^c$.** There are indeed some algorithms that use differential privacy, e.g., Abernethy et al
(NeurIPS'19). Abernethy et al's truthfulness and revenue bounds are better than ours. But their work is worse than ours
in three other aspects: (1) their truthfulness notion is weaker than our PBIC notion; (2) ERM$^c$'s running time is far less
than their algorithm's; (3) ERM$^c$ works for unbounded distributions while theirs only support bounded distributions
because they need to discretize the value space; (details are in Sec. 2.4 and App. C.4.) They also need $m = o(\sqrt{N})$ to
obtain approximate truthfulness and revenue optimality at the same time, as we do. We remark that our starting point is
to understand ERM since it is the most fundamental learning algorithm, rather than to design good algorithms.

**Exploration and exploitation at the same time.** Under the BIC truthfulness assumption, it's OK to continuously
update the reserve price in the second phase (rather than fixing it), which improves revenue without affecting truthfulness.
But for the Perfect Bayesian Incentive-Compatibility notion (see Sec. 2.4 and App. C.4), we cannot do exploration and
exploitation at the same time. Consider this example: Each auction has one bidder. In the first round bidder A submits
bid 1 and 1 is set to be the reserve price in the second round. In the second round, bidder B submits 0.9 and loses, so
she knows that the first round bid is greater than 0.9. Suppose bidder B will join the next round, then she can strategize
her bid based on the information "the first round bid $> 0.9$". So the belief of bidder B on bidder A's value distribution
is no longer $F$, and our argument fails. But doing exploration and exploitation separately is good (see App. C.4).

Review 4:

**Prior-independence.** In order to obtain good revenue in a second price auction, the number of bidders in the auction
needs to be large since it is a $1 - 1/n$ approximation. But in our two-phase model, the number of bidders in each
auction (denoted by $K_1$ and $K_2$ for the two phases, at 125) can be very small, e.g., 1 or 2. With so few bidders,
prior-independent mechanisms do not have good revenue. We remark that in our model, although the total number of
bidders is large, each auction may have only a few bidders because bidders do not participate in all rounds of auctions.

[Meta-Review · NeurIPS 2020]

This paper received a very long discussion after the rebuttal, and while there was some disagreement about how interesting the results were, it was ultimately decided that the paper has enough merit to be accept at NeurIPS this year, since it's a challenging problem with a nice solution. But do take into account the detailed comments of the reviewers when putting together your final version.